# *HOXA13* in etiology and oncogenic potential of Barrett's esophagus

Vincent T. Janmaat [1], Kateryna Nesteruk [1], Manon C. W. Spaander[1], Auke P. Verhaar[1], Bingting Yu[1], Rodrigo A. Silva [1], Wayne A. Phillips [2,3,4], Marcin Magierowski [1,5], Anouk van de Winkel[1], H. Scott Stadler[6], Tatiana Sandoval-Guzmán [7], Luc J. W. van der Laan [8], Ernst J. Kuipers [1], Ron Smits [1], Marco J. Bruno [1], Gwenny M. Fuhler[1], Nicholas J. Clemons [2,3] & Maikel P. Peppelenbosch [1✉]

Barrett's esophagus in gastrointestinal reflux patients constitutes a columnar epithelium with distal characteristics, prone to progress to esophageal adenocarcinoma. *HOX* genes are known mediators of position-dependent morphology. Here we show *HOX* collinearity in the adult gut while Barrett's esophagus shows high *HOXA13* expression in stem cells and their progeny. *HOXA13* overexpression appears sufficient to explain both the phenotype (through downregulation of the epidermal differentiation complex) and the oncogenic potential of Barrett's esophagus. Intriguingly, employing a mouse model that contains a reporter coupled to the *HOXA13* promotor we identify single HOXA13-positive cells distally from the physiological esophagus, which is mirrored in human physiology, but increased in Barrett's esophagus. Additionally, we observe that *HOXA13* expression confers a competitive advantage to cells. We thus propose that Barrett's esophagus and associated esophageal adenocarcinoma is the consequence of expansion of this gastro-esophageal *HOXA13*-expressing compartment following epithelial injury.

[1] Department of Gastroenterology and Hepatology, Erasmus MC - University Medical Center Rotterdam, Rotterdam, The Netherlands. [2] Division of Cancer Research, Peter MacCallum Cancer Centre, Melbourne, VIC, Australia. [3] Sir Peter MacCallum Department of Oncology, The University of Melbourne, Melbourne, VIC, Australia. [4] Department of Surgery (St. Vincent's Hospital), The University of Melbourne, Melbourne, VIC, Australia. [5] Department of Physiology, Faculty of Medicine, Jagiellonian University Medical College, Cracow, Poland. [6] Department of Skeletal Biology, Shriners Hospital for Children, Portland, OR, USA. [7] DFG-Center for Regenerative Therapies, Technische Universität Dresden, Dresden, Germany. [8] Department of Surgery, Erasmus MC - University Medical Center Rotterdam, Rotterdam, The Netherlands. Share senior authorship: Gwenny M. Fuhler, Nicholas J. Clemons & Maikel P. Peppelenbosch. ✉email: m.peppelenbosch@erasmusmc.nl

Barrett's esophagus (BE) and gastric intestinal metaplasia (IM) are important risk factors for adenocarcinoma of the esophagus and stomach. In the esophagus, the chronic inflammation associated with gastroesophageal reflux disease (GERD) is believed to lead to Barrett's esophagus (BE), a crypt-structured columnar epithelium with distal gastrointestinal (GI)-tract characteristics, located just above the gastro-esophageal junction (GEJ). BE is a precursor lesion for esophageal adeno-carcinoma (EAC)[1,2], a disease which has shown a strong increase in incidence in the past decades. Analogously, *H. pylori*-infection can degenerate into atrophic gastritis and gastric IM, which in turn can progress into gastric cancer, the third leading cause of cancer-related death[3]. Similarly, while absolute risk is low, het-erotopic tissues in Meckel's diverticula and gastric inlet patches of the proximal esophagus represent relatively high-risk regions for adenocarcinoma comparatively to other sites of the ileum and proximal esophagus, respectively[4,5]. Therefore, a deeper under-standing of the biology of BE and gastric IM is necessary for designing rational avenues for the prevention and treatment of GI cancers.

BE is characterized by the presence of cells with a caudal intestinal phenotype at a rostral location. Therefore, dysregulation of positional specification is likely involved in the etiology of BE. Regulation of rostral-caudal patterning of specialized tissue in embryology and adulthood is to a large extent dependent on the concerted action of two evolutionary highly conserved gene systems, the Caudal-related Homeobox (*CDX*) transcription factor gene family and the genes of the Homeobox (*HOX*) cluster. A substantial research effort has been invested in investigating the role of *CDX* genes in positional misspecification in BE[6]. However, these efforts have not yielded convincing evidence that these genes are the principal mediators of the distal phenotype in this disease[7,8]. Intriguingly, however, a microarray-based gene expression study of BE suggested potential misregulation of the *HOX* gene family in BE[9]. *HOX* genes are linked to morphological transformations and neoplasia[10,11]. Four clusters of *HOX* genes, *HOXA* to *HOXD*, have been defined. The 3′ to 5′ sequence of *HOX* gene paralogues cor-responds to the sequence in which they act along the rostrocaudal axis. This property is termed collinearity and links clustering to function. Previously, a *Hox* expression gradient was found along the murine embryonic gut[12]. Ectopic *Hox* expression in mice can alter intestinal differentiation[8]. A *HOX* gradient along the adult human gut has also been reported[13], but that study involved pooling full thickness gut specimens, limiting data interpretation. Nevertheless, we feel that there is sufficient evidence to prompt exploring the function of *HOX* gene expression with respect to positional identity in physiology and pathology of the GI-tract in general and in BE in particular.

Here, we show that single cells of the upper GI tract express the distal gene *HOXA13*, that their number is upregulated in BE and that HOXA13 conveys phenotypic metaplasia and increases proliferation.

## Results

**HOX cluster gene expression in the GI tract is collinear in men and mice**. Investigating *HOX* gene mRNA expression in the murine and human gastrointestinal tract, we observed collinearity that is similar in adult humans and mice (Fig. 1a for human *HOXA*, Supplementary Fig. 1 for all *HOX* genes and Supple-mentary Fig. 2 for graphical presentation of the studied *HOX* clusters and the locations of biopsies taken along the human ($n = 3$) and mouse ($n = 4$) GI tract). The highest *HOX* gene cluster expression was observed in the colon, except for the *HOXC* cluster. For individual paralogues, there is a higher expression of 5′ *HOXA/B* genes in the distal GI-tract from *HOXA5/B5* onward.

Of all *HOXA* paralogues, expression of *HOXA13* was highest and restricted to the colon (Fig. 1a). *HOXA13* expression is regulated by LncRNA *HOTTIP*, which is located 5′ to *HOXA13*[14]. Accordingly, *HOTTIP* and *HOXA13* share a similar expression pattern (Supplementary Fig. 1a). For *HOXD*, all paralogue genes have increased expression in the distal colon, while *HOXC* expression is mainly localized in the proximal and ileal regions. Thus, *HOX* gene expression is linked to positional identity in the mammalian gut, and collinearity is particularly strong for the *HOXA/B* paralogues.

Subsequently, we addressed the question as to whether GI HOX coding is already present at the GI stem cell stage, or is established only upon the formation of differentiated deriva-tives. For this, we used publicly available data published by Wang et al. which contains the mRNA expression of human stem cells isolated from the GI-tract and either cultured as stem cells or differentiated in an air-liquid interface (ALI)[15]. Analysis of this data shows that *HOX* gene expression patterns in stem cell and ALI cultures are similar. *HOXA* and *B* cluster genes have a significantly higher expression in the large intestine as compared to the small intestine, in particular 5′ *HOXA* genes including *HOXA13* (Fig. 1d for *HOXA* genes, for clusters *HOXB*, *C*, and *D* see Supplementary Fig. 3). No clear regulation of the *HOXC* or *D* clusters is seen in this in dataset, with exception of an upregulation of *HOXC10* in the large intestine. Hence, *HOX* coding is an inherent feature of the location-specific stem cell and is maintained in its derivatives.

**HOXA13 in BE, GI heterotopias and GI cancers**. As positional phenotype is linked to HOX status in physiology, we subse-quently characterized *HOX* mRNA expression in several metaplastic tissues known to assume the morphological phe-notype of other intestinal locations, as well as their sequelae. BE shows upregulation of *HOXA10*, *11*, and *13*, and *HOXB 6, 7, 9*, and *13* mRNA by qPCR when compared to the normal squamous esophagus (Fig. 1b and Supplementary Fig. 4), which closely resembles colonic *HOXA* and *B* expression patterns. High 5′ *HOXA* gene expression is also present in columnar-lined esophagus without goblet cells (CLE; a BE-related con-dition), esophageal adenocarcinoma (EAC), and IM of the stomach (Fig. 1b, c). In accordance with a regulatory role for *HOTTIP* on *HOXA13* expression, we find that *HOTTIP* is also overexpressed in BE, and correlates with *HOXA13* expression patterns (Supplementary Fig. 5a–c). *HOTAIR*, a lncRNA located in the *HOXC* cluster and associated with chromatin reprogramming in cancer progression[16] is upregulated as well (Supplementary Fig. 5d–f)[14]. We concluded that BE, EAC and various metaplasias with caudal histo-morphological char-acteristics have *HOXA* and *HOXB* expression patterns typical of the caudal GI-tract, with upregulation of *HOXA13* expres-sion being the prominent feature. Heterotopias, namely the gastric inlet patch in the proximal esophagus and heterotopia of the Meckel's diverticulum, are tissues which have a phy-siological appearance, but are normally found in a different location. Both these heterotopias are characterized by abun-dant *HOXA13* mRNA expression (Fig. 1c), although intrigu-ingly the direction of epithelial metaplasia for Meckel's diverticulum is of an anterior rather than posterior phenotype, indicating an exception from the pattern in case of Meckel's diverticulum. One of the existing hypotheses on the cell of origin of BE states that BE may arise from cells with progenitor properties that are able to give rise to a variety of cell types[17]. To investigate whether aberrant HOX gene expression in BE is established at the level of the epithelium-specific stem cell, we interrogated the publically available data of Yamamoto

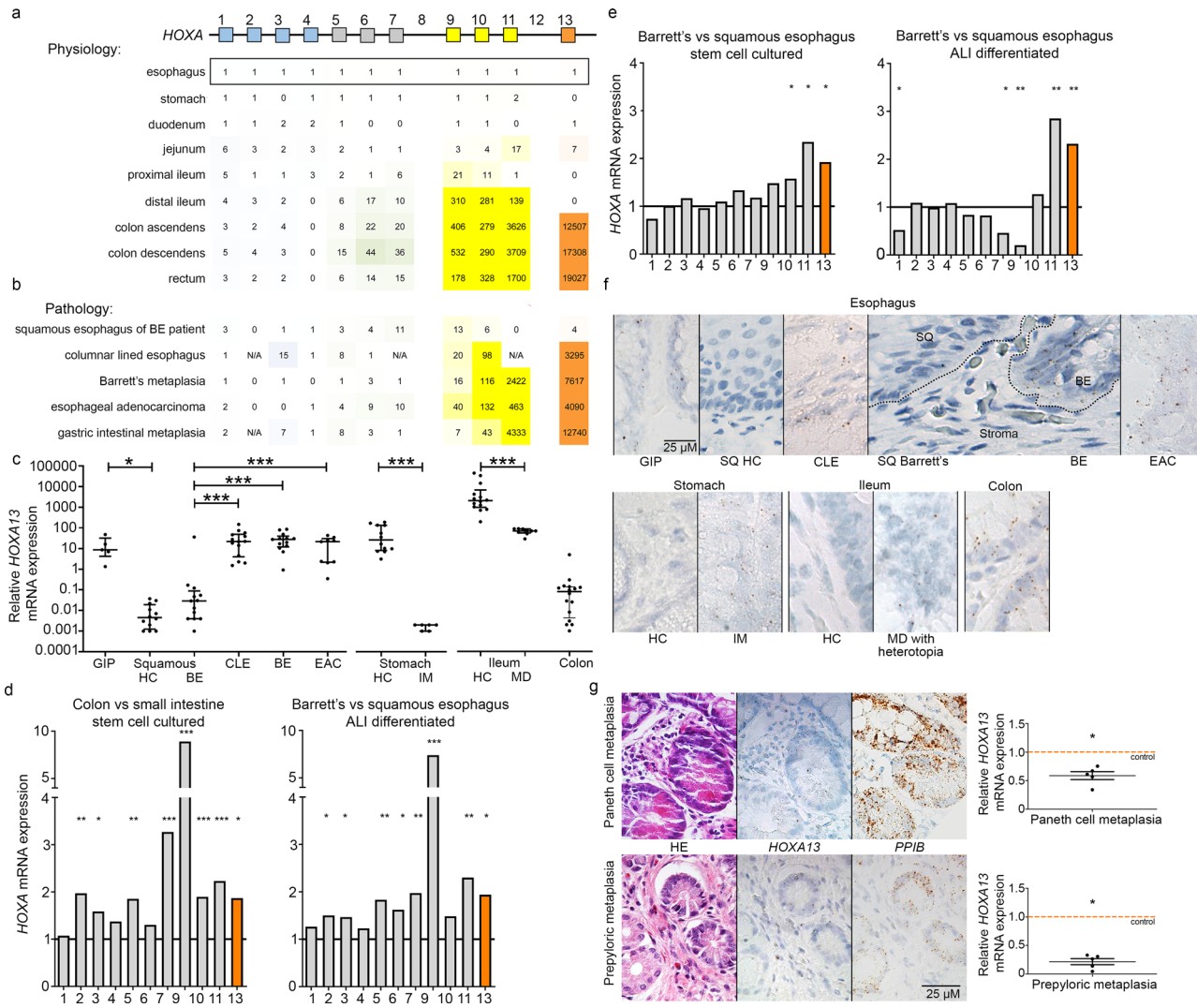

**Fig. 1 HOXA cluster gene expression shows collinearity along the adult gastrointestinal tract but is deregulated in Barrett's esophagus (BE), various metaplasias and esophageal adenocarcinoma (EAC). a** HOXA cluster genes are collinearly expressed along the gastro-intestinal (GI)-tract of adult humans ($n = 3$). Numbers represent mRNA fold changes relative to the esophagus and thus can be compared within each HOXA paralogue member but not between them. **b** HOXA cluster gene expression in the squamous esophagus of BE patients ($n = 13$), columnar lined esophagus (CLE) ($n = 14$), BE ($n = 13$), EAC ($n = 12$), and gastric intestinal metaplasia (IM) ($n = 12$) is characterized by an upregulation of 5′ HOXA genes. Numbers represent mRNA fold changes relative to the esophagus of healthy individuals. **c** HOXA13 expression quantified by qPCR in BE, CLE, IM of the stomach, and heterotopias along the GI tract with their corresponding physiological epithelia. Squamous epithelium (SQ) Barrett's and BE are derived from the same person ($n = 13$). Gastric inlet patch (GIP; $n = 5$); healthy control (HC) squamous esophagus ($n = 12$); CLE ($n = 14$); BE ($n = 13$); EAC ($n = 12$); stomach ($n = 14$); gastric IM ($n = 12$); ileum ($n = 6$); Meckel's diverticulum (MD) with gastric heterotopia ($n = 14$), and colon ($n = 9$). Median ± IQR, *$p < 0.05$; **$p < 0.01$; ***$p < 0.001$. For esophagus, Kruskal–Wallis test with Dunn's multiple comparisons test (SQ healthy vs. GIP, $p = 0.015$; SQ healthy vs. CLE, $p < 0.0001$; SQ healthy vs. BE, $p < 0.0001$; SQ healthy vs. EAC, $p = 0.0009$). For stomach and ileum, Mann–Whitney test (two-tailed), $p < 0.0001$. **d** HOXA cluster genes, in particular 5′ HOXA genes including HOXA13, have a higher expression in the large intestine ($n = 3$ in technical duplicate) compared to the small intestine ($n = 3$ in technical duplicate), in both stem cells (left panel) and differentiated cells (right panel). Normalization was performed by setting mRNA expression to 1 for the small intestine. *$p < 0.05$; **$p < 0.01$; ***$p < 0.001$. This figure includes no estimate of variance as the empirical Bayes-moderated two-sided t-statistic was used which does not generate a standard error. **e** 5′ HOXA cluster gene expression in BE is higher compared to the squamous esophagus in stem cell and air-liquid interface (ALI) differentiated cultures. $n = 12$ (BE) versus $n = 2$ (squamous esophagus) in technical duplicates are depicted for stem cell cultures and $n = 1$ each for ALI differentiated samples in technical duplicates. Normalization was performed by setting mRNA expression to squamous esophagus. *$p < 0.05$; **$p < 0.01$; ***$p < 0.001$. This figure includes no estimate of variance as the empirical Bayes-moderated two-sided t-statistic was used which does not generate a standard error. **f** Deregulation of HOXA13 expression in gastrointestinal tract pathology as evaluated with RNA in situ hybridization in clinical samples. HOXA13 is upregulated in IM and heterotopia and downregulated in pyloric and Paneth cell metaplasia in the colon. One sample of each tissue type was analyzed. **g** Downregulation of HOXA13 expression (corrected for peptidylprolyl isomerase B - PPIB expression) relative to adjacent non-metaplastic tissue, was observed for Paneth cell metaplasia ($n = 5$; FC 0.59; $p = 0.0003$) and pyloric metaplasia ($n = 5$; FC 0.22; $p = 0.0001$) (lower panels). Unpaired t-test (two-tailed). Representative images of hematoxylin and eosin (HE) staining, HOXA13 RNA-scope, and PPIB reference gene RNA-scope of Paneth cell metaplasia (from the colon) present in two glands to the bottom right (upper panels) and pyloric metaplasia (from the colon) in the top left two glands (middle panels) are shown.

et al.[15,18]. *HOX* gene expression patterns in squamous esophageal and BE stem cells as well as their respective ALI-differentiated derivatives were retrieved. HOX gene expression in stem cell cultures from these locations is similar to their ALI differentiated counterparts (Fig. 1e, Supplementary Fig. 3). In BE stem cells, an upregulation of 5′ *HOXA* genes (Fig. 1e) as well as *HOXB6*, *7*, *13*, and *HOXC10* is seen (Supplementary Fig. 3), reaching levels similar to those observed in the colon. Thus, alternative HOX coding associated with BE is established at the epithelium-specific stem cell level and is maintained in derivatives of the stem cells involved.

According to the collinearity theory, a paralogue group 13 member is more likely to confer the distal characteristics seen in BE as compared to more anterior paralogue group members[19]. Of the paralogue group 13 members, *HOXA13* and *HOXB13* are overexpressed in BE, with *HOXA13* showing much higher expression compared to *HOXB13* in BE, EAC, and IM of the stomach (Supplementary Fig. 4). Therefore, while HOX genes such as *HOXA11, B6, B9*, and *B13* are also potentially interesting candidates, here we chose to focus on the *HOXA13* gene for further in-depth analysis of different metaplastic tissues. As immunohistochemistry for *HOXA13* was unsuccessful, (two anti-HOXA13 antibodies were tested, but lacked specificity) we resorted to in situ hybridization (ISH) for *HOXA13* to further confirm the observed atypical expression of this gene in different tissues (examples shown in Fig. 1f). Metaplasia is found throughout the GI-tract. While BE and IM acquire a more distal phenotype, distally located colonic pyloric and Paneth cell metaplasia, related to inflammatory bowel disease, acquire a more rostral phenotype[20]. Accordingly, downregulation of *HOXA13* expression (corrected for *PPIB* expression as a reference gene) relative to adjacent non-metaplastic tissue, was seen for these tissues (Fig. 1g, again supporting a role for *HOXA13* in positional identity).

**Binary regulation of *HOXA13* expression.** To study in more detail which of the cells in the healthy GI-tract express Hoxa13, we employed a murine model in which the endogenous mouse Hoxa13 promoter drives the expression of a Hoxa13-GFP fusion protein. Within the epithelial compartment, the proximal expression border is located at the transition from the distal to the proximal colon as can be seen from fluorescent images and images of anti-GFP IHC staining (Fig. 2a, b and Supplementary Fig. 6a–d for bigger overview images). This proximal expression border seems to be crypt-clonal, with some crypts expressing Hoxa13 and others not (see arrows in Fig. 2b and close-up in Fig. 2c). Functional consequences of this clonality are unknown and, while beyond the scope of the present manuscript, present an interesting biological question. The distal Hoxa13-GFP expression is limited by the anal squamocolumnar junction (SCJ; Supplementary Fig. 6e, please note this cannot be appreciated in Fig. 2a, as this part was damaged for this mouse). To investigate whether these local gradients of Hoxa13 expression are also present in humans, *HOXA13* mRNA expression was assessed by qPCR in an additional set of biopsies taken from different colonic locations. Cecal biopsies are *HOXA13* negative, while *HOXA13* expression increases from the ileocecal valve to the distal transverse colon, demonstrating a similar expression pattern as observed in the mouse (Fig. 2d).

In addition to a Hoxa13 gradient along the GI tract, epithelial Hoxa13-GFP expression is also tightly regulated along the baso-luminal axis of individual crypts. Proximally, only apical expression is seen, while distally Hoxa13-GFP is expressed along the entire baso-luminal axis of the crypts (Fig. 2e). In addition, mesenchymal expression is observed in the cells just beneath the

epithelium in the proximal colon (Fig. 2e). Within the cell, the strongest signal is co-localized with nuclei, as expected, but cytoplasmic staining is also seen which can be explained by ribosomal synthesis (Fig. 2e).

We concluded that spatial regulation of HOXA13 expression is very precise, robust and colon-specific, raising questions as to the cellular origin of the HOXA13 expression observed in BE.

**Individual Hoxa13/*HOXA13*-positive cells in the upper GI tract.** No significant expression of *HOXA13* mRNA was seen in the squamous esophagus of BE patients by qPCR (Fig. 1c, e), suggesting that GERD does not provoke *HOXA13* expression per se. Indeed, when two primary immortalized squamous esophageal cell lines (EPC2-hTERT and HET-1A) were exposed to either bile or acid, only minor effects on *HOXA13* expression were observed (two to fourfold from a low baseline expression; Fig. 3a, b), more in agreement with cells having a relatively high *HOXA13* expression showing better survival of the treatment rather than upregulation of expression per se. This was confirmed by analysis of the publicly available single cell RNAseq database recently published by Owen et al.[21]. Results at single cell level demonstrate the presence of a small population of *HOXA13*-positive cells in the normal squamous esophagus of BE patients (8%). In BE tissue, the percentage of these HOXA13-positive cells increase to 30%, but their individual *HOXA13* mRNA levels are not increased as compared to HOXA13-expressing cells of the normal esophagus (Fig. 3c, d). Similarly, the number of *HOXA13*-positive cells, but not *HOXA13* expression *per cell*, is increased in IM of the stomach, early gastric cancer, and colorectal cancer (Fig. 3c, d). Thus, we further investigated Hoxa13 at the cellular level in our samples. Although *HOXA13* mRNA expression was detectable in only one of four mice in the upper GI-tract by qPCR (Supplementary Fig. 2), detailed inspection of specimens involved did identify single Hoxa13-positive cells in the stomach of Hoxa13-GFP mice by immunohistochemistry. Such signal was present at the basolateral side along the stomach starting from the GEJ, but not seen in the squamous cells along the esophagus, nor the stroma (Fig. 4a and Supplementary Fig. 7). This is of particular interest as the GEJ has been suggested as a place of origin of BE[17]. A littermate negative for Hoxa13-GFP showed no positivity (Supplementary Fig. 6d, Supplementary Fig. 7d). Subsequently, we employed ISH for *HOXA13* on surgical samples from the human GEJ of three adult patients to analyze the presence of *HOXA13*-expressing cells in the human upper GI tract. In all three specimens, the GE junction area contained a clear positive signal for *HOXA13* mRNA, some signal was seen in cells of the proximal stomach, while signal was even lower in the squamous epithelium and stroma (Fig. 4b and Supplementary Fig. 8a, b). Esophageal submucosal glands (ESMG)[21] were present in one sample and were *HOXA13* positive (Fig. 4c). Interestingly, ESMG were also highly positive for $KRT7^+KR5^+TP63^+$ cells (previously postulated as the cell of origin BE origin in GEJ[22]) although unlike HOXA13+ cells, $KRT7^+KR5^+TP63^+$ triple positive cells were not identified in the stomach (Fig. 4d). (Of note, this is showed for one sample and we were unable to assess possible *HOXA13* co-expression with these triple positive cells due to absence of specific HOXA13 antibodies). We also studied *HOXA13* expression in the GEJ of three spontaneously aborted human fetuses of 17–20 weeks of age, a gestation period characterized by transition of the esophageal epithelium from columnar to a squamous phenotype. We observed high and specific *HOXA13* expression at the gastric cardia, while more distal stomach and esophageal epithelium were less positive (Fig. 4e, Supplementary Fig. 8c, d). These data imply that *HOXA13*-positive cells are present in the human embryonic

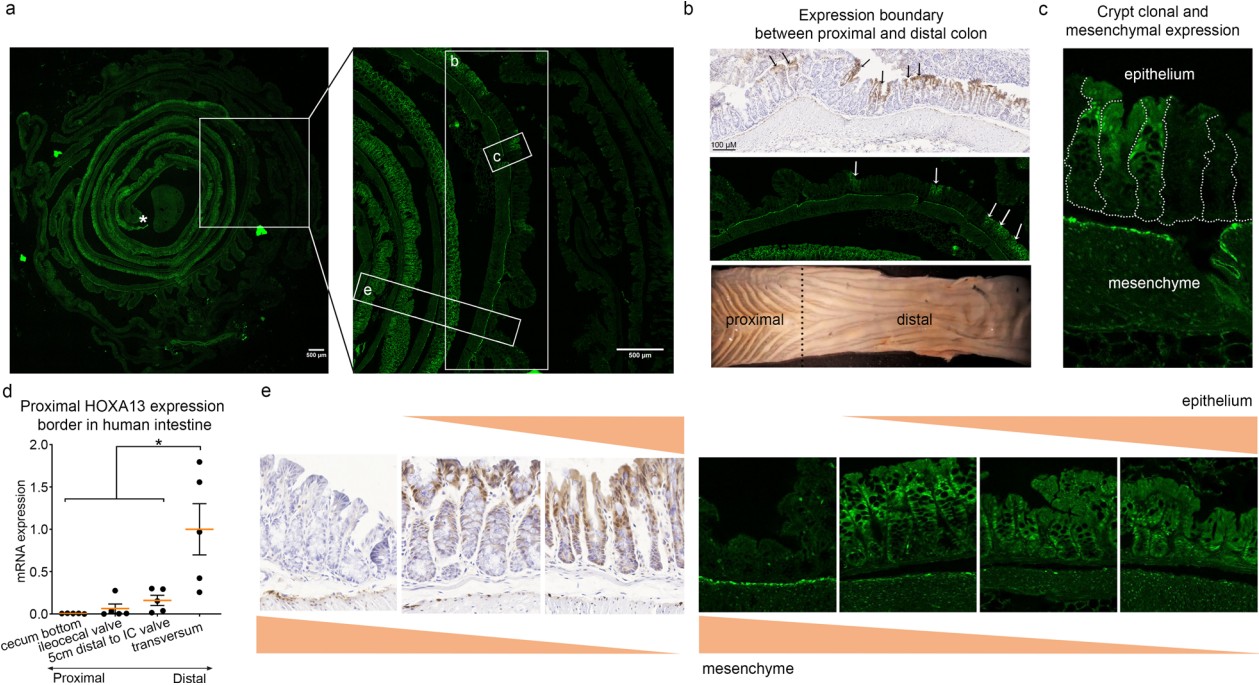

**Fig. 2 Murine and human *HOXA13* expression is subject to strict spatial control in the colon. a** A representative example from 3 mice of a "Swiss roll" configuration of the large intestine of the Hoxa13-GFP heterozygous mouse model. An asterisk indicates the most distal portion of the epithelium. Magnification of the insets are shown in panels **b**, **c** and **e**. **b** The proximal border of physiological Hoxa13 expression in the adult mouse is patchy and located between the proximal and distal colon, indicated by a black dashed line in the bottom panel (macroscopic image of an opened mouse colon). Representative images of anti-GFP IHC and confocal microscopy are shown. Arrows indicate crypts that are positive for Hoxa13 among Hoxa13-negative crypts. **c** The Hoxa13 expression is crypt clonal. This is observed for $n = 1$. **d** In adult humans the cecum bottom is negative for *HOXA13* while positivity increases distally ($n = 5$ independent sampes). Mean ± SEM, *$p < 0.05$, repeated measures ANOVA with Holm-Šídák's multiple comparisons test, $p = 0.001$. *HOXA13* mRNA levels were normalized to levels in the transverse colon. **e** Hoxa13 expression is tightly regulated along the baso-luminal axis. Distally, Hoxa13 is expressed along the entire baso-luminal axis of the colonic crypts, proximally only expression at the luminal side is seen. In addition, a mesenchymal expression is observed in the cells just beneath the epithelium, predominantly in the proximal colon. Anti-GFP IHC and confocal images are shown.

esophagus during the epithelial transition period, reduced in adult squamous esophagus, and increase again in BE. Thus, the epithelium of both the human and mouse adult upper GI tract, in particular the GEJ and ESMGs for human, is characterized by the presence of a subpopulation of *HOXA13*/Hoxa13-positive cells in an otherwise *HOXA13*/Hoxa13 negative surrounding.

**HOXA13 affects differentiation potential and posteriorizes.** Having established that individual HOXA13-positive cells reside in the physiological upper GI tract and are enhanced in BE tissue, we next set out to investigate the potential role of this population of cells in the etiology of BE. To this end, we further analyzed the single cell RNA-seq[21] data set mentioned above. In this study, the GEJ was not sampled for analysis. However, the 8% of cells of the normal esophagus that express *HOXA13* exhibit transcriptional overlap with cells derived from BE tissue as seen from the t-Distributed Stochastic Neighbour Embedding (t-SNE) plot (Fig. 5a). Gene expression analysis indicates that these cells are derived from ESMG (Fig. 5b and Supplementary Data 1)[21,23]. Specifically, and in contrast to the *HOXA13*-negative cell population, >70% of the *HOXA13*-positive cells from the normal squamous mucosa are positive for submucosal markers *LEFTY1* and *OLFM4*, designated ESMG markers, which have also been described as markers of BE progenitor cells[21]. Additionally, *HOXA13*-positive cells express mucosal markers *TFF3, Lyz* and *SOX9*, as well as columnar and BE markers *TFF1, KRT7, VIL1, MUC5B, MUC3A, MUC13, MUC1*, and *CEACAM5*, while being negative for keratinization marker *IVL* and basal epithelial cell

marker *p63* (Fig. 5b and Supplementary Data 1 for the list of genes enriched in *HOXA13*-positive cells). In BE, the percentage of cells positive for these columnar and ductal markers increase also in the *HOXA13*-negative population, suggesting either that upon differentiation some of these cells might lose *HOXA13* expression, or that there is more than one population giving rise to BE tissue. This would be in line with mouse data, as the murine esophagus lacks ESMGs and Hoxa13-positive cells. Interestingly, although rare in this dataset, within the *TFF3+* population four cells were identified to be triple positive for *KRT14* (a gene pair with *KRT5*), *TP63* and *KRT7*[21] but these were not positive for *HOXA13*.

The cell of origin with respect to formation of the BE segment should be able to generate a variety of differentiated cell types that exhibit colonic, gastric, pancreatic acinar or other phenotypes[24,25]. In chick embryos, *HOXA13* regulates regionalization after 1.5 days of development, showing the involvement of *HOXA13* in early differentiation, consistent with an effect of this gene on cellular phenotype in such pluripotent progenitor cells[26]. In an effort to experimentally test the influence of *HOXA13* on cell fate, we generated *HOXA13*-inducible pluripotent mouse embryonic stem cells (mESCs). These pluripotent mESCs can be efficiently differentiated to multipotent definitive endoderm, as determined by membrane expression of CXCR4 and E-cadherin (Fig. 5c). This was further confirmed by RNAseq, showing a strong upregulation of definitive endoderm markers such as *Sox17* and *Foxa1* in these differentiated cells, while pluripotency markers such as *Nanog* are downregulated (see

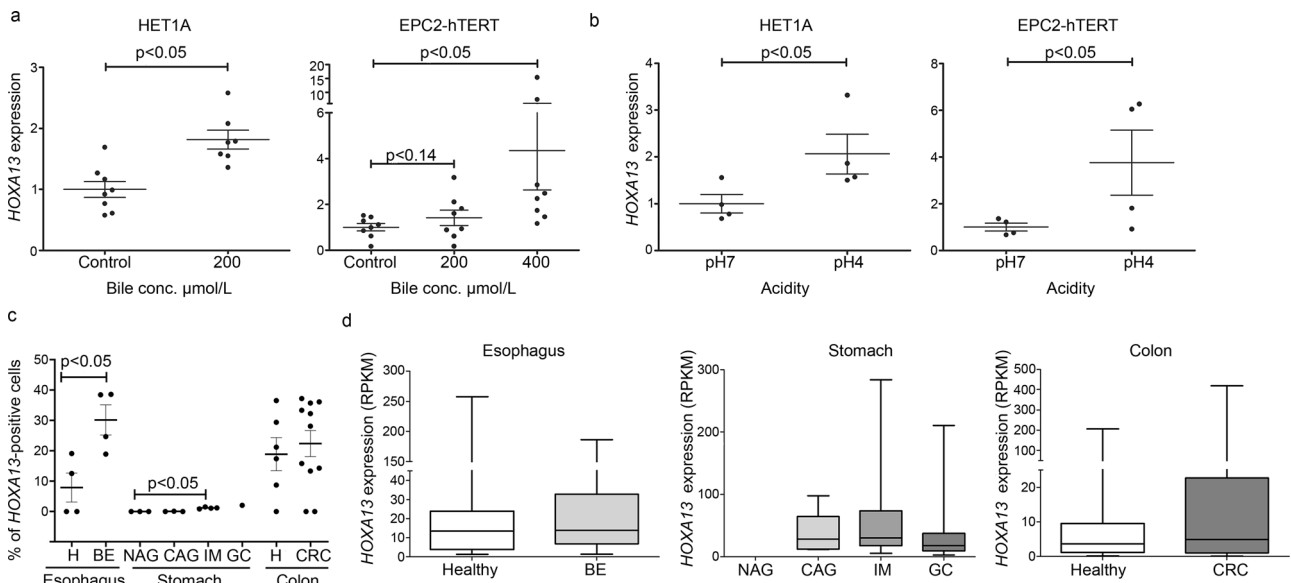

**Fig. 3 Number of *HOXA13* + cells rather than cellular expression levels are associated with metaplasia.** Exposure to bile (at pH 7) (**a**) or to acid (**b**), in two in vitro model systems of gastroesophageal reflux disease (GERD), marginally induces the expression of *HOXA13* from low baseline expression levels in two primary immortalized squamous esophageal cell lines. Error bars represent the 95% CI of the mean. $n = 4$ independent experiments. For Het1a, two-tailed *t*-test was used ($p = 0.0096$ in a, $p = 0.0322$ in b), for EPC2-hTERT Dunn's multiple comparisons test in a (control vs. 200 μM, $p > 0.99$; control vs. 400 μM, $p = 0.0112$; 200 μM vs. 400 μM, $p = 0.14$), one-tailed *t*-test in **b**, $p = 0.0486$. **c** The number of *HOXA13*-postive cells are increased in Barrett's esophagus (BE) and intestinal metaplasia (IM) as compared to normal esophagus or stomach tissue. Healthy and BE esophageal samples are derived from the same patients. Mean ± SEM are shown. $n = 4$ individuals for esophagus, $n = 3$ for NAG, CAG, $n = 4$ for IM, $n = 1$ for GC, $n = 6$ for healthy colon, $n = 11$ for CRC. Graphs are based on the analysis of single cell RNA data seq[21], GSE134520[82], GSE81861[83]. Two-tailed *t*-test was used for esophagus ($p = 0.0443$) and colon ($p = 0.6808$), one-way analysis of variance for stomach with Dunnett's Multiple Comparison Test ($p < 0.0001$). **d** *HOXA13* expression level per cell is unchanged. Expression level presented in *HOXA13*+ cells only. *H* healthy, *BE* Barret's esophagus, *NAG* non-atrophic gastritis, *CAG* chronic atrophic gastritis, *IM* intestinal metaplasia of stomach, *GC* early gastric cancer, and *CRC* colorectal cancer. $n = 37$ for H, $n = 132$ for BE, NA for NAG, $n = 5$ for CAG, $n = 163$ for IM, $n = 69$ for GC, $n = 37$ for healthy colon, $n = 87$ for CRC of single cells from the individuals mentioned in (c). For GC, statistics are not presented as data per patient was not provided. Boxplots with middle line is the median, the lower and upper hinges correspond to 25th to 75th percentiles, and whiskers representing min-max values.

Supplementary Table 1). Using ingenuity pathway analysis (IPA) to further analyze differently expressed genes, a positive association was found with "differentiation of embryonic cells" ($z = 1.82$, $p = 6.38 \times 10^{-15}$). Intriguingly, when *HOXA13* expression was induced, cells differentiated less effectively towards definitive endoderm as determined by CXCR4+/E-cadherin+ expression and morphological assessment (Fig. 5c, d). Consistent with a reduced unilinear differentiation, clones expressing *HOXA13* showed greater expansion (Fig. 5d).

We next contrasted the transcriptome of non-differentiated, pluripotent *HOXA13*-overexpressing and control cultures to identify potential molecular mediators of the *HOXA13* effects observed. Results of IPA analysis of differential gene expression are broadly consistent with *HOXA13* conferring a pluripotent phenotype. Specifically, forced *HOXA13* expression results in upregulation of the "role of *Nanog* in mammalian embryonic cell pluripotency" category ($z = 1.34$, $p = 2.32 \times 10^{-3}$), an effect that involves *Sox2*, *Nanog*, *Tbx3*, *Hesx-1*, and *Dppa*-1 amongst others[27,28] (See Table 1 for more details/results, fold changes, and *q*-values with regard to this experiment). *HOXA13* expression also appears to downregulate Wnt signaling, possibly through BMP signaling[29]. Wnt signaling is known to promote mesoendodermal differentiation[30], these results are consistent with *HOXA13*-mediated downregulation of Wnt signaling during axial elongation[31]. Thus, the transcriptional profile provoked by *HOXA13* is consistent with maintaining a relatively pluripotent phenotype which in turn may increase compartment expansion.

*HOXA13* expression does not block endodermal differentiation of mESC cells completely, suggesting that a role for *HOXA13* in

this compartment is still relevant. Definitive endoderm is a feature of the entire GI tract epithelium, and does not distinguish upper and lower GI epithelium per se. To investigate the role of *HOXA13* in this cell compartment and test our prediction that *HOXA13* expression would predispose endoderm to acquire distal phenotypes, we sorted CXCR4+/E-cadherin+ cells of *HOXA13* positive and negative cultures and contrasted their mRNA expression. *HOXA13* upregulates gene expression associated with determination of morphology in definitive endoderm cells. In IPA analysis, "actin cytoskeleton signaling" was most activated ($z = 3.00$, $p = 3.74 \times 10^{-2}$). "RhoA signaling", which stimulates actin polymerization, ($z = 2.12$, $p = 1.12 \times 10^{-2}$) was also stimulated. *HOXA13* supports distal epithelial functions with upregulation of microvillus-associated genes, *Ezr* and *Vill*, keratins, *Krt19* and *Krt20*, tetraspan network genes, *Igsf8*, and exocrine function associated genes such as *Gcnt3*, normally expressed in the distal GI-tract epithelium[32]. In addition, more transcripts of "Cell proliferation of carcinoma cell line" ($z = 1.13$, $p = 1.5 \times 10^{-6}$) and "Neoplasia of cancer cells" ($z = 1.13$, $p = 2.22 \times 10^{-4}$) categories, such as *Fgfr2* and *Nek2*, were detected. Thus, forced *HOXA13* expression during endodermal differentiation supports caudal epithelial functions and proliferative potential (see Table 1 for fold changes and *q*-values; see Supplementary Data 2 for additional relevant molecules).

Together, these data are in apparent agreement with *HOXA13*-expressing cells displaying a progenitor phenotype and having a competitive advantage, while simultaneously driving the acquisition of a more distal columnar phenotype once committed to differentiation (see Fig. 5e).

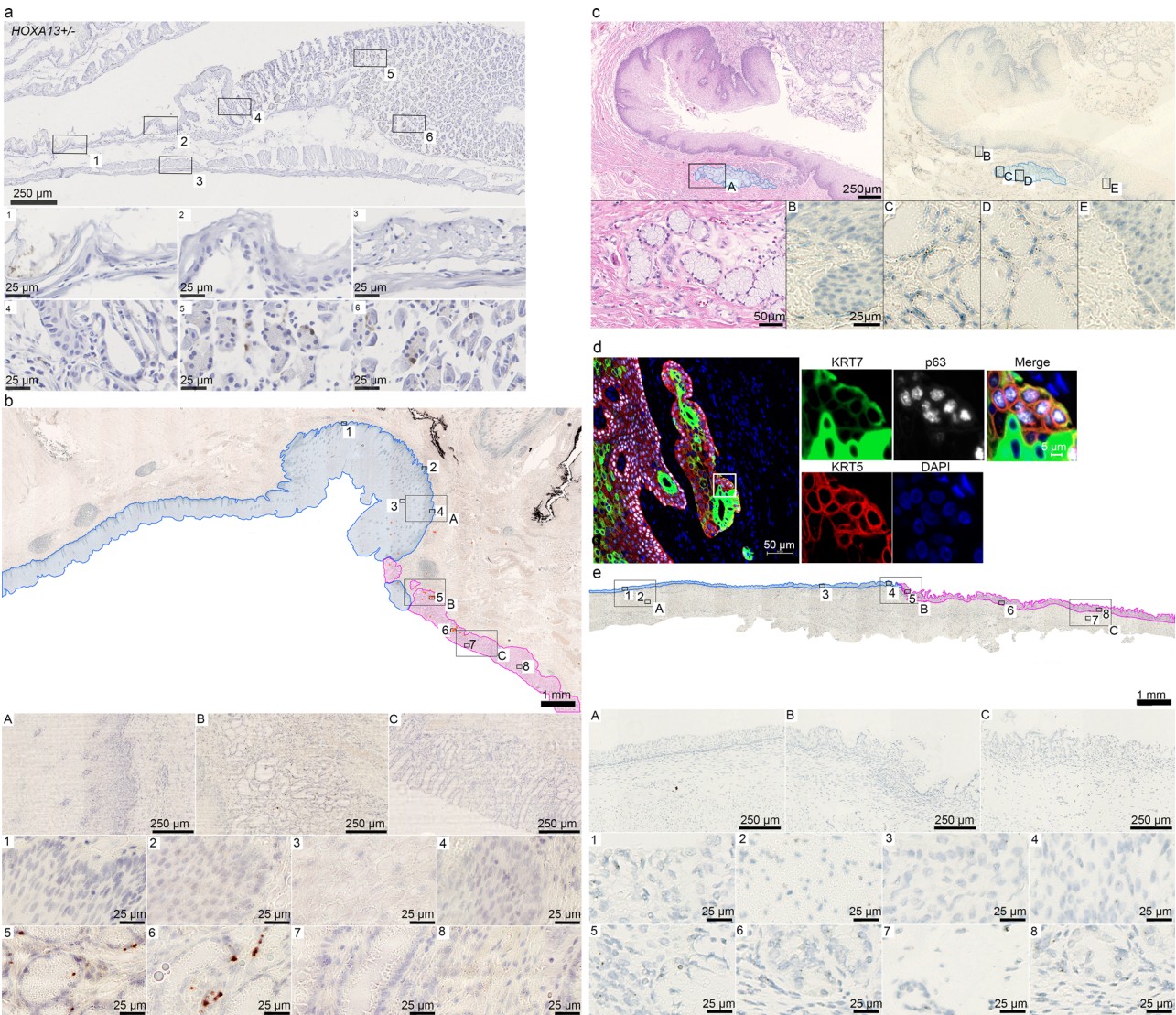

**Fig. 4 HOXA13 expression in the upper GI tract. a** Representative example of anti-GFP immunohistochemistry of a Hoxa13-GFP heterozygous mouse with gastroesophageal junctions (GEJ) (SQ; squamous epithelium, ST; stomach) ($n = 3$). Hoxa13 is expressed in single cells of the stomach starting from the GEJ (5–6) and absent in the esophagus and stroma (1–4). **b** HOXA13 expression as measured by RNA ISH in a representative example from $n = 3$ with similar results of an adult human GEJ with magnification panel of: A—esophagus, B—GEJ area, C—proximal stomach. Orange circles indicate the positive signal in the overview image. **c** HOXA13 expression as measured by RNA ISH in human esophageal submucosal gland (ESMG) with magnification panel of: A—H&E, C,D—ESMG, B, E—squamous esophagus, $n = 1$. **d** Keratin 7 (KRT7), keratin 5 (KRT5) and p63 triple positive cells are found in the ESMG, $n = 1$. **e** Overview of a representative example of a 17-week old fetus GEJ: A) Stratified esophageal epithelium of the distal esophagus (blue), B) GEJ area, C) gastric epithelium of the proximal stomach (pink), $n = 3$.

**HOXA13 and the chromosome 1 epidermal differentiation complex.** Further support for a role of *HOXA13* in the loss of the squamous phenotype and the appearance of caudal columnar phenotypes in the esophagus comes from experiments in which we investigated the effect of *HOXA13* directly on esophageal cell models. To this end, we used CRISPR-Cas9 technology to delete *HOXA13* from BAR-T, a primary monoclonal immortalized cell line derived from metaplastic tissue of a BE patient, with cells expressing both columnar and squamous markers[33]. Three separate *HOXA13* knock-out clones were selected to circumvent potential off target effects. Reversely, we provoked lentivirus-mediated *HOXA13* expression in EPC2-hTERT, an immortalized squamous esophageal cell line. For these latter experiments we used a mixed cell population of lentivirally transduced cells as to avoid clonal artifacts influencing results. Transcriptomes in these two models (Fig. 6a) were contrasted to their respective control lines.

There was substantial overlap in the gene sets significantly affected by losing *HOXA13* in BAR-T compared with those significantly affected by gaining *HOXA13* in EPC2-hTERT, taking into account the direction of regulation ($X^2$ test: $p = 4.74 \times 10^{-34}$) (see Supplementary Data 3). Investigation of this overlap across the two technically independently generated datasets limits the incidence of chance findings or single model system bias. Overlapping genes positively affected by *HOXA13* expression in esophageal cells are *IL7r, FAM196B, ADAMTS6, NRG1, LTBP1, JAG1, ELL2, SMAD7, C12ORF75, AXL, TIPARP, IKBIP, DUSP7,* and *GOLIM4*. Down-regulated by *HOXA13* expression are *SERPINB13, MYO5C, KLK7, ANXA9, TMPRSS4, TTC9, MATN2, TNFAIP2, RAB27B, HCAR2, C6ORF132, EXPH5, MAP3K5,* and *FUCA1*. IPA analysis of the results predicts an increase in "(malignant) cell transformation" ($z = 2.00$, $p = 5.81 \times 10^{-3}$) and a decrease in "inflammation of an organ" ($z = -2.59$, $p = 8.29 \times 10^{-3}$; gene function is described in

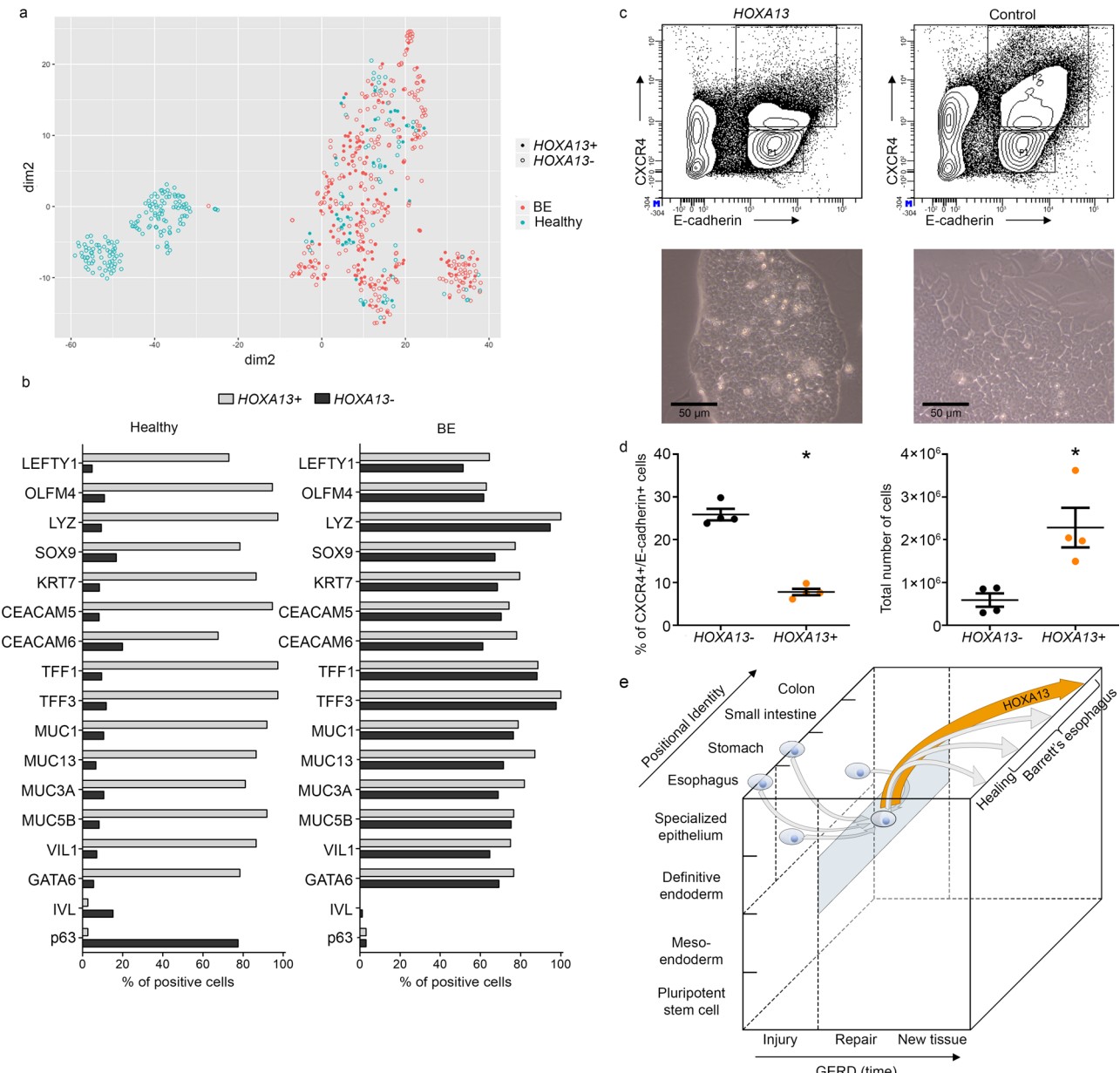

**Fig. 5 HOXA13 cellular expression modulates cell fate.** a *HOXA13*[+] cells of normal esophagus cluster together with BE cells in t-distributed Stochastic Neighbor Embedding (*T-SNE*) plot based on single cell RNA expression profiling[21]. **b** Analysis of single cell RNA seq data revealed that in contrast to *HOXA13*[−] cells, *HOXA13*[+] cells express submucosal gland markers, Barrett's esophagus (BE) markers and have decreased expression of squamous markers (p63, IVL) in healthy esophagus. This difference is not observed in BE. $n = 846$ of HOXA13[−] cells in healthy esophagus, $n = 37$ of *HOXA13*[+] in healthy esophagus, $n = 263$ HOXA13[−] cells in BE, $n = 132$ of *HOXA13*[+] cells in BE. **c** *HOXA13*-overexpressing definitive endoderm is relatively resistant to terminal differentiation. Mouse embryonic stem cells (mESC) cells with and without forced *HOXA13* expression were differentiated from pluripotent stem cells to definitive endoderm. The percentage of differentiated definitive endoderm cells, defined as CXCR4[+]/E-cadherin[+] cells, was analyzed by FACS analysis (upper panels). Lower panels (representative light microscopy images) show morphological differences in cultures of *HOXA13* overexpressing and wildtype mESCs upon differentiation to definitive endoderm, which induces a flattening of cell layers, with larger and irregular shaped cells. **d** Quantification of FACS analysis results indicates that the percentage of CXCR4[+]/E-cadherin[+] cells is decreased in *HOXA13*-overexpressing cell cultures under differentiation conditions ($p < 0.0001$). *HOXA13*-expressing cells expand faster during the differentiation process compared to control cells (total number of cells increased) ($p = 0.0135$). Mean ± SEM, ***$p < 0.001$, $n = 4$ independent experiments, t-test (two-tailed). **e** Model of cellular identity in BE development. The X-axis represents time (hypothetical units) following exposure to GERD-inducing agents. Y-axis shows differentiation during embryology and pathology. Z-axis indicates the positional identity of GI-tract tissues. Several theories exist regarding the cell of origin of BE: they may be fully differentiated esophageal or stomach cells, or less differentiated cells within these organs (depicted by the 4 cells on the Y–Z plane). Irrespective of its location or differentiation state, this cell or origin might lose its correct positional identity or maintain its aberrant positional identity and resembles a definitive endoderm like cell. This is visualized by the blue rectangle harboring the cell with the thicker blue contour. For the model of cellular identity in BE, our data suggest that *HOXA13* expressing clones in the GEJ, depicted in orange, may outcompete clones with another positional identity, providing an explanation for the distal phenotype observed in BE.

**Table 1 Fold changes and q-values for the mRNAs mentioned in the results section of the main text pertaining to cell culture models analyzed by RNA-Seq.**

| Gene name | Fold change | q-value (multiple testing corrected p value) |
|---|---|---|
| Forced *HOXA13* in mESC confers a relative competitive advantage in multipotent cell cultures through upregulation of *Nanog* signaling and downregulation of *Wnt* signaling | | |
| *Sox2* | 1.43 | 0.01 |
| *Nanog* | 1.47 | 0.00 |
| *Tbx3* | 3.53 | 0.00 |
| *Hesx-1* | 20.56 | 0.00 |
| *Dppa-1* | 64.95 | 0.00 |
| *Igf2* | 3.61 | 0.00 |
| *Wnt3* | 0.43 | 0.00 |
| *Wnt4* | 0.36 | 0.00 |
| *Wnt6* | 0.36 | 0.00 |
| *Wnt8a* | 0.48 | 0.01 |
| *Sp8* | 0.12 | 0.00 |
| *Lef1* | 0.34 | 0.00 |
| *Tbxt* | 0.41 | 0.00 |
| *Axin2* | 0.55 | 0.04 |
| *Fgf8* | 0.10 | 0.00 |
| *Cdx1* | 2.47 | 0.00 |
| *Grhl3* | 5.00 | 0.00 |
| *Vill* | 1.87 | 0.00 |
| Forced HOXA13 expression supports caudal epithelial functions and appears to promote proliferation in DE | | |
| *Sox17* | 12.55 | 0.00 |
| *Lgr5* | 5.39 | 0.00 |
| *Nanog* | 0.20 | 0.02 |
| *Ezr* | 2.20 | 0.00 |
| *Vill* | 2.58 | 0.048 |
| *Krt19* | 2.34 | 0.00 |
| *Krt20* | 2.84 | 0.01 |
| *Igsf8* | 4.67 | 0.00 |
| *Gcnt3* | 3.51 | 0.00 |
| *Fgfr2* | 2.81 | 0.00 |
| *Nek2* | 2.35 | 0.00 |
| HOXA13 downregulates the chromosome 1 epidermal differentiation complex, is pro-oncogenic, and conveys typical characteristics of the BE phenotype | | |
| *ANXA9* | 0.48 | 0.04 |
| *EVPL* | 0.61 | 0.03 |
| *SCEL* | 0.52 | 0.01 |
| *KLK7* | 0.42 | 0.01 |
| *EMP1* | 0.56 | 0.03 |
| *SERPINB13* | 0.38 | 0.00 |
| *DLL1* | 2.57 | 0.00 |
| *FURIN* | 1.49 | 0.03 |
| *JAG1* | 1.85 | 0.04 |

Supplementary Data 3) in cells expressing *HOXA13*. Intriguingly, *HOXA13* downregulates the epidermal differentiation complex (EDC); Fig. 6b, c). The EDC, located on chromosome 1q21.3, contains clustered multigene families of genes associated with cornified envelope formation in stratified squamous epithelia, such as the S100 and the small proline-rich region (SPRR) genes[34]. Among the overlapping downregulated genes in both cell models, *ANXA9* is also associated with differentiating keratinocytes[35], and *EVPL*, *SCEL*, and *KLK7* are cornified envelope genes[36–38]. *EMP1* and *SERPINB13* downregulation is associated with increased disease severity in gastric cancer (*EMP1*) and head and neck squamous cell carcinoma (SCC) (*SERPINB13*)[39,40]. See Table 1 for fold changes and *q*-values and Supplementary Data 3 for more differentially expressed molecules related to morphology. The

downregulation of a gene region known to be essential for maintaining a squamous phenotype provides mechanistic support to the notion that altered *HOXA13* expression is cardinal for provoking the BE phenotype.

These experiments also provide mechanistic support for the notion that *HOXA13* expression may offer an explanation as to why BE is prone to progression to EAC. *HOXA13* mediates down-regulation of the EDC and many EDC and cornified envelope genes are progressively down-regulated in the BE to EAC cascade[36,41]. In BE, EAC, and esophageal SCC, loss of heterozygosity (LOH) of the EDC is common[42–44]. Low EDC gene expression predicts chemotherapy non-response and LOH of the EDC is associated with reduced survival in curatively treated EAC patients[43,45]. In our experimental models, we observed *HOXA13*-mediated upregulation of genes associated with Notch signaling, specifically *DLL1*, *FURIN*, and *JAG1*. Notch signaling is associated with malignant transformation[46,47]. In IPA analysis "Non-melanoma solid tumor" ($z = 2.03$, $p = 9.28 \times 10^{-8}$) and "invasion of cells" ($z = 2.08$, $p = 2.47 \times 10^{-3}$) were shown to be activated by *HOXA13*, whereas *HOXA13* expression negatively influenced the "Apoptosis" ($z = -1.52$, $p = 2.23 \times 10^{-3}$) and "killing of cells" ($z = -2.03$, $p = 2.03 \times 10^{-3}$) categories. Many individual genes showed differential regulation in a pro-oncogenic direction (see Supplementary Data 3).

In conclusion, using *HOXA13* knock-out and overexpression in a Barrett's and a squamous cell line, we show that *HOXA13* downregulates the epithelial differentiation complex and other cornified envelope genes which normally function to maintain squamous epithelial morphology and act as tumor suppressor genes. Additionally, Notch signaling is overexpressed and many individual genes show differential regulation in a pro-oncogenic direction.

**HOXA13 supports columnar phenotype and provides pro-liferative advantage.** Having established the transcriptional effect of *HOXA13* on BAR-T and EPC2-hTERT cells, we next investigated the functional consequences of *HOXA13* in these cells. As was seen for mESCs, *HOXA13* expression significantly enhances the growth-rate of esophageal cells. For BAR-T cells, a proliferative advantage of *HOXA13* expression was seen in 2D cultures (Fig. 6d), while for EPC2-hTERT the positive effect of *HOXA13* expression on cell growth was more noticeable under 3D culture conditions (Fig. 6e). Moreover, *HOXA13* expression decreases the sensitivity of keratinocytes to bile/acid exposure (Fig. 6f), consistent with the notion that *HOXA13* confers cellular protection under GERD-like conditions.

To gain further insight into the role of *HOXA13* in cell morphology and organization, we made use of the fact that EPC2-hTERT cells can be differentiated in 3D spheroid cultures, and become organized in layers with a more flattened cytological aspect in the middle of spheroids and high expression of keratinization markers such as involucrin, similar to esophageal stratified epithelium (see example in Fig. 7a top panel for differentiated morphology)[48]. Upon overexpression of *HOXA13*, EPC2-hTERT spheroids increase in size while maintaining a less differentiated phenotype (undifferentiated morphology, Fig. 7a bottom panel). Quantification of these morphological states indicates that in control cultures, 80% of spheroids attain a stratified epithelial phenotype, while overexpression of HOXA13 reduces this number to 28.6% ($p < 0.05$, Fig. 7b left panel). This was further confirmed by staining for involucrin as a marker of keratinization, showing a decreased expression in spheroids derived from *HOXA13*-overexpressing EPC-hTERT cells ($p < 0.05$, Fig. 7b right panel). Thus, in primary immortalized esophageal cells *HOXA13* overexpression reduces keratinization.

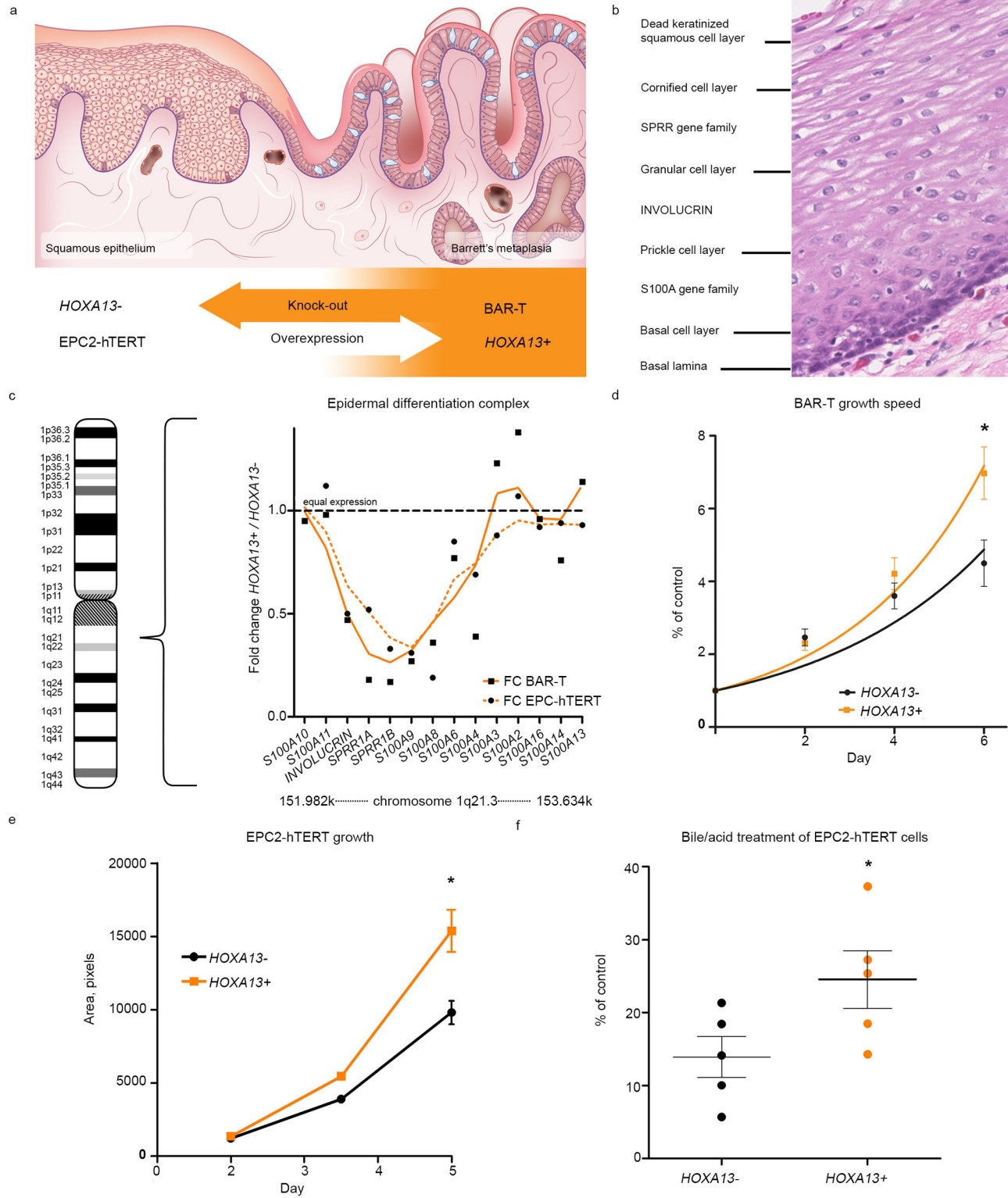

We further investigated the morphological role of HOXA13 in the BE-derived BAR-T cell line. In 2D cultures, an altered spatial distribution in growth pattern was observed, with cells growing more closely together in the absence of HOXA13 suggesting an effect on tissue morphology (Fig. 7c). The BAR-T cell model also allows testing the effect of *HOXA13* on columnar *versus* squamous differentiation in in vitro and in vivo settings. A 3D in vivo tissue reconstitution model was employed in which BAR-T cells were grafted in the lumina of devitalized and denuded rat

tracheas and implanted in NOD SCID mice. Under these conditions, parental non-transfected BAR-T cells produce both intestinal-type columnar epithelium and stratified squamous epithelium from the same clone. Hence this cell line has the potential to produce two types of morphological distinct epithelia[33,49] (Fig. 7e and Supplementary Fig. 9). Thus the epithelium in the model finds itself on a tipping point between both morphologies. This characteristic makes the in vivo tissue reconstitution model suited for studying the influence of

**Fig. 6 *HOXA13* counteracts squamous identity and increases growth of esophageal cells. a** Two models were constructed to investigate the function of *HOXA13* at the gastroesophageal junctions (GEJ). One model used EPC2-hTERT, a primary immortalized human squamous esophageal cell line, characterized by low *HOXA13* expression, in which *HOXA13* was transduced. The second model employed BAR-T, a primary immortalized human Barrett's esophagus (BE) cell line, characterized by high *HOXA13* expression, in which *HOXA13* was knocked out. **b** Hematoxylin and eosin (H&E) staining of the squamous esophagus of a patient without BE indicating the expected location of some of the products of the Ch1q21.3 epidermal differentiation complex along with other genes from the cornified envelope of the epidermis. **c** *HOXA13* leads to a downregulation of genes in the Ch1q21.3 epidermal differentiation complex in both model systems. A cubic spline fit of *HOXA13* mRNA regulation is shown, with the BAR-T control transduced cell line presented compared to its *HOXA13* knock-out counterpart, and *HOXA13* overexpressing EPC2-hTERT cells presented compared to their parental line. *FC* fold change. **d** *HOXA13* knock-out in a BE cell line reduces the growth of the cell pool, as measured by MTT assay. Mean ± SEM, \**p* < 0.05, exact *p* = 0.0204, two-tailed *t*-test, *n* = 9 independent experiments. **e** *HOXA13* overexpression in a EPC2-hTERT cell line increases its growth in 3D culture (area of spheroids, mean ± SEM, \**p* < 0.05, *p* = 0.0174, two-way ANOVA with Sidak's multiple comparisons test). **f** EPC2-hTERT cells with *HOXA13* overexpression are less sensitive to bile/acid exposure (*p* = 0.0343). MTT data presented as % of corresponding vehicle-treated controls. Mean ± SEM, \**p* < 0.05, *t*-test (two-tailed).

modulators of morphology, i.e. to show if the modulator favours intestinal-type columnar epithelium or stratified squamous epithelium. Studying the effect of *HOXA13* knock-out, two important observations were made. Firstly, *HOXA13* knock-out decreases the length of columnar-like epithelium which contains PAS positive cells and is negative for involucrin (Fig. 7e, f). Thus, loss of *HOXA13* counteracts the proliferation of the intestinal-type columnar epithelium while the stratified squamous epithelial proliferation remains present. Secondly, *HOXA13* knock-out impairs epithelial proliferation in general, as inferred from the thickness of the epithelial layer (Fig. 7d; Supplementary Fig. 9b). In vitro 2D organotypic ALI cultures of these cell lines confirm the in vivo findings, with *HOXA13* knockout reprogramming the BAR-T epithelial cells towards a squamous keratinized differentiated epithelium (Fig. 7f). In conclusion, *HOXA13* supports intestinal-type columnar epithelial differentiation and proliferation of the Barrett's epithelium confirming the notion that *HOXA13* expression can mediate both a competitive advantage as well as a predisposition to the formation of columnar phenotypes.

## Discussion

In this study, we characterized *HOX* gene expression and localization in mice and men, demonstrating a collinearity of these genes along the GI tract. Following analysis of one of these *HOX* genes, *HOXA13*, we observed single *HOXA13* + cells in the upper GI tract, which present exceptions to the *HOX* gene collinearity theory. Specifically, in the normal physiology of the esophagus and proximal stomach, non-squamous structures such as the epithelium at the GEJ, glandular cells of ESMGs and glands of stomach contain single cells expressing *HOXA13*. The fact that these cells have not been described before may be a reflection of the fact that homogenization of tissues for qPCR masks this fraction, and that single cell analysis of the GI tract for this gene has not been performed before. We observe that GI pathology with distal phenotypes like intestinal metaplasia of the esophagus and stomach are characterized by an expansion of *HOXA13*-positive cells, while conversely, a relatively low expression of *HOXA13* is found in the phenotypically rostral Paneth cell metaplasia and pyloric metaplasia of the colon, compared to the surrounding physiological tissue.

It is clear that in normal physiology, *HOXA13* contributes to the distal phenotype of the caudal GI tract, begging the question as to the role and origin of the *HOXA13*-expressing compartment now observed in the upper GI tract. We demonstrate that esophageal *HOXA13*-positive cells express columnar and BE markers and show gene expression patterns overlapping with BE-derived cells. Functionally, *HOXA13* provides cells with several properties required for development of a BE segment. *HOXA13* maintains cells in a stem-like progenitor state, while conferring a proliferative advantage, promoting cellular migration[50] and resistance to bile and acid exposure. Furthermore, in cells that are

lineage committed, *HOXA13* supports a phenotypically columnar phenotype, most likely partly driven by downregulation of the chromosome 1 epidermal differentiation complex. Thus, our data are consistent with the hypothesis that BE arises as a consequence of the expansion of resident *HOXA13*-positive cells under abrasive environments such as GERD. Several potential theories have been proposed as to the origin of BE: transdifferentiation of basal cells in the squamous epithelium, extension of a special population of cells from the GEJ, repopulation of the esophagus after injury with cells derived from progenitors ESMGs or ducts, resident embryonic stem cells or circulating bone marrow cells[51]. These potential sources of esophageal columnar epithelium are not mutually exclusive, and BE may have more than one precursor cell or location. Our study supports the previously proposed hypothesis that BE may originate from ESMGs and the GEJ as *HOXA13* is expressed in OLMF4⁺, LEFTY1⁺ cells of ESMGs, recently suggested as a cell of BE origin[21]. The fact that in addition to the GEJ, rare *HOXA13*⁺ cells are found in the human esophagus and stomach, is consistent with the observation that after esophagogastrostomy BE can reoccur in patients, indicating that the involvement of the GEJ is not an absolute prerequisite for the development of BE[52]. Furthermore, our data show that *HOXA13* is already present at stem cell level, supportive of the notion that BE may arise from a cell with stem-cell like characteristics. While *HOXA13* expression overlaps greatly with *KRT7*, a columnar cytokeratin seen in Barrett's, we did not observe direct transcriptional overlap with the previously described *KRT7⁺KRT14/5⁺TP63⁺* cell of BE origin. However, *KRT7⁺KR5⁺TP63⁺* cells gave rise to BE-like epithelium only upon ectopic expression of *CDX2*[22]. Lineage-tracing studies are needed to further confirm whether one or more types of cells of origin might exist for BE. While we focused on *HOXA13* here, it is conceivable that other *HOX* paralogues are involved in BE pathophysiology, in particular caudal genes such as *HOXA10, 11, B13*, and *C10* are interesting candidates for further investigation, in particular as disruption of collinearity was reported for cluster B in BE[9] and in duodenum of murine embryos[12].

BE is considered as the precursor lesion for EAC, a dangerous form of cancer of which the incidence has substantially increased in recent decades. Increased insight into the pathogenesis of BE may aid development of prevention and treatment strategies for EAC. *HOXA13* is involved in ESCC[53] and other types of cancer[54–57]. Here we show that expression of *HOXA13* also increases in EAC and colorectal cancer, provides proliferative advantage to the cells and activates cancer-related gene transcription like Notch signaling. Hence, we speculate that *HOXA13* may play a role in BE progression towards EAC.

*In toto*, the present study identifies a importance of regional patterning by *HOX* genes in the gut epithelium. In Barrett's esophagus, gastric IM, and heterotopia of the upper GI-tract, a colon-like *HOX* gene expression is present, especially

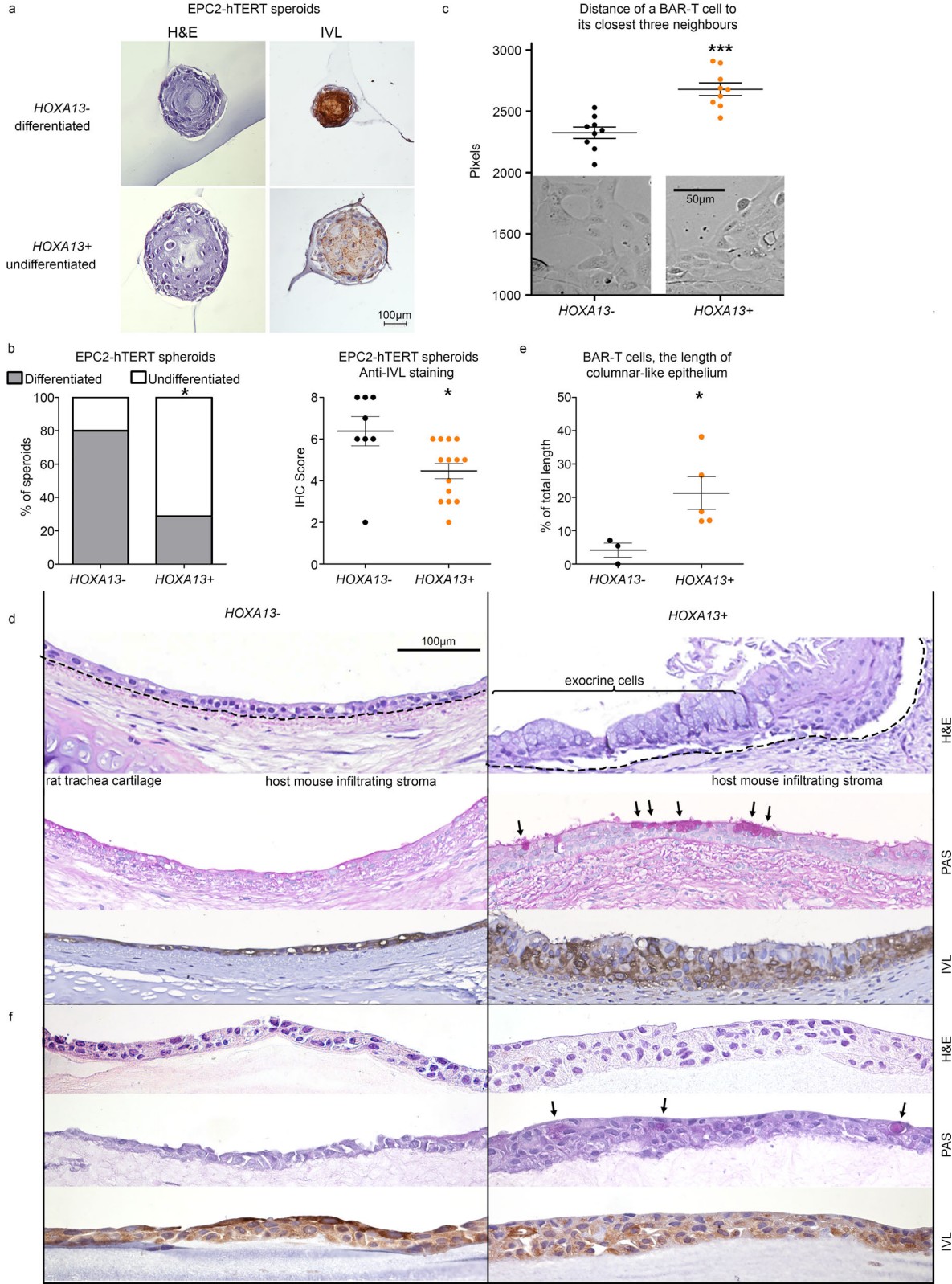

characterized by *HOXA13* upregulation. Single cells expressing the generally thought to be distally-restricted *HOXA13* gene are present in the physiological upper GI tract, in particular the GEJ, where it supports a columnar phenotype and may confer a relative competitive advantage. Thus, *HOXA13* mediates BE phenotype and proliferative potential and hence appears a rational target for strategies aimed at counteracting EAC development.

## Methods

**Collection of human material.** All human tissues used in this study were obtained at the Erasmus University Medical Center, department of Gastroenterology & Hepatology. The use of these samples was approved by the Erasmus MC medical ethical committee (MEC-2015-208, MEC-2015-209, MEC-2015-199, MEC-2010-093; tissues were handled according to the FEDERA code of conduct and informed consent was obtained for all participants[58]. "The study was designed and carried out according to the ethical principles for medical research involving human

**Fig. 7 HOXA13 supports intestinal-type columnar epithelial differentiation.** *HOXA13* overexpression impairs squamous differentiation of EPC2-hTERT spheroids as seen from the representative pictures (**a**) and quantitative assessment of morphologies based on hematoxylin and eosin (H&E) or anti-involucrin (IVL) immunohistochemistry (IHC) ($p = 0.0086$). (**b**). Median with interquartile range, *$p < 0.05$, Mann–Whitney test (two-tailed). **c** *HOXA13* knockdown (KO) affects spatial distribution of BAR-T cells. Mean ± SEM, ***$p < 0.001$, $p = 0.0001$, $t$-test (two-tailed). **e** The length of columnar and mixed BAR-T epithelium decreases upon *HOXA13* KO in the rat trachea in vivo tissue reconstitution model. Mean ± SEM, *$p < 0.05$, exact $p = 0.0439$. **d** Representative examples of H&E, PAS staining, anti-IVL IHC of BAR-T epithelium from the rat trachea in vivo tissue reconstitution model. H&E staining shows more layers of cells in animals transplanted with *HOXA13*+ wild type cells. Periodic acid–Schiff (PAS) stains polysaccharide molecules, and positivity is indicative of goblet-like cells. The arrows point to PAS positivity, which is present in the right panel but not in the left panel, where the BAR-T *HOXA13*− cells are shown. IVL staining is strong in morphologically squamous cells in the left hand panel and weaker in the *HOXA13*+ epithelium ($n = 3$ for *HOXA*− and $n = 5$ for *HOXA13*+). **f** *HOXA13*− and *HOXA13*+ representative pictures of H&E staining, PAS staining and IVL IHC of the BAR-T organotypic cell culture system indicate that *HOXA13* KO reprograms the columnar epithelial phenotype towards squamous keratinized epithelium ($n = 1$ independent experiments).

subjects from the World Medical Association Declaration of Helsinki". Biopsy specimens to investigate *HOX* collinearity were obtained by double balloon enteroscopy. Nine biopsy specimens were obtained from each patient ($n = 3$) at different locations along the GI-tract. Sequentially these locations were: esophagus, stomach, duodenum, jejunum, proximal ileum, distal ileum, ascending colon, descending colon and sigmoid/rectum (Supplementary Fig. 2). Included patients had unexplained symptoms, mostly anemia, while inflammatory bowel disease patients were excluded. All biopsies for RNA isolation were stored in RNA-later at −80 °C. Squamous esophageal biopsies ($n = 13$) originated 5 cm above the squamocolumnar junction (SCJ). Barrett's (BE) biopsies ($n = 13$) originated caudal of the SCJ and cranial of the gastric folds (all patients were on PPI therapy), stomach biopsies ($n = 12$) were from the corpus. All three types of biopsy specimens were derived from the same patients, (one stomach biopsy specimen was not obtained due to patient agitation during the gastroscopy). The squamous esophageal, BE, and stomach biopsies were taken in a paired fashion from 13 patients. Where the number of samples is indicated below, this indicates the number of individual patients. Forceps biopsy specimens of EACs ($n = 12$) were obtained. Pathological examination of simultaneously taken forceps biopsies around the study specimens had to be positive for EAC. Gastric inlet patch were sampled from proximal esophagus. To determine the proximal colonic HOXA13 border, biopsies were taken from the cecum at the appendix base, the ileocecal valve, 5 cm distal to the ileocecal valve, and from the transverse colon in each patient ($n = 5$).

**Collection of archival pathology specimens.** FFPE material was collected from gastric IM (from the antrum, angulus, and corpus, i.e., not from the cardia; $n = 12$), the gastric inlet patch ($n = 5$), CLE (from the proximal esophagus; $n = 14$), and Meckel's diverticula ($n = 14$). For RNAscope RNA-ISH, one FFPE specimen of each of these origins was used. Depending on the extent of metaplasia, remainder of tissues was used whole, macroscopically separated, or processed with the Photo-Activated Localization Microscopy with the Laser Caption Microdissection (PALM LCM) for mRNA isolation and subsequent qPCR. Nuclease-free membrane slides treated with UV light at 254 nm for 30 min were used to mount 10 μm sections, dried overnight at 56 °C, deparaffinized, stained with hematoxylin and eosin, and dehydrated. AdhesiveCap microtubes obtained from Zeiss (Oberkochen, Germany) were used to collect the tissue of interest after cutting and pulsing of the PALM LCM. Additionally, FFPE materials or fresh pinch biopsies were collected from the squamous esophagus of a patient without BE, the squamous esophagus of a BE patient, BE, EAC, stomach (the corpus), and the ileum. Colon was used as a positive control. FFPE materials that were collected only for RNA-ISH were pyloric metaplasia (from the colon; $n = 5$), Paneth cell metaplasia (from the colon; $n = 5$), fetal GEJ tissue ($n = 2$ of 17 weeks, and $n = 1$ of 20 weeks; this material originated from spontaneous abortions), and adult GEJ tissue consisting out of continuous strips of tissue containing squamous esophageal epithelium, GEJ, and oxyntic stomach epithelium ($n = 3$). Two strips came from surgical specimens without evidence of BE, with a neuroendocrine tumor and decompensated achalasia (male of 71 and female of 56 years old). The third patient had surgery to remove an EAC (male of 63 years old). All tissues were obtained from the gastroenterology and pathology departments of the Erasmus MC according to the FEDERA code of conduct[58]. The use of archival pathology specimens was authorised by an institutional review board (METC –Erasmus MC) on the basis that the material were obtained as left over specimens from routine diagnostics and thus not subject to the WMO (Law Medical Research from the Netherlands) and informed consent was waived by the institutional review board.

**Animal studies.** For the *Hoxa13* mRNA expression analysis throughout the murine gastrointestinal (GI) tract, four *C57BL/6J* wildtype mice were used between three and five months of age. The GI-tract was divided into 1: esophagus; 2: stomach; 3: duodenum; 4: jejunum; 5: proximal ileum; 6: distal ileum; 7: cecum; 8: proximal colon; 9: distal colon, of which sections were opened and rinsed in PBS followed by storage in RNAlater at −80 °C (Supplementary Fig. 1b). For determining which cells express *Hoxa13* in the GI-tract, tissues from a *C57BL/6J-Hoxa13-GFP* heterozygous mutant mouse model were employed, in which GFP

expression is driven by the endogenous mouse *Hoxa13* promotor through the creation of a fusion protein[59]. Mice were generally kept with 12:12 h light–dark, the animal room temperature is between 20 and 24oC and the relative humidity is 55 ± 10%. These tissues were taken out and embedded in O.C.T. Compound bought from Qiagen Inc. (Hilden, Germany) and frozen at −80 °C. Cryosections were made which were mounted in fluoroshield mounting medium with DAPI obtained from Abcam (Cambridge, UK). Subsequently, the GEJ and the distal GI-tract were analyzed directly for GFP expression using the Zeiss confocal laser scan microscope LSM 510. Additionally, immunohistochemistry staining was performed with anti-GFP antibody (#AB3080, Bio-Connect BV) (see below). These murine experiments were approved by the Ethical Committee for Animal Experiments of the Erasmus MC and were performed according to the guidelines of the same institution.

**Immunohistochemistry.** For immunohistochemistry, slides were blocked in 10% of normal goat serum, antigens were retrieved by boiling samples in citrate buffer (pH6), and samples were incubated overnight at 4 °C with primary antibody. Dilutions and manufactures of primary antibodies are presented in the Supplementary Table 4. After incubation with HRP-conjugated secondary antibody (Dako EnVision+System-HRR labeled Polymer Anti Mouse, Dako) endogenous peroxidase was blocked in 3% $H_2O_2$ and antibody binding was visualized by DAB staining. IHC analysis for HOXA13 was tried using antibodies ab106503 and ab26084, however these failed to show specificity and have since been discontinued by the companies offering them. H&E staining was performed by[60]. For H&E stainings de-parafinized 4 μM slides were incubated during 3 min in hematoxylin solution, followed by tap water washes and 15 s of incubation with eosin. For PAS staining, de-parafinized slides were incubated with 0.5% Periodic Acid solution for 10 min, followed by two ddH2O washes and incubation in Schiff's reagent (Sigma Aldrich) for 15 min and hematoxylin for 3 mins.

**Multiplex immunofluorescent staining.** Triplex staining for keratin 5, keratin7 and p63 was done by automated multiplex IF using the Ventana Benchmark Discovery (Ventana Medical Systems Inc.). In brief, following deparaffinization and heat-induced antigen retrieval with CC1 (#950-500, Ventana) for 64 min at 97 °C, the tissue samples were incubated firstly with Keratin 5 antibodies for 32 min at 37 °C followed by detection with Ultramap anti-rabbit HRP (#760–4315, Ventana) for 12 min followed by visualization with Red610 for 8 min (#760-245, Ventana). Antibody denaturing was performed using CC2 (#950-123, Ventana) for 20 min at 100 °C. Secondly, Keratin 7 antibodies were incubated for 32 min at 37 °C followed by detection with Ultramap anti-rabbit HRP (#760–4315, Ventana) followed by visualization with FAM (#760-243, Ventana) for 4 min. Antibody denaturing was performed using CC2 (#950-123, Ventana) for 8 min at 100 °C. Thirdly, P63 antibodies were incubated for 32 min at 37 °C followed by detection with Ultramap anti-mouse HRP (#760–4313, Ventana) for 12 min followed by visualization with Cy5 for 12 min (#760-238, Ventana). Slides were incubated in PBS with DAPI for 15 min and covered with anti-fading medium (DAKO, S3023).

**RNA isolation.** RNA was isolated using the NucleoSpin RNA isolation kit (Macherey Nagel, Düren, Germany). Biopsies and animal tissues were homogenized by the TissueRuptor obtained from Qiagen Inc. RNA concentrations were measured using a Nanodrop spectrophotometer and samples were stored in RNA storage solution (Sodium Citrate pH 6.4), bought from Ambion (Foster City, USA) and kept at −80 °C. RNA integrity was checked with 1% agarose gel-electrophoresis. FFPE material was deparaffinized with xylene and ethanol, lysed, digested with proteinase K, and RNA was isolated with the High Pure FFPET RNA isolation kit obtained from Roche (Basel, Switzerland). RNA isolation from de-differentiated KH2 mouse embryonic stem cells (mESCs) was done using a pico-pure RNA isolation kit (Thermo-Fisher Scientific, Waltham, USA). After RNA isolation all samples for RNA-Sequencing were tested on the Agilent 2100 Bioanalyzer to determine RNA integrity and quantity.

**cDNA and qPCR.** cDNA was made from 1 μg RNA using Primescript RT Master Mix according to manufacturer's instructions (Takara, Otsu, Japan), for 15 min at 37 °C and 5 s at 95 °C, and stored at −20 °C. qPCR was performed for 40 cycles in the iQ5 Real-Time PCR detection system that was obtained from BioRad Laboratories (Veenendaal, The Netherlands). For each reaction 10 μl cDNA template, 12.5 μl SYBR GreenER purchased from Invitrogen (Carlsbad, CA), and 2.5 μl 10 pM/μl primer were used. Reactions were performed in duplicate. Primers used are shown in Supplementary Table 2 and were ordered at Sigma-Aldrich (Darmstadt, Germany). qPCR data were analyzed with Microsoft Excel using the ΔΔCt method. Reference genes used for PCRs on human materials were *RP2*, *β-ACTIN*, and *GAPDH*. Reference genes used for PCRs on mice materials were *Eef2*, *Rpl37*, and *Leng8*. Differences in expression were analyzed with a two sided Student's t-test using Prism 5.01, obtained from GraphPad Software (San Diego, USA). Values from individual samples were excluded if they deviated more than 2 SD from the mean. Correlations between *HOTTIP* expression in the squamous esophagus and BE, and correlations between *HOTTIP* and *HOXA13* expression levels in the squamous esophagus and BE were tested using nonparametric Spearman correlations. This is depicted in graphs by connecting lines between datapoints, also indicating the paired nature of the specimens, i.e. they are derived from the same patient, used for this analysis.

**In situ hybridization by RNAscope.** RNAscope was performed according to the instructions of the manufacturer of the probes and the reagent kit (VS Reagent Kit 320600; Advanced Cell Diagnostics), on proteinase K (0.1%, 10 min at 37 °C) treated paraffin sections (5 μm). Subsequently, slides were hybridized with the RNA probe from RNAscopeVS Hs-*HOXA13*, (art. #ACDA 400226), or the control probe also from RNAscopeVS Hs-*PPIB* (art. #ACDA 313901)[61]. *PPIB* (peptidylprolyl isomerase B) is a ubiquitously expressed gene. The RNAscope probe Hybridization in situ Multiplex was bought from Advanced Cell Diagnostics (Newark, USA). Pyloric metaplasia and Paneth cell metaplasia of the colon were quantified using FIJI, for which a macro was made (Supplementary method 1)[62]. For illustrations of RNA-Scope slides in the paper, background grey signal reduction was performed using Photoshop.

**Analysis of GSE datasets.** Expression profiles from clonogenic human gastrointestinal stem cell cultures were obtained from Gene Expression Omnibus datasets GSE57584[15] and GSE65013[18]. In silico analyses were performed using the NCBI Gene Expression Omnibus (GEO) database. Analyses in the GEO database were performed by using the GEO2R tool (www.ncbi.nlm.nih.gov/geo/geo2r/), R 3.2.3., Biobase 2.30.0, GEOquery 2.40.0, limma 3.26.8[63]. The results were represented as a $^2$log-fold change ($^2$log-FC). In Microsoft Excel, this $^2$log-FC was converted to fold change (FC). For each $^2$log-FC an empirical Bayes moderated t-statistic was calculated. *p*-values were corrected for multiple testing using the Benjamini & Hochberg false discovery rate method.

**Analysis of single cell RNA seq datasets.** BE and ESMG Single Cell Experiment Matrix from supplementary Data files 6 ll three Experiment Matrixes have been mapped to hg38 standard human genome ('TxDb.Hsapiens.UCSC.hg38.known-Gene' R-package), normalized as Reads Per Kilobase per Million mapped reads (RPKM). Genes expressed in less than 0.5% cells were filtered out. Low-quality cells were excluded based on: (1) the number of expressed genes—for 10x Single-Cell sequence data, cells expressing less than 400 or more than 7000 genes, for smartSeq data cells expressing less than 1000 and more than 7000 were removed. Different numbers were chosen due to the different sequencing depth. (2) Boxplot representation of all cells—outliers, i.e. cells mapping higher or lower than 1.5x the first or third quartile - were removed. (3) Based on % of reads - cells were removed if there were more than 20% of reads mapping to mitochondrial or ribosomal genes. *HOXA13*-related genes query: *HOXA13*-positive cells from normal esophagus were selected with R. Genes that were expressed in at least 70% of these *HOXA13*-positive cells (20445) were analyzed for their expression in *HOXA13*-negative cells of normal esophagus as well as *HOXA13* negative and positive cells in BE tissue. For *T-SNE* plot, 638 cells were included (388 cells from Barrett's tissue, 250 cells from normal esophagus) and plotted based on their location of origin (colour) as well as *HOXA13* expression (open vs closed symbols).

**Cell culture.** All cells were cultured with penicillin (100 μ/ml) and streptomycin (100 μ/ml) and were regularly STR-verified and checked for mycoplasma by handing in samples prepared according to instructions at GATC Biotech (Konstanz, Germany). Primary human esophageal epithelial cells transformed with hTERT (EPC2-hTERT) (gift of K.K. Krishnadath)[64], were cultured with Keratinocyte SFM medium, supplemented with bovine pituitary extract at 50 μg/ml and EGF at 1 ng/ml (Thermo-Fisher Scientific). HET1A, the primary immortalized human squamous esophageal cell line Het-1A was a gift of J.W.P.M. van Baal (University Utrecht, The Netherlands). These cells were grown in EPM2 medium obtained from AthenaES, (Baltimore, Maryland, USA). The primary immortalized human BE cell line (BAR-T) was a gift of dr. J.W.P.M. van Baal who had, in turn, received them from dr. R.F. Souza (University of Texas Southwestern Medical Center, USA). These cells were grown in supplemented keratinocyte basal medium (KBM2), bought from Lonza (Basel, Switzerland), according to the method of

Jaiswal et al.[65]. KH2 mESCs were a gift of J. Gribnau and maintained in DMEM with 10% FCS, Non-Essential Amino Acids, sodium pyruvate, LIF, and β-mercapto-ethanol (embryonic stem cell medium; Supplementary Table 3). Dishes were coated with attachment factor protein solution (Thermo-Fisher Scientific). Irradiated mouse embryonic fibroblasts (3T3-Swiss albino cells (gift of J.W.P.M. van Baal), cultured in DMEM with 10% FCS, were used as feeder cells. HEK293T cells were cultured in DMEM with 10% FCS.

**Generation of EPC2-hTERT *HOXA13* overexpression model.** The human *HOXA13* gene including its single intron was amplified using Q5 polymerase from gDNA using primers (AgeI HoxA13 F; GGTGGTACCGGTGCCACCATGACA GCCTCCGTGCTCCT, and XbaI HoxA13 R; ACCACCTCTAGATTAACT AGTGGTTTTCAGTT) and cloned into pEN_TmiRc3 using AgeI and XbaI restriction sites, a gift from Iain Fraser (Addgene (Cambridge, USA) #25748)[66]. Subsequently, the *HOXA13* insert was transferred into pSLIK-Venus, using a Gateway reaction[67]. pSLIK-Venus was a gift from Iain Fraser (Addgene #25734)[66]. A similar plasmid but without the *HOXA13* insert served as control. Both plasmids were sequenced by LGC Genomics (Teddington, UK). Next, plasmids were packaged into lentiviral particles following transfection in HEK293T cells with third generation packaging plasmids. The supernatant was collected and ultracentrifuged. EPC2-hTERT cells were transduced with the virus and Fluorescence-Activated Cell Sorted (FACS) for YFP (pSLIK-Venus) positive cells on the BD FACSCantoTM II that was bought from BD Biosciences (San Jose, USA). These cells were grown and analyzed as a cell pool. *HOXA13* was induced by the addition of 1,25 μg/ml doxycycline to the culture medium. Overexpression was determined by qPCR according to scientific standards[50].

**Generation of KH2 embryonic stem cells *HOXA13* overexpression model.** The human *HOXA13* gene including its single intron was amplified using Q5 polymerase from gDNA using primers with an added N-terminal FLAG-tag sequence (GACTACAAAGACGATGACGACAAG) and Kozak sequence (GCCGCCACC; Supplementary Table 3). Next, this PCR product was ligated into EcoRI digested pgk-ATG-frt (Addgene #20734) using Gibson Cloning (New England BioLabs Inc., Ipswich, USA). pgk-ATG-frt was a gift from Rudolf Jaenisch[68]. KH2 mESCs were passaged the day before the electroporation and four hours before electroporation medium was replaced. Approximately 1.5 10$^7$ KH2 cells were electroporated with 50 μg of pgk-ATG-frt-*HOXA13* and 25 μg of pCAGGS-FLPe-puro (Addgene #20733)[69]. Cells were electroporated in 4 mm cuvettes, with two consecutive pulses (400 V/250 μF) using a Gene PulserXcell (Bio-Rad Laboratories). The next day 140 μg/ml Hygromycin B (Thermo-Fisher Scientific) was added for the selection of correctly targeted colonies. DNA from resistant colonies was isolated with the Kleargene XL blood DNA extraction kit (LGC, Teddington, UK) and analyzed by Q5 PCR using the following primers: PGK-F1 or PGK-F2 and T1E2-HygroR6 and T1E2-HygroR7 (Supplementary Table 3). Correctly-targeted clones were checked for proper *HOXA13* induction by the addition of 1.25 μg/ml doxycycline to the culture medium for 3 days. Three *HOXA13* overexpression versus three control biological replicates were selected and used for experiments.

**Differentiation of KH2 mouse embryonic stem cells.** An optimized version of the Ogaki protocol was used[70]. Cells were plated on 50% confluent pre-cultured M15 cells, a mesoderm-derived feeder cell line[71] (gift of N. Hastie, University of Edinburgh, UK). Cells grew six days in differentiation medium consisting of ESC medium without LIF, with the addition of Activin-A, basic Fibroblast Growth Factor, CHIR, and Noggin (Supplementary Table 3). *HOXA13*-expression was induced on day four using doxycycline at 1.25 μg/ml. On day six, cells were analyzed by FACS by double staining with 0.8 μg PE Rat Anti-Mouse CD184 (CXCR4) and 2.0 μg Anti-CD324 Alexa Fluor® 488 (E-Cadherin) at 4 °C for 45 min (Supplementary Table 3). The cells were analysed with a BD FACSCantoTM II (BD Biosciences, USA). Data were analyzed with BD FACSDiva v8.0.1 software, which was obtained from BD Biosciences, and processed using Microsoft Excel. Example of analysis is provided in the Supplementary Fig. 11. Double-positive cells were sorted and cultured for another day with doxycycline at 1.25 μg/ml before harvesting and RNA isolation took place, using the picopure RNA isolation kit (Thermo-Fisher Scientific).

**Generation of the BAR-T *HOXA13* knock-out model.** Functional *HOXA13* was removed from BAR-T cells using CRISPR/Cas9-mediated gene editing. A *HOXA13* sgRNA targeting exon 1 was cloned into pTLCV2, by ligating two annealed oligonucleotides, *i.e.* Guide1sgRNA F and R (Supplementary Table 4). TLCV2 was a gift from Adam Karpf (Addgene #87360)[72]. Following sequence verification, the pTLCV2-*HOXA13*sgRNA plasmid was packaged into lentiviral particles by cotransfection into HEK293T cells with pSPAX2 and pMD2.G, gifts from Didier Trono (Addgene #12260 and #12259). The supernatant was harvested and ultracentrifuged after which BAR-T cells were transduced. Mixed populations of transduced cells were plated at very low confluence, single cell clones could subsequently be isolated using glass cloning cylinders and low melting point agarose from Sigma-Aldrich, followed by DNA isolation using the Kleargene kit, followed by sequence verification with primers TIL*HOXA13*R3 and Pre-*HOXA13*-FW2 flanking the sgRNA-site (Supplementary Table 4). Three cell lines in which both

alleles were affected by unique out-of-frame deletions were selected along with three control cell lines.

**RNA-sequencing**. The EPC2-hTERT samples ($n = 8$) were treated with the Tru-Seq Stranded mRNA Library Prep Kit. Sequencing took place according to the Illumina TruSeq v3 protocol on an Illumina HiSeq2500 sequencer. Sample preparation and sequencing was performed at the Erasmus MC. Reads of 50 base-pairs were generated and mapped against reference genome hg19 with Tophat (version 2.0.10). Expression was quantified using HTseq-count (0.6.1). Stranded libraries of the BAR-T ($n = 6$), and both non-differentiated and differentiated KH2 mESCs ($n = 6$ each) were prepared with the NEBNext RNA Ultra sample prep kit. Sequencing took place according to the Illumina NestSeq 500 protocol on an Illumina HiSeq2500 sequencer. Sample preparation and sequencing was performed at GenomeScan in Leiden, The Netherlands. Reads of 75 base-pairs were generated, mapped against reference genome hg19 or mm9 with Tophat (version 2.1.0), and quantified using HTSeq (version 0.6.1p1). Data were processed using R. version 3.2.5[73], in combination with the module DeSeq2[74]. Generated FCs and p-values adjusted for multiple testing, i.e. $q$-values, were analyzed using Ingenuity Pathway Analysis (IPA) version 42012434, obtained from Qiagen Inc. (Hilden, Germany)[75]. We limited the number of genes analyzed to a maximum of 1000 by eliminating genes with a (relatively) low fold change if differentially expressed genes number was above 1000. The dataset cut-offs used were always a $q$ value of 0.05, the fold change cut-off was set at: nondifferentiated KH2-mESCs, FC 2, 888 genes; differentiation of KH2-mESCs, FC 5, 924 genes; differentiated KH2-mESCs, FC not restricted, 665 genes; BAR-T, FC not restricted, 146 genes; EPC2-hTERT, FC 1.3, 990 genes. Activity scores are known in IPA as "$z$-scores" which represents the number of standard deviations from the mean of a normal distribution. For analysis and visualization of gene expression in the epidermal differentiation complex the raw counts from both models normalized to total reads were used. Genes for which one of both cell models had less than ten reads in the control or experimental samples were excluded. Overlap in multiple testing corrected differentially expressed genes in the BAR-T and EPC2-hTERT datasets was calculated as follows; the proportion of overexpressed genes in the EPC2-hTERT dataset was determined. Half of the differentially expressed genes in the BAR-T dataset would be expected to be regulated in the same direction if regulation would be random. This expected overlap if regulation was random, and the observed overlap, were used as input for an $X^2$ test. Information included in Supplementary Data 2 and 3 in the "known function" and "Detailed description" columns was obtained through non-systematic review and should not be considered as an exhaustive overview of the literature. Association of expression of molecules in the distal GI-tract with their regulation by *HOXA13* expression was reviewed using the human proteome atlas and depicted in Supplementary Data 2[32].

**Acid and bile exposure**. For assessment of *HOXA13* mRNA expression upon acid/bile exposure, EPC2-hTERT and HET-1A cells were treated for 30 min with cell culture medium adjusted to a pH of 7.0 or 4.0 using HCl. Cells were subsequently washed using PBS and given standard medium. Acid experiments were performed four times in duplo. Cells were separately exposed to medium with a bile acid mixture in concentrations of 0, 200, (and 400 for EPC2-hTERT) µmole/L for 30 min at a pH of 7.0. The bile acid mixture consisted of 25% deoxycholic acid, 45% glycocholic acid and 30% taurochenodeoxycholic acid. Cells were subsequently washed using PBS and given normal medium. Bile experiments were performed twice in duplicate. After 24 h, the cells were harvested and RNA was isolated. Methods were derived from Bus et al.[76]. To assess the effect of bile/acid on expansion of cells, EPC2-hTERT cells transduced with *HOXA13* or control vector as described above were seeded in 96-well plate with at least 2 wells per condition. Next day, medium was replaced with 100 µl of bile/acid mixture in cell culture medium (50 µM of sodium glycocholatenhydrate, 50 µM taurochenodeoxycholic acid, pH=4.95). After incubation for 4 days, MTT test was performed as described below. Experiment was performed at least five times.

**BAR-T spatial distribution experiments**. These were performed with three biological replicate cell lines containing *HOXA13* knock-out and three control cell lines. 40.000 BAR-T cells were seeded in a 6 well plate and pictures were taken the second day after seeding. Per well three pictures were taken. These pictures were analyzed using FIJI, using the multipoint tool, an X and Y (pixel) coordinate table was generated[62]. The distance between each cell and its three closest neighbors was quantified using Microsoft Excel and analyzed by two sided student's t-test. The experiment was performed in three independent cell lines and repeated three times.

**MTT assay**. For assessment of cell growth of EPC2 and BAR-T cells, we performed a 3-(4,5-dimethylthiazol-2-yl)-2,5-diphenyltetrazolium bromide (MTT) assay[77]. We seeded 1000 cells per well in 96 well plates for each of the three wild-type and three *HOXA13* knock-out cell lines. Per condition at least 2 wells were used. On days one, three, five, and seven 10 µl MTT at 5 µg/ml was added and incubated for three hours, the medium was removed, and the precipitate was dissolved in 100 µl DMSO, which was incubated for five minutes under continuous shaking. For BAR-T cells, absorption was measured in a BioRad microplate reader Model 680 XR at 490 and 595 nm, the average absorption was used to process the data. For EPC2

cells it was measured with Tecan microplate reader Model Infinite 200 pro at 565 nm with reference wavelength 670 nm. The experiment was repeated three times and a two sided Student's t-test was used to test for statistical significance.

**3D culture EPC2-hTERT cells**. 3D culturing of EPC2-hTERT cells was performed as previously described[48]. 4000 EPC2-hTERT cells in culture medium were mixed 1:1 with ice-cold Matrigel basement membrane matrix (Corning BV), seeded in 50 µl drops in a 24 well plate for cell suspension, and incubated at 37 ℃ for 30 min. After solidification, 500 µl of culturing medium supplemented with 0.6 mM CaCl2 was added. Y27632 (10 µM) was included in medium only the first 24 h after seeding. Medium was refreshed and pictures were taken every three days. The morphology of spheroids (based on number of extrusions, or 'invadosomes') was counted on day 5. The area of the spheroids was measured with FIJI[60]. For H&E staining and IHC analysis of involucrin (see above), spheroids were fixed in 4% formaldehyde for 7 min on day 11, washed with PBS, put in 2% agarose, and embedded in paraffin, then 4 µM slices were sectioned. Quantification was based on the percentage of positive cells and the intensity of the staining (scores ranged from 0, 2 to 9).

**Organotypic air-liquid interface culture**. Plate inserts (Sigma-Aldrich, Germany) were covered with bovine collagen I (Thermo Fisher Scientific, USA). The fibroblast (3T3-Swiss albino) feeder layer was embedded within a collagen matrix and was allowed to mature for 7 days, after which time BAR-T *HOXA13* knock-out and control epithelial cells were seeded on top and allowed to grow to confluence for an additional 3 days as described[78]. Then the culture media level of the upper well was reduced, exposing the apical side of keratinocytes to the air, while maintaining liquid levels at the basolateral side. On day 15, cultures were harvested for histologic examination. 4 µM paraffin-embedded sections were deparaffinized, and staining with hematoxylin and eosin, PAS staining and immunohistochemistry for involucrin were performed.

**Rat trachea in vivo tissue reconstitution model**. 500,000 parental BAR-T cells (derived from six independent clones) or *HOXA13* knock-out clones (three independent clones) in 30 µl of medium were sealed in the lumen of devitalized and denuded rat tracheas and implanted under the dorsal skin of NOD SCID gamma mice as described by Croagh et al.[79]. Mice were housed in microisolator cages with a 14 h light/10 h dark cycle, standard chow and water ad libitum, and temperature and humidity maintained at $21 \pm 1$ °C and $50 \pm 10\%$, respectively. Mice were sacrificed after four or six weeks. Harvested rat tracheas were formalin fixed, decalcified, embedded in paraffin and sectioned. Staining with hematoxylin and eosin, alcian blue, PAS staining and immunohistochemistry using antibodies against human mitochondria, CK7, TFF3, CDX2, p63, CK5, and involucrin (a gift from Prof. Pritinder Kaur, Curtin University, Australia or #I9018-100UL from Sigma-Aldrich) were performed (Supplementary Fig. 9 and Table 4). These murine experiments were approved by the Peter MacCallum Cancer Centre Animal Experimentation Ethics Committee and were performed according to the guidelines of the same institution.

**Reporting summary**. Further information on research design is available in the Nature Research Reporting Summary linked to this article.

## Data availability
The RNA sequence data discussed in this publication have been deposited in NCBI's Gene Expression Omnibus[80] and are accessible through GEO Series accession number GSE173170[81]. There are no restrictions regarding data availability. Supplemental figures, tables, and a method are included. Source data are provided with this paper and all relevant data are available from the authors.

Datasets used in the manuscript: RNA seq data GSE57584[15], GSE65013[18], single cell RNA data seq[21], GSE134520[82], GSE81861[83]. Source data are provided with this paper.

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

## Acknowledgements

We would like to acknowledge H.F.B.M. Sleddens, H. Stoop, M.H.W. van Dullemen, P. Vasic, I.T.A. Edelijn, M.J. van der Lee, P.J. Zwalua, W.W. van Dam, E. Zielhuis, J. Knoop, M. Doukas, A.L. Nigg and T.P.P. van den Bosch, Erasmus MC - University Medical Center Rotterdam, for their involvement in investigation, W.N.M. Dinjens for providing material, and F. McKeon, The Jackson Laboratory for Genomic Medicine, for conceptualization and providing resources. FAPESP n. 2016/01139-0; 2017/01046-5 for funding.

M.Magierowski was supported by a grant from National Science Centre (Poland): UMO-2016/23/D/NZ4/01913.

## Author contributions

Conceptualization, V.T.J., M.P.P., A.W., J.C., E.J.K., L.J.W.L., W.A.P., M.C.W.S., K.N., G.F. and R.S.; Methodology, V.T.J., G.M.F., M.P.P., R.A.S., N.C., R.S., K.N.; Formal Analysis, V.T.J., K.N., R.A.S.; Investigation, V.T.J., R.A.S., N.C., K.N.; Resources, M.C.W.S., H.S.S., E.M.T., T.S.G.; Writing – Original Draft, V.T.J., K.N. G.M.F. and A.P.V.; Writing – Review & Editing, V.T.J., K.N., M.P.P., M.C.W.S, A.P.V., A.W., G.M.F., R.A.S, R.S., M.J.B., N.C., W.A.P., J.C., E.J.K., L.J.W.L., H.S.S., E.M.T., T.S.G., J.C., M.M., E.J.K., L.J.W.L., K.N.; Funding Acquisition, M.C.W.S, M.P.P., M.J.B., V.T.J., W.A.P.; Visualization, V.T.J., N.C., A.P.V., K.N.; Supervision, M.P.P, M.J.B., M.C.W.S., G.M.F.

## Competing interests

M.J.B.: Cook Medical; consultant, support for industry and investigator initiated studies. Boston Scientific; consultant, support for industry and investigator initiated studies. Pentax Medical; support for investigator initiated studies. Mylan; support for investigator initiated studies. ChiRoStim; support for investigator initiated studies. 3 M; support for investigator initiated studies. The remaining authors declare no competing interests.
