## [Peer Review File · Nature Communications]

Reviewers' Comments:

Reviewer #1:

Remarks to the Author:

Janmaat et al has employed several model systems to study the potential role of HOXA13 in the etiology and progression of Barrett's esophagus. Using existing gene expression data on BE and intestinal stem cells, they found HOXA13 expresses in BE stem cells and colonic stem cells. They found that HOXA13-expressing cells exist in SCJ junctions of both mouse and human. Using ES, BAR-T culture model system, they attempt to establish the role of HOXA13 in cell fate determination. While Janmaat et al. employed a number of biological models and bioinformatics methods trying to address a very critical question of how BE originates and finally progresses to EAC, the authors did not provide convincing and conclusive evidence to support their hypothesis that HOXA13 plays sufficient and essential role in BE. I listed my major concerns about the paper here.

1. Using the data previously published, the authors examined HOX gene expression in the regiospecific stem cells of human intestine and their in vitro differentiated epithelia. The authors provided descriptive data that implicate the roles of HOX genes in the regulation of the identity of these stem cells. It is unclear how the observation of regiospecific stem cells can be linked with BE stem cell study that the authors presented next since BE and colon are two distinct types of epithelium.
2. The authors then studied various metaplasia for HOXA13 expression. While the data are somewhat interesting, but they are quite descriptive and don't help to address the link between HOXA13 and BE.
3. The authors then used a transgenic mouse model in which they can trace HOXA13 protein expression using GFP due to the lack of valid HOXA13 ab for immunohistochemistry. Interestingly, the authors found a group of cells at the SCJ of mouse stomach that express HOXA13 based on GFP expression. The authors did not provide very convincing images showing that the GFP expression is not detected in either proximal or distal stomach. In addition, the authors mentioned the recent paper regarding transitional basal cells and the confusion and controversies in the BE cellular origin field. The authors should use this transgenic model further address these controversies, for example perform immunostaining using Krt5, p63, Krt7 antibodies and examine the expression patterns of these markers in comparison GFP expression.
4. The authors then switched the model system to murine embryonic stem cells and began to study the transcriptome analysis of HOXA13 overexpressing mES. While the authors obtained some interesting results, it is hard to link the observation in mES cell model with Barrett's esophagus. It also doesn't address critical question on the cellular origin of BE, whether it comes from the HOXA13 cells at the SCJ for example?
5. The authors then used the EPC2-hTERT (squamous) and BAR-T (BE) cell lines to further examine the HOXA13 function. The authors mainly used bioinformatics tools to predict the possible roles of HOXA13 in cell differentiation and proliferation. It seems that the authors are now trying to examine the possibility of HOXA13 involvement in transcommitment. However, there is no phenotypic characterization. For instance, KO HOXA13 will change EPC2 to BE? The authors also mentioned earlier that acid treatment did not induce BE phenotype in squamous cell lines, so it is hard to understand the relevance of KO HOXA13 in esophagus with BE etiology. In general the bioinformatics analysis is quite descriptive and is not the basis for drawing any conclusion.
6. The authors then used a xenograft model to study BAR-T cells and HOXA13 KO in vivo. The parental BAR-T cells were derived from esophageal biopsy and immortalized using HTERT as a mixed cell population including both esophageal squamous and BE cells. It is misconception to use a pool of cells instead of single-cell derived clones to claim that BE cells can transdifferentiate into

squamous lineages. For HOXA13 KO cell lines that the authors used in this manuscript, the authors should establish single cell derived clones from BAR-T cell line and then knock out HOXA13. It is important for the author comparing the KO pedigree with the parental pedigree in order to draw any conclusion. Thus, Fig 6 in current manuscript is not a valid experiment.

7. The discussion in Janmatt et al tried to cover a lot of controversial papers and it is somewhat confusing. It is unclear whether the current manuscript addressed any critical questions in the BE field. For example, what is the cellular origin of BE? How BE progresses to dysplasia and cancer?

In summary, Janmatt et al. has some very exciting data regarding a population of HOXA13-expressing cells at the GEJ of both mouse and humans, and if developed could say much about the origins of Barrett's that remains controversial. However, they do much to obscure this in the figure itself, and the manuscript needlessly wanders away from this point. A more complete presentation of this finding and a refocus of this work could add much to solving questions regarding Barrett's esophagus.

Reviewer #2:

Remarks to the Author:

In this manuscript Janmaat et al suggest that HOXA13 is a putative transcription factor responsible for the metaplastic switch in the oesophagus from squamous epithelium to a caudal phenotype with intestinal differentiation. The authors further suggest that HOXA13 expression confers a selective growth advantage with activation of pathways with oncogenic potential. The critical finding in support of the hypothesis is that a patch of cells located at the gastro-esophageal junction which express HOXA13 that interrupts the expected co-linearity of Hox gene expression. They thus extrapolate that upon acid reflux induce damage these cells expressing HOXA13 proliferate and outgrow the neighbouring cell type and give rise to Barrett's esophagus.

There is a lot of detailed data presented and it is impressive to see human, embryonic, murine and in vitro models presented. However, the manuscript is challenging to follow and one gets the impression that the field of enquiry is quickly narrowed to fit a pre-determined set of ideas. For example, why did the authors focus entirely on HOXA13 given some of their other Hox gene findings. For example, A11, B6,9,13 seem to have a similar expression change in Barrett's vs squamous epithelium and in air-liquid interface cultures (but labelling of axes is very difficult to read). This is particularly surprising given other published data (and abstracts) about the putative role of Hox genes in Barrett's metaplasia that are consistent with a role of some of these other Hox genes which are likely to act in concert. I would be more convinced if the data evidence was expanded beyond HOXA13.

Major points:

1) The key piece of evidence in the manuscript to support their hypothesis is the expression of HOXA13 at the junction. The RNA in situ hybridisation signal in Fig 4I is convincing but we see one representative image of 3. Did the other two samples show the same expression or is it variable? Why weren't more samples analysed? Did the authors examine bulk RNA expression from the normal GEJ, samples could be taken right at the junction of healthy individuals as done for the samples within different epithelia shown in figure 1. If the expression levels are too low did they consider single cell RNA seq? In the discussion the authors discuss how these data could be consistent with the p63+KRT5+KRT7+ basal cells proposed as Barrett's progenitor (Nat Comm ref 43) – did they look to see if these cells co-express HOXA13 or vice versa?

2) The authors hypothesise that the HOXA13 positive cells might give rise to Barrett's in the context of inflammation from reflux. Did they examine for evidence of reflux-induced inflammation in the individuals that they sampled? It would be very feasible to sample the GEJ in patients with reflux esophagitis to see if the HOXA13 population is expanded or nearer to the surface.

3) For the embryonic data it should be noted that human oesophagus undergoes transition from columnar to squamous phenotype at 4-5 months of development. Therefore, the authors should confirm that the images in figure 4J show a fully mature squamo-columnar junction, not one that is undergoing transition to a squamous phenotype (please note the dilated nature of glands in images J1 and J3). It would be fascinating to have longitudinal data over this 4-5 month transition period though clearly this may not be feasible.

4) There is a lack of direct evidence presented that HOXA13 cells are activated upon tissue damage. A DTA-Krt14Cre HOXA13-GFP mouse model (Wang Cell 2011) would be helpful to observe the expansion of the HOXA13 cells.

5) The nature of the BAR-T cells is intriguing as the authors suggest they might undergo both squamous and columnar development in in vivo culture. The origin of this phenotype should be addressed as BAR-T cells might be a mixture of the squamous and BE cell lines and HOXA13 manipulations presented in figure 6E might result from changes in relative number of cells or complete depletion of the columnar phenotype from the knock-out experiments as they were formed from expanded single cell clones. The comparison could be performed by KRT7 or 8/18 (for BE) and KRT13 (for squamous oesophagus) staining. In addition, the non-transfected controls and time courses should be shown for comparison – the changes in the phenotype were not clear to me.

6) The authors claim that the low level up-regulation of HAXA13 following exposure to acid and bile, is due to a selective advantage of cells with high expression of HOXA13. This conclusion is questionable and should be substantiated showing viability of cells under treatment with acid and bile compared to cells transfected with HOXA13.

7) The claims about the oncogenic potential of these cells is over-stated given the limited data shown. The authors should restrict the discussion to the induction of metaplasia in keeping with the data shown.

Text inaccuracies and other comments:

I found it very challenging to follow the figures while reading the text and in fact on some occasions the authors appears to refer to wrong figures. Often the experiments being conducted were not clear from the text and could only be understood from the figure legends. Examples of inaccuracies are:

- on page 4 “We observed a decrease in 5-methylcytosine and an increase in 5-90 hydroxymethylcytosine in the promoters of HOXA13 and HOTTIP (extended data fig. 5)”. I cannot find these data in the figure
- Or when again they refer to extended Fig 5 in the legend of Fig 2. I think this is wrong
- I also find not very helpful to keep referring to other figures in figure legends.

The last paragraph in page 3 and the second paragraph in page 4 are repetitive, and present the same concept.

It is confusing that the authors refer to different reference in relation to the use of publicly available data (ref 15 and 18). In relation to this Figure 2 should mention that the data presented refer to publicly available data.

In the introduction the authors say that gastric inlet patches have high risk for adenocarcinoma. I disagree with this. Malignant transformation of inlet patch is anecdotal and at odds with the relatively high incidence of inlet patches in the general population.

Fig 3A. the bottom ISH inserts are not necessary and should be removed . They are the same as in the lower panel and I find inappropriate that the authors placed black marks on the ish dots to make them more visible.

What is the functional explanation of the crypt clonality and the epithelial mesenchymal pattern of expression of HOXA13 in the murine colon? I find this hard to interpret.

Figure 5 was difficult to follow and contains more explanatory cartoons than actual data.

Fig 6 F. The difference between left and right is not clear. In addition, what do the arrows represent?

Lines 106-125 (page 4) discuss four subpanels of figure 3 (A-D) however the figure only has 3 panel with figure legend missing information about the images in the bottom left corner.

Figure 5C: what is the meaning of DE cells on the graph.

Rebecca Fitzgerald, University of Cambridge

Reviewer #3:

Remarks to the Author:

The pathogenesis of Barrett's esophagus and heterotopias are of clear clinical relevance, and a diverse set of original experimental evidence is presented here to claim that HOXA13 expression in derivatives of the normal foregut is a major factor in this pathology. Besides the clinical relevance a theoretical innovation is implicit in this work which goes against the consensus based on many years of investigation in many laboratories. To establish an important exception to the collinearity view of Hox gene expression in the gastro-intestinal system, exceptionally well documented evidence is necessary. I am afraid that the evidence provided in this manuscript does not reach the requisite quality of documentation.

Describing the adult expression domains of developmental control genes like Hox genes is a major challenge, given that these genes typically retain well detectable steady state transcript levels only in rare contexts. Antibody selectivity is very rarely established to be exclusive for a single gene product, due to functionally important regions of amino acid homology, therefore frequent epitope sharing by immunogens. This is especially limiting in case of Hox/HOX genes like the 4, 9 and 13 paralogous groups, where four gene products must be discriminated. The possibility of using GFP and Cre knock-in alleles in the case of mouse Hoxa13 is a precious opportunity to perform very high quality analysis, including organ, tissue, cellular and sub-cellular levels. The compilation of diverse analyses presented in Figures 3 and 4 makes progress in this direction, but in my view the results remain preliminary at best.

1/As far as collinearity is concerned, this manuscript presents an extensive investigation of distinct anatomical regions of both human and mouse adult GI in nine segments from esophagus to rectum (Figure 1 and extended Figure 1). In general, it is remarkable how well the overall relative adult expression patterns conform to the collinear generalization derived from fetal mouse samples. Some notable exceptions seem to occur, as for instance mouse Hoxd8, which scored highest in esophagus as opposed to more caudal regions. It is likely to involve some artefact however, because this observation was not reproduced in human specimens. The observations of pathological specimens likely to represent genuine HOXA13 gain of expression (Figure 2 and 3). Overall this part of the work and the modulation of HOXA13 (see comment 8) in functional assays are the strong claims that justify publication.

2/An important element of the pathological mechanism envisaged by the current manuscript is the

role of Hoxa13 expressing cells in the pre-morbid esophagus in adult patients. The argument is based on the esophagus and Colon sample comparisons. In Figure 3A we see that qPCR analysis of Hoxa13 in GIP and Colon give rather similar expression levels, certainly not an order of magnitude difference is visible here. Below the "Esophagus" subtitle RNAscope histological photomicrographs are depicted: hardly anything is visible in GIP in the form of very fine brown granules, while in Colon the brown reaction product is easily apparent due to the much larger diameter of the granules. Apparently, the same histological panels are also shown under the qPCR graph, but there their overall RNAscope signal load appears quite similar, due to a "manual black dot correction". In other words, similar RNA level with qPCR corresponds to orders of magnitude more robust RNAscope signal in case of Colon, which than manually became homogenized to give the visual impression that in esophagus/GIP the level of HOXA13 expression is comparable to Colon. The poor correspondence of qPCR and RNAscope signals is also indicated by the comparison of finding very high qPCR expression associated with very weak RNAscope signal in MD gastric heterotopia, compared to the order of magnitude lower qPCR signal associated to comparatively robust RNAscope signal in case of Colon. All this is complicated to explain, if at all possible. In any case, I find it inadmissible that RNAscope signal be shown as black dots added to photomicrographs, that effectively hide the actual experimental result. If the images are not good enough to be shown in a technical publication, like Nature Communications, they should not be taken as evidence of any claim. These considerations together with the analysis of Figure 4 below cast serious doubt on the claim that there exists a HOXA13 expressing band of cells in normal adult human esophagus.

3/Figure 4 A-F and H2 of this manuscript document observations of the Hoxa13GFP knock-in heterozygous adult mouse colon. The overall analysis of the authors may perhaps be accurate, but from these panels the only fluorescent image that is convincing is shown in panel E. There the green fluorescent signal is clearly restricted to the epithelium, while the underlying mesoderm is free of fluorescence. The strong signal in the columnar intestinal epithelium is highlighted, and in the squamous anal epithelium is considered as negative. Yet, unexpectedly, the strongly marked columnar epithelial cells show signal both in nucleus and in cytoplasm. Given that Hoxa13GFP is presumed to be localized to the nucleus, the signal in the cytoplasm needs to be explained. In addition, the image on panel A suggests continued fluorescent domain into the anal squamous epithelium, in contrast to the author's claim, and a more permissive scoring of the high power image in panel E seem to confirm expression in both columnar rectal and squamous anal epithelia.

4/An intriguing claim of the authors' is crypt-clonal expression of Hoxa13GFP in the columnar epithelium, Figure 4D. Such mosaic character was not expected from public fetal analyses (http://www.eurexpress.org/ee/databases/showAssayImgs.jsp?image=euxassay_000981_01.jpg epithelial in descending colon), and this phenomenon in itself would merit an in-depth study. However, since in this case the observation arises in sub-optimal methodical context, the result should be considered preliminary, at best.

5/The most relevant piece of evidence concerning adult mice, in panel H2, shows the presence of seven highly fluorescent cellular profiles at the squamous glandular epithelial junction in stomach. At least the first of these profiles is apparently not in the epithelial layer, and the last two show a clear hole at the central position, that suggests cytoplasmic, rather than the expected nuclear localization of the signal. Both these details cast doubt on the presence of Hoxa13GFP fusion protein in a patch of epithelial cells in the adult mouse gastro-esophageal junction.

In fluorescence imaging a source of potential artefacts may arise from the use of frozen samples. Observing GFP fluorescence after freezing, storage, thawing and cryo-sectioning compromise cellular integrity, thus the consistent evaluation of tissue distribution. Typically, such observations are made on fresh non-fixed tissues. When cryo-sections are imaged, the Hoxa13GFP signal still should remain predominantly nuclear (see for instance in case of Hoxa13GFP: Development 2001 128: 4177-4188, Fig 8G,H). At this stage I remain dubious as to the reliability of fluorescent imaging in this set of analyses.

6/A further issue of assay reliability concerns Figure 4 J1,2,3. In all cases a robust ISH signal is expected to be cytoplasmic, and fairly homogenous over the cytoplasm. The positive signal claimed here is present as small puncta over many cells, occasionally present over nuclear profiles, J3. This complicates the evaluation of J3 near human fetal gastro-esophageal junction, especially since no concordant Hoxa13GFP expression was reported here or elsewhere at this region in fetal mice, and the distribution is also different from the rare cell profiles seen in adult human gastro-esophageal junction (Figure 4I, see below). Regrettably, despite the accessibility of the material, the authors report no effort to detect Hoxa13GFP in developing mouse stomach, although they claim HOXA13 transcript accumulation at the respective-duodenal epithelial transition in fetal human samples.

7/The truly relevant adult human esophageal region is documented in panel Figure 4I. Here seven foci of brown labelling are shown, perhaps four of which should include sections of nuclei. The prediction again is that nucleoplasm is free of homogenous ISH signal. In all four of these profiles the brown signal seems to cover the entire cell homogeneously, raising the possibility of artefacts. Taken as the authors intend, it again may at best be considered as a preliminary indication of HOXA13 expression near adult human gastro-esophageal junction. This type of analysis should be extended to more specimens to provide solid evidence of in situ HOXA13 mRNA accumulation in normal adult esophagus, which is the most controversial information concerning the etiology of Barrett's esophagus pathology in this manuscript.

In summary of imaging studies, it seems plausible that RNAscope findings in adult normal esophagus, and in adult mouse transitional epithelium are a reflection of variations in assay conditions, giving variable noise on a backdrop of essentially negative Hoxa13 expression status (<https://gtexportal.org/home/gene/HOXA13>).

8/Figures 5 and 6 document ingenious in vitro and organotypic tissue reconstruction functional assays involving HOXA13 overexpression and CrisprCas9 mediated loss of function in diverse cell lines. In aggregate, the results suggest that HOXA13 contributes to control of epithelial cell proliferation and differentiation in ways consistent with a role in pathogenesis of intestinal metaplasia in the GI system.

In conclusion normal human fetal, human adult and mouse adult esophageal-gastric epithelial junctions seem to show positive Hoxa13 expression signals reported in this manuscript, but the patterns do not really correspond, and are never truly convincing in any of the three cases, while the fetal mouse corresponding region is either negative or non-analyzed. This combined evidence does not lend strong support to the presence of a Hoxa13/HOXA13 positive region at the squamous-glandular esophageal-gastric epithelial transition region (see Extended data Figure7).

It is intriguing, nevertheless, that a previous study of ten independent specimens of Barrett's esophagus did not indicate any overexpression of Hoxa13, as compared to normal esophagus (10.1073/pnas.1116933109) and inversely, the current study does find an overexpression of the same HOXB genes in specimens from thirteen independent patients with similar pathology. In the previous study midsection HOXB genes, in this one HOXA13 are pinpointed. Perhaps both or neither are to be taken at face value. Furthermore, it is instructive to consider the observation of hamartomas containing both squamous and gastric glandular epithelium in stomach of transgenic mice overexpressing a midsection HoxC gene ([https://doi.org/10.1016/0092-8674\(92\)90388-S](https://doi.org/10.1016/0092-8674(92)90388-S)). Although very artificial, that experiment really raised the possibility that forced massive expression of a HOX gene may cause foregut epithelial misspecification. If thus may be consistent with the experiments presented here, there is no conclusive evidence that an anterior expression of HOXA13 in normal human esophagus has causal relevance to initiation, maintenance or aggravation of the pathological process in either of the human studies.

Minor points:

The reference given to indicate the source of the Hoxa13GFP mouse line, and the methods of its

analyses is in error. *J. Cell. Biochem.* 112: 1022–1034, 2011 does not concern that mouse line in any way.

Extended data Figure 3 do not refer to the HOXA complex, it is difficult to see how it is relevant to this HOXA13 centered manuscript.

The summary figure (see Extended data Figure 7), ends with reference to malignancy, which at this stage seems more a hypothesis than summary of such data issuing from conclusive investigation.

Reviewer #1 (Remarks to the Author):

Janmaat et al has employed several model systems to study the potential role of HOXA13 in the etiology and progression of Barrett's esophagus. Using existing gene expression data on BE and intestinal stem cells, they found HOXA13 expresses in BE stem cells and colonic stem cells. They found that HOXA13-expressing cells exist in SCJ junctions of both mouse and human. Using ES, BAR-T culture model system, they attempt to establish the role of HOXA13 in cell fate determination. While Janmaat et al. employed a number of biological models and bioinformatics methods trying to address a very critical question of how BE originates and finally progresses to EAC, the authors did not provide convincing and conclusive evidence to support their hypothesis that HOXA13 plays sufficient and essential role in BE. I listed my major concerns about the paper here.

1. Using the data previously published, the authors examined HOX gene expression in the regiospecific stem cells of human intestine and their in vitro differentiated epithelia. The authors provided descriptive data that implicate the roles of HOX genes in the regulation of the identity of these stem cells. It is unclear how the observation of regiospecific stem cells can be linked with BE stem cell study that the authors presented next since BE and colon are two distinct types of epithelium.

Reply: We performed this analysis in order to objectify the notion that *HOX* coding is established in the intestine in general at the stem cell level. Indeed, our data shows that in both the normal epithelium, where *HOX* gene shows a tight regional regulation along the rostro-caudal axis, as well as BE epithelium, where *HOX* genes are aberrantly expressed, *HOX* genetic imprinting has already occurred at the stem cell stage. Whereas obviously BE and colon epithelia are not identical (for example in terms of heterogeneity), histological features of BE greatly overlap with colonic epithelium, including the presence of enterocytes, enteroendocrine cells, and goblet cells [1]. In apparent agreement, we demonstrate that BE epithelia also have molecular overlap with colonic epithelia in terms of a similar *HOX* gene expression pattern (**Figure 1**). The point we aim to make using data from Wang *et al* [2], is that this *HOX* coding is established early in differentiation and is already present in the location-specific stem cells along the gastrointestinal tract. Our subsequent interrogation of the data from Yamamoto *et al* [3] demonstrates that this colon-specific pattern is also present in the BE stem cell. We consider this evidence of importance because existing theories regarding cells of origin of BE (except the transdifferentiation theory), suggest that BE arises from stem-like progenitor cells that can give rise to all different cell types of BE [4]. Thus, this data shows that inherent location identity drivers are a feature of the early gastrointestinal stem cells, and that the BE stem cells resemble colonic stem cells in their *HOX* expression pattern. The manuscript has been extensively rewritten and the text has been amended to clarify our rationale.

2. *The authors then studied various metaplasia for HOXA13 expression. While the data are somewhat interesting, but they are quite descriptive and don't help to address the link between HOXA13 and BE.*

Reply: The aim of these data is to show that the biological principle of deregulated *HOXA13* expression in metaplasia extends beyond Barrett's metaplasia alone. Our findings of an interruption of collinearity in these other metaplasias suggest that the molecular principles that we study in more detail for Barrett's are also relevant to other heteropias of the gut, extending the importance of our findings to other, related fields of study.

3. *The authors then used a transgenic mouse model in which they can trace HOXA13 protein expression using GFP due to the lack of valid HOXA13 ab for immunohistochemistry. Interestingly, the authors found a group of cells at the SCJ of mouse stomach that express HOXA13 based on GFP expression. A) The authors did not provide very convincing images showing that the GFP expression is not detected in either proximal or distal stomach.*

Reply: To address the reviewers concerns regarding the quality of images, we have now also performed anti-GFP immunohistochemistry on these samples, which confirms our previous findings obtained by immunofluorescence, and provide more convincing images of *Hoxa13* expression in mouse GI tracts. This approach excludes possible false positivity due to auto-fluorescence. Furthermore, rather than showing a representative example, we now included images of three GFP-positive mice and one GFP-negative littermate (**Extended data fig. 7**). In line with our previous results, we still do not see *Hoxa13* expression proximal to the GEJ and in stroma of upper GI tract of mice. However, in addition to the GEJ, we detect rare *Hoxa13*-positive cells along the basal side of stomach epithelium of all three mice. We also provide a novel analysis of publicly available single cell RNAseq data (#EGAS00001003144, recently described by Owen *et al.* in Nature Communications [6], #GSE134520 by Zhang *et al.* in Cell Rep [5]), demonstrating that in men, *HOXA13*-positive cells are present in the human normal squamous esophagus in low numbers in the esophageal submucosal glands and stomach, compared to BE esophagus and gastric intestinal metaplasia, respectively. The fact that some *HOXA13*-positive cells are found in human esophagus, while not in mice, may represent species differences, although technical differences may also play a part: cutting sections from tissue reduces the number of cells assessed simultaneously, while sorting of cells for RNAseq enriches for cells able to withstand flow-sort shear stress without undergoing apoptosis – *HOXA13*-positive cells may be among this population. With these new data, we now conclude that rare, single *HOXA13*⁺ cells are indeed present in normal physiology, including at the GEJ and number of these cells are increased in BE.

B) *In addition, the authors mentioned the recent paper regarding transitional basal cells and the confusion and controversies in the BE cellular origin field. The authors should use this transgenic model further address these controversies, for example perform immunostaining using Krt5, p63, Krt7 antibodies and examine the expression patterns of these markers in comparison GFP expression.*

Reply: We appreciate this valid comment. To address this question, have analyzed the single cell RNAseq by Owen *et al* [6], as we felt this would be more robust than performing double

immunostaining on sections, or staining of consecutive sections. Although the GEJ was not sampled in this study, 4.18% of normal esophageal cells expressed *HOXA13*. Gene expression analysis indicates that this *HOXA13*-positive population derives from submucosal glands (new **Figure 6B**). This might be another explanation for the absence of *Hoxa13* in the mouse esophagus, as these animals lack such submucosal glands. Owen *et al* [6] suggested that cells located in submucosal glands of the normal esophagus may be progenitors of BE because they exhibit transcriptional overlap with BE cells. Our analysis shows that *HOXA13*-positive cells from the normal esophagus also cluster with BE cells (new **Figure 6A**). *HOXA13* expression overlaps greatly with *KRT7*, a columnar cytokeratin seen in Barrett's, moderately with *KRT14*, but not with the basal cell marker *TP63* (see **Table 1** for the reviewer below). This suggests that *HOXA13* may be linked to a Barrett's gene signature and distal phenotype, and it is interesting to note that this is similar for both *HOXA13* expressing cells from the normal esophagus and from BE. A small population of cells co-expressing *KRT14* (a gene pair with *KRT5*), *TP63* and *KRT7* is seen in the normal esophagus proximal to the junction (n=19), however, these do not express *HOXA13*. The percentage of cells with this gene signature appears to be reduced in BE. Thus, these data suggest that the *HOXA13*⁺ compartment is not the same as the earlier suggested *KRT7*⁺*KRT14*⁺*TP63*⁺ cell of BE origin in the squamous esophagus. To note, *KRT7*⁺*KRT14*⁺*TP63*⁺ cells gave rise to BE-like epithelium only upon ectopic expression of *CDX2* [7].

Table 1 for reviewer 1. The total number of cells analyzed from the normal esophagus (N=886) and BE (N=395) were analyzed for expression of *HOXA13*, *TP63*, *KRT13* and *KRT7*. The absolute number of cells expressing these genes as well as the percentage of cells within each tissue are shown. The last column indicates the percentage overlap (e.g. 2.7% of *HOXA13*⁺ cells of the normal esophagus express p63).

Genes	N of cells in healthy esophagus (from 886)	% of total in healthy esophagus	N of cells in BE (from 395)	% of total in BE	% of HOXA13 ⁺ in healthy esophagus	% of HOXA13 ⁺ in BE
HOXA13	37	4.18	132	33.42		
p63	659	74.38	12	3.04		
KRT14	567	64.00	51	12.91		
KRT7	104	11.74	285	72.15		
p63, KRT7	17	1.92	10	2.53		
p63, KRT14	479	54.06	3	0.76		
KRT7 , KRT14	37	4.18	27	6.84		
HOXA13 , p63	1	0.11	4	1.01	2.70	3.03
HOXA13 , KRT14	4	0.45	15	3.80	10.81	11.36
HOXA13 , KRT7	32	3.61	105	26.58	86.49	79.55
HOXA13 , p63, KRT7	0	0.00	2	0.51	0.00	1.52
HOXA13 , p63, KRT14	0	0.00	1	0.25	0.00	0.76

HOXA13, KRT7, KRT14	4	0.45	13	3.29	10.81	9.85
KRT7, KRT14, TP63	19	2.14	2	0.51		
HOXA13, KRT7, KRT14, TP63	0		0			

4. The authors then switched the model system to murine embryonic stem cells and began to study the transcriptome analysis of *HOXA13* overexpressing mES. While the authors obtained some interesting results, it is hard to link the observation in mES cell model with Barrett's esophagus. It also doesn't address critical question on the cellular origin of BE, whether it comes from the *HOXA13* cells at the SCJ for example?

Reply: With this experiment we did not aim to solve the question of the cellular origin of BE *per se*. Rather, these studies help to address a different critical question regarding the molecular mechanisms underlying BE development. Taken the assumption that BE originates from cells with stemness properties [4] and *HOX* gene patterns are established at the level of stem cells rather than upon differentiation (**Figure 2**), we performed this analysis to investigate how expression of *HOXA13* in stem cells may affect their cellular and molecular properties and may affect their (caudal) differentiation potential. We employed the mES cell model because BE stem cell models are not readily available. This experiment indicates that *HOXA13* expression during endodermal differentiation supports proliferative potential. So while these experiments in themselves do not formally allow the conclusion that the single *HOXA13*-positive cells observed in the GI tract give rise to BE, they support the fact that these cells have the properties to do so.

5. The authors then used the EPC2-hTERT (squamous) and BAR-T (BE) cell lines to further examine the *HOXA13* function. The authors mainly used bioinformatics tools to predict the possible roles of *HOXA13* in cell differentiation and proliferation. A) It seems that the authors are now trying to examine the possibility of *HOXA13* involvement in transcommitment. However, there is no phenotypic characterization. For instance, KO *HOXA13* will change EPC2 to BE?

Reply: A) The set-up we chose to investigate the role of *HOXA13* in BE was to take a BE cell line (BAR-T) and knock down *HOXA13*, while reversely, overexpressing *HOXA13* in a normal esophageal line (EPC2, which does not express *HOXA13* endogenously). Studying the set of genes that go down in BAR-T upon knockdown, but simultaneously go up in EPC2 cells upon overexpression of *HOXA13*, gives a more robust indication as to the molecular consequences of *HOXA13* in processes directing BE development, compared to studying either cell model alone. We appreciated the reviewers suggestion regarding EPC2 phenotypic characterization and have now expanded our data to include functional and morphological characterization of these cells upon *HOXA13* overexpression. Cells were grown as 3D spheroids as described before [8], allowing investigation of differentiation of these cells. We have now included new quantitative data demonstrating that this cell line changes its morphology upon overexpression of *HOXA13*, with a loss of squamous cell stratification, confirming that *HOXA13* overexpression decreases differentiation of squamous epithelial cells (new **Figure 8A**). In addition, *HOXA13* overexpression confers a proliferative advantage to these cells and makes them less sensitive to bile/acid exposure (new **Figure 7E, F**). Additionally, we have further characterized BAR-T cell morphology upon knockout of *HOXA13* by performing PAS and anti-involucrin staining of the

in vivo trachea reconstitution model, and added quantitative analysis of morphological changes (**Figure 8C, D**). Together, these new data support a role for *HOXA13* in reducing normal squamous differentiation while facilitating colonic-like columnar differentiation.

B) The authors also mentioned earlier that acid treatment did not induce BE phenotype in squamous cell lines, so it is hard to understand the relevance of KO HOXA13 in esophagus with BE etiology. In general the bioinformatics analysis is quite descriptive and is not the basis for drawing any conclusion.

Reply: Indeed, we entertained the notion that reflux might affect *HOXA13* expression and thereby drive BE formation. However, results revealed that while *HOXA13* expression itself is not affected by bile/acid treatment, *HOXA13* provides a proliferative advantage to cells and makes them less sensitive to bile/acid exposure (new **Figure 7D, E, F**) The bioinformatics analysis we performed showed regulation of the epidermal differentiation complex by *HOXA13*, which is consistent with the new experimental data. Staining of IVL, which belongs to this complex, in both BAR-T knocked down for *HOXA13* and EPC2 cells with *HOXA13* overexpression confirms these data (**Figure 8A, D, E**). In addition, we used the BAR-T KO model for functional experiments demonstrating the relevance of *HOXA13* for proliferation of BE cells, the results of which are presented in **Figures 7 and 8**. We appreciate that while the link between bioinformatics data and functional data was perhaps not evident in the previous version of the manuscript, we hope that with the extensive alterations we have made, it has become more clear now.

6. The authors then used a xenograft model to study BAR-T cells and HOXA13 KO in vivo. The parental BAR-T cells were derived from esophageal biopsy and immortalized using HTERT as a mixed cell population including both esophageal squamous and BE cells. It is misconception to use a pool of cells instead of single-cell derived clones to claim that BE cells can transdifferentiate into squamous lineages. For HOXA13 KO cell lines that the authors used in this manuscript, the authors should establish single cell derived clones from BAR-T cell line and then knock out HOXA13. It is important for the author comparing the KO pedigree with the parental pedigree in order to draw any conclusion. Thus, Fig 6 in current manuscript is not a valid experiment.

Reply: We apologize for the confusion and for not providing enough detail with regard to our methodological approaches. Contrary to the reviewers suggestion, according to Jaiswal et al. [9], a single cell colony was selected to generate the BAR-T cell line. While mixed expression of columnar and squamous markers in this line has been described, single cell analysis was not performed, and this likely reflects the bilinear differentiation potential of BAR-T cells rather than heterogeneity from the original culture [10]. Before performing our experiments, we first tested these cells by FACS and found that co-expression of CK14 and CK7 occurs in nearly 100% of BAR-T cells, suggesting relative homogeneity under normal culture conditions (**Figure 1 for the reviewer** below).

Figure 1 for the reviewer. Staining of BAR-T [9] cells for CK7 and CK14. Cells were fixed in 2% paraformaldehyde for 10 min at room temperature, permeabilised with ice cold methanol for 10min at 4C and blocked in 5% bovine serum albumin in 0.05% Tween-20 in tris-buffered saline. Then cells were incubated with mouse anti-CK7 (1:1000, clone OV-TL 12/30 Dako cat#M7018) and guinea pig anti-CK14 (1:600, Progen Biotechnik, cat#GP-CK14). Donkey anti-mouse IgG AlexaFluor647 (1:2000, Molecular Probes, Cat# A3157) and goat anti-guinea pig IgG AlexaFluor488 (1:1500, Molecular Probes, Cat# A11073) were used as secondary antibodies. Cells were analysed by FACSCanto II (BD Biosciences).

More importantly, to verify the potential of individual BAR-T cells to differentiate into two lineages, we analyzed 10 single cell colonies derived from the parental BAR-T cell line, all of which showed commitment to two different lineages, squamous and columnar. This data is depicted in **extended data Figure 9** (the rat trachea epithelial reconstitution model), and proves that the capability to form two independent lineages is an inherent feature of individual BAR-T cells rather than a representation of a heterogeneous cell population. For our KO experiments in BAR-T cells, single cell colonies were derived upon gene editing by Crip1 Cas, and thus data do indeed represent individual clones, derived from individual cells. This is exactly as the reviewer suggests, and we apologize that this was not clear in our previous version of the manuscript. Thus, we feel that the validity of the results obtained with these lines should no longer be in question. We have now added the information of origin of BAR-T cells to the results section to avoid any potential confusion.

7. A) *The discussion in Janmatt et al tried to cover a lot of controversial papers and it is somewhat confusing. It is unclear whether the current manuscript addressed any critical questions in the BE field. For example, what is the cellular origin of BE? How BE progresses to dysplasia and cancer?*

Reply: The discussion has been extensively rewritten to address these important points, and we hope it now meets the reviewer's approval.

In summary, Janmaat et al. has some very exciting data regarding a population of HOXA13-expressing cells at the GEJ of both mouse and humans, and if developed could say much about the origins of Barrett's that remains controversial. However, they do much to obscure this in the figure itself, and the manuscript needlessly wanders away from this point. A more complete presentation of this finding and a refocus of this work could add much to solving questions regarding Barrett's esophagus.

Reply: We thank the reviewer for the constructive suggestions and positive comments on the inherent interest of our results. We feel we have now substantially improved the data on the HOXA13-positive population by showing better quality images, adding immunochemical stainings of GFP-positivity, and providing single cell RNAseq analyses to further characterize this population of cells. We paid more attention to the HOXA13-positive compartment in the

physiological setting, and have extensively modified our manuscript to include all these changes. We hope that with these changes we have considerably improved the manuscript, and have addressed all the reviewer's concerns.

Reference list:

1. Nakagawa, H., K. Whelan, and J.P. Lynch, *Mechanisms of Barrett's oesophagus: intestinal differentiation, stem cells, and tissue models*. Best practice & research Clinical gastroenterology, 2015. **29**(1): p. 3-16.
2. Wang, X., et al., *Cloning and variation of ground state intestinal stem cells*. Nature, 2015. **522**(7555): p. 173-8.
3. Yamamoto, Y., et al., *Mutational spectrum of Barrett's stem cells suggests paths to initiation of a precancerous lesion*. Nature Communications, 2016. **7**: p. 10380.
4. Que, J., et al., *Pathogenesis and Cells of Origin of Barrett's Esophagus*. Gastroenterology, 2019. **157**(2): p. 349-364.e1.
5. Zhang, P., et al., *Dissecting the Single-Cell Transcriptome Network Underlying Gastric Premalignant Lesions and Early Gastric Cancer*. Cell Rep, 2019. **27**(6): p. 1934-1947 e5.
6. Owen, R.P., et al., *Single cell RNA-seq reveals profound transcriptional similarity between Barrett's oesophagus and oesophageal submucosal glands*. Nature Communications, 2018. **9**(1): p. 4261.
7. Jiang, M., et al., *Transitional basal cells at the squamous-columnar junction generate Barrett's oesophagus*. Nature, 2017. **550**(7677): p. 529-533.
8. Kasagi, Y., et al., *The Esophageal Organoid System Reveals Functional Interplay Between Notch and Cytokines in Reactive Epithelial Changes*. Cellular and molecular gastroenterology and hepatology, 2018. **5**(3): p. 333-352.
9. Jaiswal, K.R., et al., *Characterization of telomerase-immortalized, non-neoplastic, human Barrett's cell line (BAR-T)*. Dis Esophagus, 2007. **20**(3): p. 256-64.
10. Bajpai, M., et al., *Repeated exposure to acid and bile selectively induces colonic phenotype expression in a heterogeneous Barrett's epithelial cell line*. Lab Invest, 2008. **88**(6): p. 643-51.

Reviewer #2 (Remarks to the Author):

In this manuscript Janmaat et al suggest that HOXA13 is a putative transcription factor responsible for the metaplastic switch in the oesophagus from squamous epithelium to a caudal phenotype with intestinal differentiation. The authors further suggest that HOXA13 expression confers a selective growth advantage with activation of pathways with oncogenic potential. The critical finding in support of the hypothesis is that a patch of cells located at the gastro-esophageal junction which express HOXA13 that interrupts the expected co-linearity of Hox gene expression. They thus extrapolate that upon acid reflux induce damage these cells expressing HOXA13 proliferate and outgrow the neighbouring cell type and give rise to Barrett's esophagus.

There is a lot of detailed data presented and it is impressive to see human, embryonic, murine and in vitro models presented. However, the manuscript is challenging to follow and one gets the impression that the field of enquiry is quickly narrowed to fit a pre-determined set of ideas. For example, why did the authors focus entirely on HOXA13 given some of their other Hox gene findings. For example, A11, B6,9,13 seem to have a similar expression change in Barrett's vs squamous epithelium and in air-liquid interface cultures (but labelling of axes is very difficult to read).

This is particularly surprising given other published data (and abstracts) about the putative role of Hox genes in Barrett's metaplasia that are consistent with a role of some of these other Hox genes which are likely to act in concert. I would be more convinced if the data evidence was expanded beyond HOXA13.

Reply: We apologize that the previous version of the manuscript was hard to read. We have extensively rewritten the manuscript, which we hope now makes it easier to place all data in their appropriate context. We agree with the reviewer that *HOX* genes are likely to act in tandem and that *HOXB 6, 9, 13* and probably also *HOXA11* and *HOXC10* would be interesting candidates for study as well. Our reason for emphasizing *HOXA13* is because it was the highest overexpressed paralogue group 13 member identified in Barrett's tissue compared to squamous tissue in absolute measurements (**Extended data fig. 4**). According to the collinear theory a paralogue group 13 member would confer more distal characteristics compared to a more anterior paralogue group member. An additional reason to focus on cluster A, with particular emphasis on *HOXA13*, is that it was less well studied compared to the *HOXB* cluster. At the same time, we of course do not discard the importance of the *HOXB* cluster or any of the other *HOX* genes. Having seen the major consequences on cellular morphology and proliferation by manipulation of only one *HOX* gene (*HOXA13*), we can only speculate whether manipulation of multiple *HOX* genes simultaneously would have an even stronger effect on the described processes. Additive and synergistic effects of *HOX* genes were described before [1-3]. Analysis of publicly available single cell RNA seq data (#EGAS00001003144, recently described by Owen *et al* [6]) demonstrates that *HOXA13* is expressed in non-squamous (*TFF3+*) ESMG cells of healthy esophagus and overlaps with other *HOX* genes within same cells. This overlap is imperfect, and mainly limited to the caudal *HOX* genes *HOXA11* and *HOXB13*, as well as *HOXB6* (See **Tables 1** for the reviewer below that show number of positive cells for several *HOX* genes, **Figure 1** for the reviewer – correlation of expression of these genes to *HOXA13* **Table 2 and 3** – summary of correlation of all detected *HOX* genes in *TFF3+* population with *HOXA13*). Thus, it is of interest to note in particular *HOX* genes of interest suggested by the reviewer (*A11, B6,9,13*) to a large extent correlate with *HOXA13* expression levels. However, in our dataset, modulation of *HOXA13* in two cell lines did not significantly affect any of the other *HOX* genes (as indicated by RNAseq data), demonstrating that in this instance, only *HOXA13* effects were studied in these

cell models. Studying the concerted action of these *HOX* genes in such a complex arrangement and an in depth molecular and cellular analysis of all these genes, like the ones performed for *HOXA13*, are beyond the scope of the present manuscript. The text was amended now to better delineate our rationale for choosing *HOXA13* for further study. Additionally, we now suggest other *HOX* genes as a topic for further research in the discussion section of the revised manuscript.

Table 1 for the reviewer 2. Expression of *HOX* genes in *TFF3*-positive cells of esophagus

Genes	Healthy esophagus		BE	
	N of positive cells	% of total (N=136)	N of positive cells	% of total (N=389)
HOXA13	36	26.47	132	33.93
HOXA11	25	18.38	87	22.37
HOXB6	47	34.56	182	46.79
HOXB9	9	6.62	31	7.97
HOXB13	23	16.91	154	39.59
HOXC10	34	25.00	105	26.99
HOXA13, HOXA11	15	11.03	36	9.25
HOXA13, HOXB6	24	17.65	77	19.79
HOXA13, HOXB9	4	2.94	7	1.80
HOXA13, HOXB13	12	8.82	64	16.45
HOXA13, HOXC10	14	10.29	44	11.31

Figure 1 for the reviewer 2. Spearman correlation of expression of *HOXA11*, *HOXB6*, *HOXB9*, *HOXB13*, *HOXC10* genes with *HOXA13* (Expression level per single cell in RPKM - Reads Per Kilobase Million , RNA seq data).

Table 2 for the reviewer 2. *HOX* gene expression correlation with *HOXA13* in *TFE3+* population of healthy esophagus (N of single cells =136). Spearman correlation was calculated for all detected *HOX* genes with *HOXA13* based on the presents or absent of expression in each cell: 0 – no expression detected, 1 – gene is expressed. Thus, correlation depends on the presents or absents this gene in one cell with *HOXA13* but not level on the expression in RPKM).

Healthy esophagus					N of positive cells
	Spearman r	95% CI	P value (two-tailed)	P value summary	
HOXA11	0.36	0.1999 to 0.5025	< 0.0001	***	25
HOXC8	0.09	-0.08818 to 0.2558	0.3173	ns	10
HOXA9	0.20	0.03205 to 0.3645	0.0171	*	8
HOXA1	-0.05	-0.2192 to 0.1266	0.581	ns	6
HOXA2	-0.07	-0.2435 to 0.1012	0.3964	ns	2
HOXA3	-0.04	-0.2130 to 0.1329	0.6329	ns	10
HOXA5	-0.07	-0.2395 to 0.1054	0.4239	ns	12
HOXA6	0.01	-0.1680 to 0.1785	0.9503	ns	11
HOXA13	1	1.000 to 1.000	< 0.0001	***	36
HOXB6	0.41	0.2493 to 0.5404	< 0.0001	***	47
HOXB8	0.18	0.002317 to 0.3385	0.041	*	9
HOXB5	0.27	0.1032 to 0.4248	0.0014	**	28
HOXB3	0.11	-0.06524 to 0.2773	0.2055	ns	38
HOXB1	-0.07	-0.2435 to 0.1012	0.3964	ns	2
HOXA7	-0.20	-0.3563 to -0.02257	0.0229	*	13
HOXC13	-0.01	-0.1789 to 0.1676	0.9465	ns	4
HOXC11	0.012	-0.1576 to 0.1889	0.8521	ns	14
HOXC12	-0.05	-0.2229 to 0.1227	0.5505	ns	1
HOXD1	-0.12	-0.2847 to 0.05719	0.1741	ns	5
HOXD3	0.15	-0.02543 to 0.3137	0.0845	ns	5
HOXD9	N/A				0
HOXD10	-0.05	-0.2229 to 0.1227	0.5505	ns	1
HOXD11	N/A				0
HOXD13	N/A				0
HOXB13	0.26	0.09387 to 0.4171	0.002	**	23
HOXD4	-0.05	-0.2229 to 0.1227	0.5505	ns	1
HOXD12	-0.03	-0.2009 to 0.1453	0.7405	ns	5
HOXB9	0.11	-0.06604 to 0.2765	0.2088	ns	9
HOXC5	N/A				0
HOXB2	0.07	-0.1072 to 0.2378	0.436	ns	53
HOXD8	N/A				0
HOXC9	-0.03	-0.1974 to 0.1489	0.7726	ns	13
HOXC10	0.19	0.01986 to 0.3539	0.0248	*	34
HOXB4	-0.03	-0.1976 to 0.1487	0.7709	ns	17
HOXA4	-0.01	-0.1789 to 0.1676	0.9465	ns	4
HOXC6	0.11	-0.05986 to 0.2822	0.1841	ns	6
HOXC4	N/A				0

Table 3 for the reviewer 2. **HOX genes expression correlation with HOXA13 in TFF3+ population in BE (N of single cells = 389)**. Spearman correlation was calculated for all detected *HOX* genes with *HOXA13* based on the presents or absent of expression in each cell: 0 – no expression detected, 1 – gene is expressed. Thus, correlation depends on the presents or absents this gene in one cell with *HOXA13* but not level on the expression in RPKM).

BE					N of positive cells
	Spearman r	95% CI	P value (two-tailed)	P value summary	
HOXA11	0.08	-0.01812 to 0.1852	0.0964	ns	87
HOXC8	0.07	-0.03605 to 0.1678	0.1901	ns	34
HOXA9	0.002	-0.1008 to 0.1039	0.9758	ns	41
HOXA1	0.09	-0.01307 to 0.1901	0.0781	ns	10
HOXA2	-0.03	-0.1355 to 0.06903	0.509	ns	5
HOXA3	-0.02	-0.1240 to 0.08065	0.6667	ns	42
HOXA5	-0.03	-0.1301 to 0.07447	0.5803	ns	40
HOXA6	-0.003	-0.1054 to 0.09938	0.9526	ns	21
HOXA13	1	1.000 to 1.000	< 0.0001	***	132
HOXB6	0.17	0.06457 to 0.2637	0.001	**	182
HOXB8	0.004	-0.09792 to 0.1068	0.9295	ns	23
HOXB5	0.01	-0.08924 to 0.1155	0.7944	ns	100
HOXB3	0.17	0.07226 to 0.2709	0.0006	***	180
HOXB1	-0.001	-0.1035 to 0.1013	0.9825	ns	3
HOXA7	0.05	-0.05347 to 0.1508	0.3334	ns	8
HOXC13	0.02	-0.07818 to 0.1265	0.6315	ns	2
HOXC11	0.04	-0.06243 to 0.1420	0.4292	ns	46
HOXC12	-0.06	-0.1645 to 0.03945	0.2137	ns	3
HOXD1	-0.04	-0.1382 to 0.06624	0.4743	ns	1
HOXD3	N/A				0
HOXD9	0.06	-0.03981 to 0.1641	0.2164	ns	5
HOXD10	-0.05	-0.1531 to 0.05112	0.3108	ns	2
HOXD11	-0.0011	-0.1035 to 0.1013	0.9825	ns	3
HOXD13	-0.07	-0.1668 to 0.03703	0.1967	ns	8
HOXB13	0.13	0.02837 to 0.2297	0.0101	*	154
HOXD4	-0.001	-0.1035 to 0.1013	0.9825	ns	3
HOXD12	-0.06	-0.1645 to 0.03945	0.2137	ns	3
HOXB9	-0.07	-0.1717 to 0.03205	0.1649	ns	31
HOXC5	0.10	-0.002087 to 0.2006	0.048	*	2
HOXB2	0.09	-0.01552 to 0.1877	0.0866	ns	148
HOXD8	0.02	-0.07818 to 0.1265	0.6315	ns	2
HOXC9	0.02	-0.08233 to 0.1223	0.691	ns	38
HOXC10	0.10	-0.000003261 to 0.2026	0.0436	*	105
HOXB4	0.02	-0.08576 to 0.1189	0.7417	ns	50
HOXA4	0.02	-0.07818 to 0.1265	0.6315	ns	2
HOXC6	-0.004	-0.1064 to 0.09837	0.9366	ns	36
HOXC4	-0.001	-0.1035 to 0.1013	0.9825	ns	3

Major points:

1) *The key piece of evidence in the manuscript to support their hypothesis is the expression of HOXA13 at the junction. A) The RNA in situ hybridisation signal in Fig 4I is convincing but we see one representative image of 3. Did the other two samples show the same expression or is it variable? Why weren't more samples analysed?*

Reply: We have now provided scanned images of all three samples, which show a similar positive signal for *HOXA13* at the GEJ (**Extended data figure 8**). Given three out of three samples showed signal and three individual experiments were performed we felt that proof of concept was sufficiently provided. Unfortunately, we cannot increase N of samples due to limited availability of material.

B) Did the authors examine bulk RNA expression from the normal GEJ, samples could be taken right at the junction of healthy individuals as done for the samples within different epithelia shown in figure 1. If the expression levels are too low did they consider single cell RNA seq? In the discussion the authors discuss how these data could be consistent with the p63+KRT5+KRT7+ basal cells proposed as Barrett's progenitor (Nat Comm ref 43) – did they look to see if these cells co-express HOXA13 or vice versa?

Reply: The reviewer makes an excellent point, and we now include new data that proves their supposition. We have performed new anti-GFP immunohistochemistry stainings on our mouse samples. In line with our previous results, we still do not see *Hoxa13* expression proximal to the GEJ and in stroma of the upper GI tract of mice. However, in addition to the GEJ, we detect rare *Hoxa13*-positive cells along basal side of stomach epithelium of all three mice (new **Figure 5A, Extended data fig. 7**). We also provide a novel analysis of publicly available single cell RNAseq data (#EGAS00001003144, recently described by Owen *et al* [6]). While the GEJ was not sampled in this study, this dataset shows that in man, some *HOXA13*-positive cells are present in the physiological human adult squamous esophagus. The difference between the human and mouse findings may originate from technical differences as cutting sections from tissue reduces the number of cells assessed simultaneously, while sorting of cells for RNAseq enriches for cells able to withstand flow-sort shear stress without undergoing apoptosis – *HOXA13*-positive cells may be among this population. However, it is also likely that species differences play a role. Gene expression analysis indicates that the *HOXA13*-positive population in humans derives from submucosal glands (new **Figure 6**), while mice lack these structures. With these new data, we now conclude that rare, single *HOXA13*⁺ cells are indeed present in physiology, at the ESMG's and the GEJ, the numbers of which are increased in BE.

Our analysis of the single cell RNAseq data from Owen *et al* [6] shows that *HOXA13* expression overlaps greatly with *KRT7*, a columnar cytokeratin seen in Barrett's (around 80% of *HOXA13*⁺ cells), moderately with *KRT14*, but not with the basal cell marker *TP63* (see **Table 3** for the reviewer below). This suggests that *HOXA13* may be linked to a Barrett's gene signature and distal phenotype, and it is interesting to note that this is similar for both *HOXA13*-expressing cells from the normal esophagus and from BE. A small population of cells co-expressing *KRT14* (a gene pair with *KRT5*), *TP63* and *KRT7* is seen in the normal esophagus proximal from junctions (n=19), however, these cells do not express *HOXA13*. The percentage of cells with this gene signature appears to be reduced in BE. Thus, these data suggest that the *HOXA13*⁺ compartment is not the same as the earlier suggested *KRT7*⁺*KRT14*⁺*TP63*⁺ cell of BE origin. Of note, *KRT7*⁺*KRT14*⁺*TP63*⁺ cells gave rise to BE-like

epithelium only upon ectopic expression of *CDX2* [4]. More details regarding the *HOXA13*-positive population can be found in the revised manuscript which was amended to reflect the new data obtained from this analysis (new **Figure 6**).

Table 4 for the reviewer 2. The total number of cells analysed from the normal esophagus (N=886) and BE (N=395) were analysed for expression of *HOXA13*, *TP63*, *KRT13* and *KRT7*. The absolute number of cells expressing these genes as well as the percentage of cells within each tissue are shown. The last column indicates the percentage overlap (e.g. 2.7% of *HOXA13*+ cells of the normal esophagus express p63).

Genes	N of cells in healthy esophagus (from 886)	% of total in healthy esophagus	N of cells in BE (from 395)	% of total in BE	% of HOXA13 ⁺ in healthy esophagus	% of HOXA13 ⁺ in BE
HOXA13	37	4.18	132	33.42		
p63	659	74.38	12	3.04		
KRT14	567	64.00	51	12.91		
KRT7	104	11.74	285	72.15		
p63, KRT7	17	1.92	10	2.53		
p63, KRT14	479	54.06	3	0.76		
KRT7, KRT14	37	4.18	27	6.84		
HOXA13, p63	1	0.11	4	1.01	2.70	3.03
HOXA13, KRT14	4	0.45	15	3.80	10.81	11.36
HOXA13, KRT7	32	3.61	105	26.58	86.49	79.55
HOXA13, p63, KRT7	0	0.00	2	0.51	0.00	1.52
HOXA13, p63, KRT14	0	0.00	1	0.25	0.00	0.76
HOXA13, KRT7, KRT14	4	0.45	13	3.29	10.81	9.85
KRT7, KRT14, TP63	19	2.14	2	0.51		
HOXA13, KRT7, KRT14, TP63	0		0			

2) *The authors hypothesize that the HOXA13 positive cells might give rise to Barrett's in the context of inflammation from reflux. Did they examine for evidence of reflux-induced inflammation in the individuals that they sampled? It would be very feasible to sample the GEJ in patients with reflux esophagitis to see if the HOXA13 population is expanded or nearer to the surface.*

Reply: Indeed, this is a very interesting idea. In our study, GEJs were taken from surgical specimens, which may have been exposed to reflux in varying degrees. Two strips came from surgical specimens without evidence of BE, one with a neuroendocrine tumor and the other decompensated achalasia, the third patient had surgery to remove an EAC. The proximal to distal location in the GI tract can be established quite accurately in these samples. The location of junctions were confirmed by expert pathologist. Therefore, we can claim these

cells are located in the glandular epithelium and that these cells are located in close proximity to the GEJ. However, we make no statements on how close to the surface these cells are located as this can be difficult to tell given the possibility of tangential cutting of the tissue. Using forceps biopsies to obtain samples from patients with reflux esophagitis could easily lead to missing the small area where positive cells are located, which is why we considered this to be an unsuitable option.

We checked publicly available RNA expression data of subjects with and without eosinophilic esophagitis (GSE113341), however, *HOXA13* expression was not detected, likely because of the scarcity of this cell population. No single cell RNAseq data are as yet available for this disease.

Taking into account these technical and logistical limitations, we employed the EPC2 cell line with forced *HOXA13* overexpression to address the reviewer's question. As is now shown in new **Figure 7F** in the manuscript, *HOXA13*-overexpressing cells are less sensitive to bile/acid exposure. Data from literature suggests that this may not only be specific to bile/acid treatment, as *HOXA13* also makes ESCC cells more resistant to chemotherapy [5]. Together, these data suggest that *HOXA13* indeed makes cells less susceptible to abrasive treatments.

3) For the embryonic data it should be noted that human oesophagus undergoes transition from columnar to squamous phenotype at 4-5 months of development. Therefore, the authors should confirm that the images in figure 4J show a fully mature squamo-columnar junction, not one that is undergoing transition to a squamous phenotype (please note the dilated nature of glands in images J1 and J3). It would be fascinating to have longitudinal data over this 4-5 month transition period though clearly this may not be feasible.

Reply: The reviewer makes an excellent point. Indeed, the illustrations are derived from human fetal tissue at 17 weeks of age, at which stage the transformation of the columnar to the squamous morphology of the stratified esophagus will not be complete. We have now employed 7-layer Nanozoomer scanning to obtain images of better quality, allowing a more detailed analysis (new **Figure 5C, Extended data fig 8**). While all GE junctions are highly positive for *HOXA13*, we observe some positivity also in the esophageal epithelium, which is not seen in the fully mature epithelium of adults (new **Figure 5C**). One of our samples was 20 weeks of age, but positivity was still seen in the esophagus. Unfortunately, longitudinal data over the whole transition period could not be obtained due to the restricted availability of human embryonic tissue, as foreseen by the reviewer. Thus, our data indicate that *HOXA13*-positive cells are widely present during embryogenesis in the upper GI tract in non-squamous and non-mature epithelial structures, while only a small fraction of *HOXA13*-positive cells remain in the adult esophagus, but increase in BE, and we have now adjusted our manuscript accordingly.

4) There is a lack of direct evidence presented that HOXA13 cells are activated upon tissue damage. A DTA-Krt14Cre HOXA13-GFP mouse model (Wang Cell 2011) would be helpful to observe the expansion of the HOXA13 cells.

Reply: As mentioned above, we now include novel data describing that *HOXA13*-overexpressing cells are less sensitive to bile/acid treatment. Data from literature suggests that *HOXA13* also confers protection against other damaging factors, as for instance, *HOXA13* protects ESCC cells from chemotherapy [5]. BE can be considered as a wound healing process initiated by esophageal injury from GERD [6]. Data from literature support idea that *HOXA13* is involved in processes related to wound healing and regeneration after tissue damage of zebrafish bony fin ray [7], *Xenopus* limb and tail [8], spermatogenic cells of mice

and cell lines in response to UV irradiation [9]. Furthermore, *HOXA13* promotes wound healing of *in vitro* gastric cancer cell cultures [10]. Thus, our new data is in line with other studies suggesting a role for *HOXA13* resistance to stressors and wound healing.

5) *The nature of the BAR-T cells is intriguing as the authors suggest they might undergo both squamous and columnar development in in vivo culture. A) The origin of this phenotype should be addressed as BAR-T cells might be a mixture of the squamous and BE cell lines and HOXA13 manipulations presented in figure 6E might result from changes in relative number of cells or complete depletion of the columnar phenotype from the knock-out experiments as they were formed from expanded single cell clones. The comparison could be performed by KRT7 or 8/18 (for BE) and KRT13 (for squamous oesophagus) staining.*

Reply: We understand the reviewer's concern regarding the possible bias that could be introduced by using a heterogeneous cell line. However, there is no evidence in the literature that BAR-T is a mixed population of BE and normal squamous cells. In fact, this cell line was described to be established from a single cell colony [11]. While mixed expression of columnar and squamous markers in this line has been described, single cell analysis was not performed, and this likely reflects the bilinear differentiation potential of BAR-T cells rather than heterogeneity from the original culture [12]. Before performing our experiments, we first tested these cells by FACS and found that co-expression of CK14 and CK7 occurs in nearly 100% of BAR-T cells, suggesting relative homogeneity under normal culture conditions (**Figure 1** for the reviewer below).

Figure 2 for the reviewer 2. Staining of BAR-T [11] cells for CK7 and CK14. Cells were fixed in 2% paraformaldehyde for 10 min at room temperature, permeabilised with ice cold methanol for 10min at 4C and blocked in 5% bovine serum albumin in 0.05% Tween-20 in tris-buffered saline. Then cells were incubated with mouse anti-CK7 (1:1000, clone OV-TL 12/30 Dako cat#M7018) and guinea pig anti-CK14 (1:600, Progen Biotechnik, cat#GP-CK14). Donkey anti-mouse IgGAlexaFluor647 (1:2000, Molecular Probes, Cat# A3157) and goat anti-guinea pig IgG AlexaFluor488 (1:1500, Molecular Probes, Cat# A11073) were used as secondary antibodies. Cells were analysed by FACSCanto II (BD Biosciences).

More importantly, to verify the potential of individual BAR-T cells to differentiate into two lineages, we analyzed 10 single cell colonies derived from parental BAR-T cells, all of which showed commitment to two different lineages, squamous and columnar. This data is depicted in **extended data Figure 9** (the rat trachea epithelial reconstitution model), and proves that the capability to form two independent lineages is an inherent feature of individual BAR-T cells rather than a representation of a heterogeneous cell population. For our KO experiments in BAR-T cells, single cell colonies were derived upon gene editing by Crip Cas, and thus data do indeed represent individual clones, derived from individual cells. The reviewer's concern that we may have selected a clone that lacks the bi-differential capacity seems unfounded, as quantification of the epithelial lining shows that the capacity of BAR-T cells to form columnar epithelium is diminished, but not completely lost upon knockout of *HOXA13* (now presented in new **Figure 8C**). This suggest that *HOXA13* tips the differentiation decision scale of a cell that is inherently able to switch to both lineages.

B) In addition, the non-transfected controls and time courses should be shown for comparison – the changes in the phenotype were not clear to me.

Reply: We apologise for the confusion. Non-transfected controls were presented in the **extended data Figure 9**. We have now mentioned it additionally in the results section. To clarify the changes in phenotype upon *HOXA13* KO, we performed PAS staining and anti-involucrin staining of samples from the rat trachea reconstitution model in addition to the H&E staining presented before (new **Figure 8**). Furthermore, we performed quantitative analysis of the differentiation potential by measuring the length of columnar-like epithelium in WT and KO samples (shown in new **Figure 8C**). Regarding the time course, we had samples from this model from time point 4 weeks and time point 6 weeks, but the epithelium was very damaged from the last time point and we lost most samples. Thus, quantitative analyses were only performed for the week 4 samples. We concluded that *HOXA13* KO reduces the presence of epithelia with a columnar-like phenotype.

6) The authors claim that the low level up-regulation of HOXA13 following exposure to acid and bile, is due to a selective advantage of cells with high expression of HOXA13. This conclusion is questionable and should be substantiated showing viability of cells under treatment with acid and bile compared to cells transfected with HOXA13.

Reply: We thank the reviewer for this comment. We have treated EPC2 cells with bile/acid and found that indeed *HOXA13* provides a selective advantage to cells under bile/acid treatment. This new data has been included in the manuscript in new **Figure 7F**.

7) The claims about the oncogenic potential of these cells is over-stated given the limited data shown. The authors should restrict the discussion to the induction of metaplasia in keeping with the data shown.

Reply: Indeed, the reviewer is right to state that the role of *HOXA13* in oncogenesis may have been somewhat speculative in our previous version of the manuscript. However, using qPCR we found that *HOXA13* mRNA levels are increased in EAC (**Figure 1B, C, D**) indicative of a capacity of these cells to proliferate and outcompete other cells. In addition, our new data suggest that *HOXA13* provides proliferative advantage in our cell line models and activates cancer-related Notch signaling. *HOXA13* has previously been shown to be involved in ESCC [13] and other types of cancer [10, 14-16]. Thus, there is strong evidence for a potential role of *HOXA13* in EAC as well. Nevertheless, in keeping with the fact that our manuscript focuses mainly on metaplastic tissues, we have now limited our discussion of the oncogenic role of *HOXA13*.

Text inaccuracies and other comments:

I found it very challenging to follow the figures while reading the text and in fact on some occasions the authors appears to refer to wrong figures. Often the experiments being conducted were not clear from the text and could only be understood from the figure legends.

Reply: We sincerely apologize for mistakes which found their way into our previous manuscript. We have extensively rewritten the paper, and we hope that you will find this current version much easier to read. We hope to have removed all errors.

Examples of inaccuracies are:

- on page 4 “We observed a decrease in 5-methylcytosine and an increase in 5-90 hydroxymethylcytosine in the promoters of *HOXA13* and *HOTTIP* (extended data fig. 5)”. I cannot find these data in the figure

Reply: indeed this reference was to data removed from the manuscript at a late stage in preparation and was not correct. Thus, statement has been removed from the manuscript.

- or when again they refer to extended Fig 5 in the legend of Fig 2. I think this is wrong.

Reply: we apologise, indeed, this should have referred to extended data Figure 3. It is corrected now and refers to the new Figure 1.

- I also find not very helpful to keep referring to other figures in figure legends.

Reply: We have now omitted the references to other figures in the figure legends, unless otherwise required by journal guidelines.

-The last paragraph in page 3 and the second paragraph in page 4 are repetitive, and present the same concept.

Reply: The publicly available data sets on which these data are based came from two separate studies from the same research group, with a similar methodology, which is why there was overlap. The text is corrected now to make this more clear.

-It is confusing that the authors refer to different reference in relation to the use of publicly available data (ref 15 and 18). In relation to this Figure 2 should mention that the data presented refer to publicly available data.

Reply: These publicly available data sets came from the same group, but were published in two different studies, which may have been confusing. We have amended the manuscript to make this clear.

-In the introduction the authors say that gastric inlet patches have high risk for adenocarcinoma. I disagree with this. Malignant transformation of inlet patch is anecdotal and at odds with the relatively high incidence of inlet patches in the general population.

Reply: We agree they do not have an absolute high risk of transformation. However, data from Orosey et al.[17] supports our statements that the inlet patch is a relatively high risk areas for adenocarcinoma. We have now changed this statement in the introduction to be more specific.

- Fig 3A. the bottom ISH inserts are not necessary and should be removed. They are the same as in the lower panel and I find inappropriate that the authors placed black marks on the ish dots to make them more visible.

Reply: Figure has been amended.

- What is the functional explanation of the crypt clonality and the epithelial mesenchymal pattern of expression of *HOXA13* in the murine colon? I find this hard to interpret.

Reply: These data suggests that one single *HOXA13*-positive cell may grow out to populate a crypt, while other crypts start life without such a *HOXA13* positive cell being present. This finding was striking to us, as they show that spatial localization of *HOXA13* is regulated to the level of individual crypts, which has not been shown before.

- *Figure 5 was difficult to follow and contains more explanatory cartoons than actual data.*

Reply: We had hoped that these cartoons would facilitate the interpretation of our complex data. We agree that it might be redundant to present both cartoons. We have now opted to show one, but if the reviewer feels it does nothing to improve the readability of the manuscript we are of course willing to omit it entirely.

- *Fig 6 F. The difference between left and right is not clear. In addition, what do the arrows represent?*

Reply: We apologize for not being clear. We aimed to make following points with this figure: 1) HE staining shows more layers of cells in the right-hand figure, i.e. the BAR-T *HOXA13* expressing wild type cells. 2) PAS stains polysaccharide molecules which make up mucous, and are indicative of goblet-like cells. The arrows point to PAS positivity, which is present in the right panel but not in the left panel, where the BAR-T *HOXA13* KO cells are shown. 3) Involucrin staining is strong in morphologically squamous cells in the left hand panel and weaker in the *HOXA13*-positive epithelium to the right. In the current manuscript we have added new data to this figure, including a quantitative analysis. The figure legend was amended to better explain the results.

-*Lines 106-125 (page 4) discuss four subpanels of figure 3 (A-D) however the figure only has 3 panel with figure legend missing information about the images in the bottom left corner.*

Reply: we apologize for this confusion caused by faulty labeling. It is corrected now.

-*Figure 5C: what is the meaning of DE cells on the graph.*

Reply: DE means definitive endoderm, this was corrected.

Reference list:

1. Di-Poi, N., et al., *Additive and global functions of HoxA cluster genes in mesoderm derivatives*. Developmental Biology, 2010. **341**(2): p. 488-498.
2. Condie, B.G. and M.R. Capecchi, *Mice with targeted disruptions in the paralogous genes *hoxa-3* and *hoxd-3* reveal synergistic interactions*. Nature, 1994. **370**(6487): p. 304-307.
3. Yamamoto, S., et al., *Hoxa13 regulates expression of common Hox target genes involved in cartilage development to coordinate the expansion of the autopodal anlage*. Dev Growth Differ, 2019. **61**(3): p. 228-251.
4. Jiang, M., et al., *Transitional basal cells at the squamous-columnar junction generate Barrett's oesophagus*. Nature, 2017. **550**(7677): p. 529-533.
5. Shi, Q., et al., *Downregulation of HOXA13 sensitizes human esophageal squamous cell carcinoma to chemotherapy*. Thorac Cancer, 2018. **9**(7): p. 836-846.
6. Que, J., et al., *Pathogenesis and Cells of Origin of Barrett's Esophagus*. Gastroenterology, 2019. **157**(2): p. 349-364.e1.

7. Geraudie, J. and V. Borday Birraux, *Posterior hoxa genes expression during zebrafish bony fin ray development and regeneration suggests their involvement in scleroblast differentiation*. Dev Genes Evol, 2003. **213**(4): p. 182-6.
8. Christen, B., et al., *Regeneration-specific expression pattern of three posterior Hox genes*. Dev Dyn, 2003. **226**(2): p. 349-55.
9. Liang, M. and K. Hu, *Involvement of lncRNA-HOTTIP in the Repair of Ultraviolet Light-Induced DNA Damage in Spermatogenic Cells*. Molecules and Cells, 2019. **42**(11): p. 794-803.
10. Qin, Z., et al., *Elevated HOXA13 expression promotes the proliferation and metastasis of gastric cancer partly via activating Erk1/2*. OncoTargets and therapy, 2019. **12**: p. 1803-1813.
11. Jaiswal, K.R., et al., *Characterization of telomerase-immortalized, non-neoplastic, human Barrett's cell line (BAR-T)*. Dis Esophagus, 2007. **20**(3): p. 256-64.
12. Bajpai, M., et al., *Repeated exposure to acid and bile selectively induces colonic phenotype expression in a heterogeneous Barrett's epithelial cell line*. Lab Invest, 2008. **88**(6): p. 643-51.
13. Gu, Z.D., et al., *HOXA13 promotes cancer cell growth and predicts poor survival of patients with esophageal squamous cell carcinoma*. Cancer Res, 2009. **69**(12): p. 4969-73.
14. Deng, Y., et al., *The expression of HOXA13 in lung adenocarcinoma and its clinical significance: A study based on The Cancer Genome Atlas, Oncomine and reverse transcription-quantitative polymerase chain reaction*. Oncol Lett, 2018. **15**(6): p. 8556-8572.
15. Quagliata, L., et al., *High expression of HOXA13 correlates with poorly differentiated hepatocellular carcinomas and modulates sorafenib response in in vitro models*. Lab Invest, 2018. **98**(1): p. 95-105.
16. Dong, Y., et al., *HOXA13 is associated with unfavorable survival and acts as a novel oncogene in prostate carcinoma*. Future Oncol, 2017. **13**(17): p. 1505-1516.
17. Orosey, M., M. Amin, and M.S. Cappell, *A 14-Year Study of 398 Esophageal Adenocarcinomas Diagnosed Among 156,256 EGDs Performed at Two Large Hospitals: An Inlet Patch Is Proposed as a Significant Risk Factor for Proximal Esophageal Adenocarcinoma*. Dig Dis Sci, 2017.

Reviewer #3 (Remarks to the Author):

The pathogenesis of Barrett's esophagus and heterotopias are of clear clinical relevance, and a diverse set of original experimental evidence is presented here to claim that HOXA13 expression in derivatives of the normal foregut is a major factor in this pathology. Besides the clinical relevance a theoretical innovation is implicit in this work which goes against the consensus based on many years of investigation in many laboratories. To establish an important exception to the collinearity view of Hox gene expression in the gastro-intestinal system, exceptionally well documented evidence is necessary. I am afraid that the evidence provided in this manuscript does not reach the requisite quality of documentation. Describing the adult expression domains of developmental control genes like Hox genes is a major challenge, given that these genes typically retain well detectable steady state transcript levels only in rare contexts. Antibody selectivity is very rarely established to be exclusive for a single gene product, due to functionally important regions of amino acid homology, therefore frequent epitope sharing by immunogens. This is especially limiting in case of Hox/HOX genes like the 4, 9 and 13 paralogous groups, where four gene products must be discriminated. The possibility of using GFP and Cre knock-in alleles in the case of mouse Hoxa13 is a precious opportunity to perform very high quality analysis, including organ, tissue, cellular and sub-cellular levels. The compilation of diverse analyses presented in Figures 3 and 4 makes progress in this direction, but in my view the results remain preliminary at best.

Reply: We agree with the reviewer that studying *HOX* genes is technically challenging and has limitations such as absence of selective antibodies. In our view, this is what makes our data combining the GFP-HOXA13 Cre mouse model, RNA scope, and new single cell RNA seq analysis, a valuable contribution to the field. In this paper, we applied a variety of different technical approaches (qPCR, RNA scope, dataset analysis) to provide evidence on the role of *HOXA13* in BE and obtained consistent results. Below, we hope to have addressed all specific concerns of the reviewer regarding technical approaches in each case. While our data may appear to go against the consensus based on many years of investigation in many laboratories, techniques such as single cell sequencing have not been applied to this question before, and our newly added analysis of recently published in Nature Communications RNAseq databases [1] now shows clear evidence of a *HOXA13*-positive compartment in the normal esophagus, which is increased in BE. In addition, most studies investigated *HOX* genes during embryonic development, rather than adult human gut. Other data indicating interruption of collinearity in BE [2] and in the murine embryonic stomach and duodenum [3] have already been reported. Thus, we feel our data are robust and, while challenging the prevailing dogma, provide crucial information which is important to the field.

1/As far as collinearity is concerned, this manuscript presents an extensive investigation of distinct anatomical regions of both human and mouse adult GI in nine segments from esophagus to rectum (Figure 1 and extended Figure 1). In general, it is remarkable how well the overall relative adult expression patterns conform to the collinear generalization derived from fetal mouse samples. Some notable exceptions seem to occur, as for instance mouse Hoxd8, which scored highest in esophagus as opposed to more caudal regions. It is likely to involve some artefact however, because this observation was not reproduced in human specimens. The observations of pathological specimens likely to represent genuine HOXA13 gain of expression (Figure 2 and 3). Overall this part of the work and the modulation of HOXA13 (see comment 8) in functional assays are the strong claims that justify publication.

Reply: We thank reviewer for their positive comments. To verify our results for *HOXD8*, we have now performed sequencing of the qPCR amplification product derived with either mouse and human PCR primer sets. Blast analysis of the sequenced amplicon only matched to mouse and human *HOXD8* transcripts, for mouse and human primers respectively (blast results shown in **Figure 1 for reviewer 3** below). Thus, the primers are selective for their intended target, and therefore differences found between mouse and human samples in our study really appear to be a reflection of underlying differences in expression.

Figure 1 for the reviewer 3. Blast analysis of the sequenced qPCR amplification product for mouse and human confirmed *HOXD8* transcript.

2/An important element of the pathological mechanism envisaged by the current manuscript is the role of *Hoxa13* expressing cells in the pre-morbid esophagus in adult patients. The argument is based on the esophagus and Colon sample comparisons. In Figure 3A we see that qPCR analysis of *Hoxa13* in GIP and Colon give rather similar expression levels, certainly not an order of magnitude difference is visible here. Below the "Esophagus" subtitle RNAscope histological photomicrographs are depicted: hardly anything is visible in GIP in the form of very fine brown granules, while in Colon the brown reaction product is easily apparent due to the much larger diameter of the granules. Apparently, the same histological panels are also shown under the qPCR graph, but there their overall RNAscope signal load appears quite similar, due to a "manual black dot correction". In other words, similar RNA

level with qPCR corresponds to orders of magnitude more robust RNAscope signal in case of Colon, which than manually became homogenized to give the visual impression that in esophagus/GIP the level of HOXA13 expression is comparable to Colon. The poor correspondence of qPCR and RNAscope signals is also indicated by the comparison of finding very high qPCR expression associated with very weak RNAscope signal in MD gastric heterotopia, compared to the order of magnitude lower qPCR signal associated to comparatively robust RNAscope signal in case of Colon. All this is complicated to explain, if at all possible. In any case, I find it inadmissible that RNAscope signal be shown as black dots added to photomicrographs, that effectively hide the actual experimental result. If the images are not good enough to be shown in a technical publication, like Nature Communications, they should not be taken as evidence of any claim. These considerations together with the analysis of Figure 4 below cast serious doubt on the claim that there exists a HOXA13 expressing band of cells in normal adult human esophagus.

Reply: We apologize that the quality of some presented RNA scope images was not sufficient. We have now performed 7-layer nanozoomer scanning for the mouse and human resection specimens of the normal esophagus to enhance visualization of the RNAscope signal. From reviewer's statement "hardly anything is visible in GIP in the form of very fine brown granules, while in Colon the brown reaction product is easily apparent due to the much larger diameter of the granules" we infer that the reviewer takes into the account the intensity/size of signal when comparing colon signal to GIP or MD gastric heterotopia. To clarify, one dot represents a single RNA target regardless of intensity or size (<https://acdbio.com/science/how-it-works>), although some larger dots can be explained by clusters resulting from overlapping signals from multiple mRNA molecules. Thus, we felt that indicating the presence of dots by manual correction was allowable, as it was merely meant to show the location of individual dots. Alternatively we could have added arrows but we felt that would have cluttered the figure. We would like to emphasize that the manual correction was clearly stated in the figure legend and there was no intent to hide the actual RNA scope signal. However, we have now removed the images containing these manual indications from the manuscript, and show only the larger, unenhanced images, where the RNAseq signal can be better appreciated. These RNAseq examples complement the qPCR data shown in **Figure 1D**.

3/Figure 4 A-F and H2 of this manuscript document observations of the Hoxa13GFP knock-in heterozygous adult mouse colon. The overall analysis of the authors may perhaps be accurate, but from these panels the only fluorescent image that is convincing is shown in panel E. There the green fluorescent signal is clearly restricted to the epithelium, while the underlying mesoderm is free of fluorescence. The strong signal in the columnar intestinal epithelium is highlighted, and in the squamous anal epithelium is considered as negative. Yet, unexpectedly, the strongly marked columnar epithelial cells show signal both in nucleus and in cytoplasm. Given that Hoxa13GFP is presumed to be localized to the nucleus, the signal in the cytoplasm needs to be explained. In addition, the image on panel A suggests continued fluorescent domain into the anal squamous epithelium, in contrast to the author's claim, and a more permissive scoring of the high power image in panel E seem to confirm expression in both columnar rectal and squamous anal epithelia.

Reply: To further substantiate our results we have now stained these samples with anti-GFP antibodies. The results from immunohistochemistry are similar to the previously observed fluorescent GFP signal in the mouse intestine. The signal observed in mesoderm was also

confirmed with IHC. We now include images of three GFP positive mice and one GFP negative mouse in our manuscript, as well as immunofluorescence data.

With respect to the intracellular localization of Hoxa13, most of the signal can be found in the nucleus and some signal in the cytoplasm, which was also confirmed by immunohistochemistry. The cytoplasmic signal could be explained by synthesis of the protein by ribosomes located in the cytosol. Many transcription factors are known to shuttle in and out of the nucleus depending on intracellular/extracellular signals as a means to achieve transcriptional regulation [4-6]. Additionally, physiological degradation of Hoxa13 in proteasomes as part of protein turnover may contribute to the signal seen in the cytoplasm. Active shuttling to and from the cytoplasm to regulate function of HOX proteins has been described as well [7].

Regarding the fluorescent signal in the anal squamous epithelium, we apologize for the confusion. The 'swiss roll' overview depicted in **Figure 3A** does not include anal squamous epithelium, thus appearing to have continuous staining. The high power magnification image in panel E was taken from another heterozygous mouse.

4/An intriguing claim of the authors' is crypt-clonal expression of Hoxa13GFP in the columnar epithelium, Figure 4D. Such mosaic character was not expected from public fetal http://www.eurexpress.org/ee/databases/showAssayImgs.jsp?image=euxassay_000981_01.jpg epithelial in descending colon), and this phenomenon in itself would merit an in-depth study. However, since in this case the observation arises in sub-optimal methodical context, the result should be considered preliminary, at best.

Reply: This phenomenon is now confirmed also with anti-GFP IHC (**Figure 3B**). The expression of Hoxa13GFP in the columnar epithelium of the colon is continuously positive or negative for the majority of the length of the colon. However, at the expression border it does appear crypt clonal in two out of three mice. As investigation of this phenomena is out of scope of the present manuscript, we have amended text stating the preliminary nature of this observation.

5/ The most relevant piece of evidence concerning adult mice, in panel H2, shows the presence of seven highly fluorescent cellular profiles at the squamous glandular epithelial junction in stomach. At least the first of these profiles is apparently not in the epithelial layer, and the last two show a clear hole at the central position, that suggests cytoplasmic, rather than the expected nuclear localization of the signal. Both these details cast doubt on the presence of Hoxa13GFP fusion protein in a patch of epithelial cells in the adult mouse gastro-esophageal junction. In fluorescence imaging a source of potential artefacts may arise from the use of frozen samples. Observing GFP fluorescence after freezing, storage, thawing and cryo-sectioning compromise cellular integrity, thus the consistent evaluation of tissue distribution. Typically, such observations are made on fresh non-fixed tissues. When cryo-sections are imaged, the Hoxa13GFP signal still should remain predominantly nuclear (see for instance in case of Hoxa13GFP: Development 2001 128: 4177-4188, Fig 8G,H). At this stage I remain dubious as to the reliability of fluorescent imaging in this set of analyses.

Reply: We appreciate the reviewer's concerns and have now further characterized the individual *HOXA13*-positive cells we first described by immunofluorescence. First, we have performed immunohistochemistry with an anti-GFP antibody, precluding any potential artifacts due to autofluorescence. The IHC analysis is in line with our previously observed immunofluorescence analysis, but allows a better localization of the signal due to counterstaining with hematoxylin and high resolution scanning by Nanozoomer. This analysis

shows the presence of a very small fraction of single Hoxa13-expressing cells along basal side of the mouse stomach epithelium starting from the GEJ. These new data are added to the manuscript (new **Figure 5** and **extended data Figure 7**). In addition to representative images in the main manuscript, we have also attached images of all mice tested in the supplement.

6/ A further issue of assay reliability concerns Figure 4 J1,2,3. In all cases a robust ISH signal is expected to be cytoplasmic, and fairly homogenous over the cytoplasm. The positive signal claimed here is present as small puncta over many cells, occasionally present over nuclear profiles, J3. This complicates the evaluation of J3 near human fetal gastro-esophageal junction, especially since no concordant Hoxa13GFP expression was reported here or elsewhere at this region in fetal mice, and the distribution is also different from the rare cell profiles seen in adult human gastro-esophageal junction (Figure 4I, see below). Regrettably, despite the accessibility of the material, the authors report no effort to detect Hoxa13GFP in developing mouse stomach, although they claim HOXA13 transcript accumulation at the respective-duodenal epithelial transition in fetal human samples.

Reply: We regret that the reviewer has doubts regarding the specificity of our RNAscope data. The probes themselves are extensively verified for their target [3]. RNAscope detects mRNA which is synthesized in nuclei during transcription and transported to the cytoplasm. Thus, this readily explains the localization of signal. However, it should be noted that in RNAscope, the exact cellular localization is difficult to pinpoint due to the lack of counterstaining in this technique, and in addition cytoplasmic signal can appear nuclear because of presence to cytoplasm on top of nuclei. Hence, we make no statement as to the intracellular localization based on RNAscope, and use these data only to regionalize the signal in our tissues and provide a semiquantitative analysis of *HOXA13* in tissues which does not require homogenization of tissue (as is the case for qPCR). We now provide scans of samples of 3 different embryos to demonstrate the robustness of this result and that it is not an artefact presented in one sample. The same probe was used on colon and ileum, as negative and positive controls, respectively, and with expected results (**Figure 1D**). The results we obtained with RNAscope do not stand alone, but validate our data obtained from immunofluorescence/immunohistochemistry analysis of HoxA13 in mice and qPCR/ single cell analyses (newly added data) in human.

7/ The truly relevant adult human esophageal region is documented in panel Figure 4I. Here seven foci of brown labelling are shown, perhaps four of which should include sections of nuclei. The prediction again is that nucleoplasm is free of homogenous ISH signal. In all four of these profiles the brown signal seems to cover the entire cell homogenously, raising the possibility of artefacts. Taken as the authors intend, it again may at best be considered as a preliminary indication of HOXA13 expression near adult human gastro-esophageal junction. This type of analysis should be extended to more specimens to provide solid evidence of in situ HOXA13 mRNA accumulation in normal adult esophagus, which is the most controversial information concerning the etiology of Barrett's esophagus pathology in this manuscript. In summary of imaging studies, it seems plausible that RNAscope findings in adult normal esophagus, and in adult mouse transitional epithelium are a reflection of variations in assay conditions, giving variable noise on a backdrop of essentially negative Hoxa13 expression status (<https://gtexportal.org/home/gene/HOXA13>).

Reply: To address reviewers concerns we have scanned all our analyzed samples and present them in the supplement to the manuscript to ensure that results described may be appreciated

for all samples. In these specimens the exact location of signal in relation to histological features can be seen. The reviewer refers to the gtxportal to demonstrate that the tissues we are investigating are devoid of *HOXA13* – but this exactly pertains to the point we are trying to make: single *HOXA13*-positive cells are present in the upper GI epithelium, but due to their low numbers their combined *HOXA13* signal is not high enough to be picked up by qPCR of homogenized tissues (as also shown by our qPCR experiments on the collinearity of *HOX* genes along the entire GI tract in the **Figure 1**, as well as databases such as gtxportal). What the reviewer describes as “Variable noise” is a signal that is seen at the GEJ area of three separate mice and three separated humans. It is unlikely that artifacts would be present in the exact same location in three independent slides. Furthermore, these data are now corroborated by immunohistochemistry for mice and analysis of single cell RNAseq data for man. We do see some differences between species and developmental stages, as are to be expected, and which have now been better described in the manuscript.

8/Figures 5 and 6 document ingenious in vitro and organotypic tissue reconstruction functional assays involving HOXA13 overexpression and CrisprCas9 mediated loss of function in diverse cell lines. In aggregate, the results suggest that HOXA13 contributes to control of epithelial cell proliferation and differentiation in ways consistent with a role in pathogenesis of intestinal metaplasia in the GI system.

Reply: We thank the reviewer for their positive comment.

In conclusion normal human fetal, human adult and mouse adult esophageal-gastric epithelial junctions seem to show positive Hoxa13 expression signals reported in this manuscript, but the patterns do not really correspond, and are never truly convincing in any of the three cases, while the fetal mouse corresponding region is either negative or non-analyzed. This combined evidence does not lend strong support to the presence of a Hoxa13/HOXA13 positive region at the squamous-glandular esophageal-gastric epithelial transition region (see Extended data Figure7). It is intriguing, nevertheless, that a previous study of ten independent specimens of Barrett's esophagus did not indicate any overexpression of Hoxa13, as compared to normal esophagus (10.1073/pnas.1116933109) and inversely, the current study does find an overexpression of the same HOXB genes in specimens from thirteen independent patients with similar pathology. In the previous study midsection HOXB genes, in this one HOXA13 are pinpointed. Perhaps both or neither are to be taken at face value. Furthermore, it is instructive to consider the observation of hamartomas containing both squamous and gastric glandular epithelium in stomach of transgenic mice overexpressing a midsection HoxC gene ([https://doi.org/10.1016/0092-8674\(92\)90388-S](https://doi.org/10.1016/0092-8674(92)90388-S)). Although very artificial, that experiment really raised the possibility that forced massive expression of a HOX gene may cause foregut epithelial misspecification. If thus may be consistent with the experiments presented here, there is no conclusive evidence that an anterior expression of HOXA13 in normal human esophagus has causal relevance to initiation, maintenance or aggravation of the pathological process in either of the human studies.

Reply: The discrepancy mentioned by the reviewer may be due to the use of probe-based array technology in the PNAS paper, which may not be ideally suited for highly homologous sequences, although we did confirm the earlier-reported findings validated by qPCR regarding upregulation of *HOXB6* and 7 in BE. Indeed, data of [https://doi.org/10.1016/0092-8674\(92\)90388-S](https://doi.org/10.1016/0092-8674(92)90388-S) are consistent with our observations. Furthermore, we now provide an analysis of single cell RNAseq data which confirms our data, and suggest that enhanced

HOXA13 mRNA levels observed by qPCR in homogenized tissues may be a result of upregulation of the number of *HOXA13*-positive cells in these tissues rather than the expression levels per cell. Our rationale for picking *HOXA13* as target for further in depth analysis was that its absolute overexpression is the largest and its 5' nature would make it a prime candidate for conveying posterior characteristics. We have now explained this more clearly in the manuscript. However, we do not exclude a role of other *HOX* genes and have mentioned them in the discussion as suggestion for future research. Overall, we appreciate the concerns raised by the reviewer and acknowledge that the quality of some of the images in our previous version of the manuscript may have hampered interpretation of the results. The current manuscript shows better quality images, more representative examples, immunohistochemical analysis of tissues as well as analysis of RNAseq data. Furthermore, the manuscript has been extensively rewritten to allow a better interpretation of the findings. With these alterations, we hope to have alleviated the reviewer's concerns.

Minor points:

The reference given to indicate the source of the Hoxa13GFP mouse line, and the methods of its analyses is in error. J. Cell. Biochem. 112: 1022–1034, 2011 does not concern that mouse line in any way.

Reply: We apologize for our mistake. Reference number 2 refers to “Perez, W. D., Weller, C. R., Shou, S. & Stadler, H. S. Survival of Hoxa13 homozygous mutants reveals a novel role in digit patterning and appendicular skeletal development. *Dev Dyn* 239, 446-457 (2010).” We carefully checked our references using continuous numbering to avoid further confusion.

Extended data Figure 3 do not refer to the HOXA complex, it is difficult to see how it is relevant to this HOXA13 centered manuscript.

Reply: Extended data Figure 3 is related to **Figure 2** which contains the HOXA data. While we first describe collinearity of all *HOX* genes by qPCR, we focused on *HOXA13* for further in depth analysis in terms of location and function. Investigating all *HOX* genes to this extent is beyond the scope of the current manuscript, but the data may be beneficial for further studies into these other *HOX* genes by the scientific community.

The summary figure (see Extended data Figure 7), ends with reference to malignancy, which at this stage seems more a hypothesis than summary of such data issuing from conclusive investigation.

Reply: Indeed, the reviewer is right to state that the role of *HOXA13* in oncogenesis may have been somewhat speculative in our previous version of the manuscript. However, using qPCR we found that *HOXA13* mRNA levels are increased in EAC (**Figure 1C**) and our new data at single cell level confirm that this is due to a rise in *HOXA13*-positive cells in the case of gastric cancer (**Figure 4C, D**), indicative of a capacity of these cells to proliferate and outcompete other cells. In addition, our new data suggest that *HOXA13* provides proliferative advantage in our cell line models and activates cancer-related Notch signaling. *HOXA13* has previously been shown to be involved in ESCC [8] and other types of cancer [9-12]. Thus, there is strong evidence for a potential role of *HOXA13* in EAC as well. Nevertheless, in keeping with the fact that our manuscript focuses mainly on metaplastic tissues, we have now limited our discussion on the oncogenic role of *HOXA13*.

Reference list:

1. Owen, R.P., et al., *Single cell RNA-seq reveals profound transcriptional similarity between Barrett's oesophagus and oesophageal submucosal glands*. Nature Communications, 2018. **9**(1): p. 4261.
2. di Pietro, M., et al., *Evidence for a functional role of epigenetically regulated midcluster HOXB genes in the development of Barrett esophagus*. Proceedings of the National Academy of Sciences of the United States of America, 2012. **109**(23): p. 9077-9082.
3. Kawazoe, Y., et al., *Region-specific gastrointestinal Hox code during murine embryonal gut development*. Dev Growth Differ, 2002. **44**(1): p. 77-84.
4. Tedesco, M., et al., *STRA8 shuttles between nucleus and cytoplasm and displays transcriptional activity*. J Biol Chem, 2009. **284**(51): p. 35781-93.
5. Ernst, S. and G. Muller-Newen, *Nucleocytoplasmic Shuttling of STATs. A Target for Intervention?* Cancers (Basel), 2019. **11**(11).
6. Fu, X., et al., *The Rules and Functions of Nucleocytoplasmic Shuttling Proteins*. Int J Mol Sci, 2018. **19**(5).
7. Deneyer, N., et al., *HOXA2 activity regulation by cytoplasmic relocation, protein stabilization and post-translational modification*. Biochim Biophys Acta Gene Regul Mech, 2019. **1862**(9): p. 194404.
8. Gu, Z.D., et al., *HOXA13 promotes cancer cell growth and predicts poor survival of patients with esophageal squamous cell carcinoma*. Cancer Res, 2009. **69**(12): p. 4969-73.
9. Qin, Z., et al., *Elevated HOXA13 expression promotes the proliferation and metastasis of gastric cancer partly via activating Erk1/2*. OncoTargets and therapy, 2019. **12**: p. 1803-1813.
10. Deng, Y., et al., *The expression of HOXA13 in lung adenocarcinoma and its clinical significance: A study based on The Cancer Genome Atlas, Oncomine and reverse transcription-quantitative polymerase chain reaction*. Oncol Lett, 2018. **15**(6): p. 8556-8572.
11. Quagliata, L., et al., *High expression of HOXA13 correlates with poorly differentiated hepatocellular carcinomas and modulates sorafenib response in in vitro models*. Lab Invest, 2018. **98**(1): p. 95-105.
12. Dong, Y., et al., *HOXA13 is associated with unfavorable survival and acts as a novel oncogene in prostate carcinoma*. Future Oncol, 2017. **13**(17): p. 1505-1516.

Reviewers' Comments:

Reviewer #2:

Remarks to the Author:

General remarks:

The authors have significantly improved on the previous version of the manuscript. The inclusion of all images as supplementary data is very good together with a more accurate description of the results and the figures. Furthermore, the authors have supplemented the manuscript with additional analysis of RNA-seq and single cell RNA-seq data. Finally, they have provided a much cleaner explanation of the results and made it clear that the data present focuses on the role of HOXA13 in the development of BE, regardless of its putative site of origin.

There are two outstanding issues which need to be addressed:

1. The authors note that SMG were not present in the specimen used for RNAscope analysis. However, since SMG can be obtained from samples of the distal esophagus/GEJ region, I would suggest RNAscope staining for HOXA13 of diagnostic FFPE slides (in addition to already isolated specimen). This would confirm (or refute) the single cell RNA-seq-based observation that HOXA13 is expressed in the SMG which is important to clarify in view of the main findings and conclusions from this manuscript.

2. Additional staining of the GEJ region should be performed for KRT7+/KRT5+/TP63+ transitional epithelium. Previous report (DOI: 10.1038/nature24269) suggests that these cells form a small group of cells directly at the GEJ regions. In the current study HOXA13 staining seems to extend beyond the GEJ region into the gastric cardia (although this conclusion is difficult to make due to illegible size bars, see minor comments), suggesting that, in line with single cell RNA-seq data, HOXA13 cells are not KRT7+/KRT5+/TP63+

Minor comments:

1. Figures are not ordered by the order followed in the main text. For example lines 45-107, Figure 1A is followed by 2A after which authors return to figure 1B and 1C, followed by 2B and then return to figure 1D and E. I would suggest to either merge figures 1 and 2 or to reorder the panels to fit text

2. The size of size bars in all images should be either provided in figure legend or on the images.

Reviewer #3:

Remarks to the Author:

The authors constructively dealt with my questions and remarks concerning the previous version of this manuscript. I appreciate their effort in providing clarifications and new data. Given the broad ranging sampling and analyses presented, points of divergent interpretation may not be fully resolved in a peer review process. I admit that the revised manuscript gives important perspectives on the issues concerning Barrett's oesophagus. There seems one perplexing anomaly though, the case of Meckel's diverticulum (Figure 1C, ileum samples). In this case a robust fold change of Hoxa13 gain of expression was found, the largest of all that was measured in this set of analyses. The direction of epithelial metaplasia however was of an anterior rather than posterior character. This seems difficult to fit into the explicative context invoking the overall role of HOXA13 in promoting posterior identity. Jozsef Zakany

Reviewer #4:

Remarks to the Author:

I have read both the revised manuscript and the comments of the authors. I think they have addressed everything satisfactory. I recommend publication of this work.

HOXA13 in etiology and oncogenic potential of Barrett's esophagus

REVIEWER COMMENTS

Reviewer #2 (Remarks to the Author):

General remarks:

The authors have significantly improved on the previous version of the manuscript. The inclusion of all images as supplementary data is very good together with a more accurate description of the results and the figures. Furthermore, the authors have supplemented the manuscript with additional analysis of RNA-seq and single cell RNA-seq data. Finally, they have provided a much cleaner explanation of the results and made it clear that the data present focuses on the role of HOXA13 in the development of BE, regardless of its putative site of origin.

There are two outstanding issues which need to be addressed:

1. The authors note that SMG were not present in the specimen used for RNAscope analysis. However, since SMG can be obtained from samples of the distal esophagus/GEJ region, I would suggest RNAscope staining for HOXA13 of diagnostic FFPE slides (in addition to already isolated specimen). This would confirm (or refute) the single cell RNA-seq-based observation that HOXA13 is expressed in the SMG which is important to clarify in view of the main findings and conclusions from this manuscript.

Reply: We thank the reviewer for the positive comments. We agree that demonstrating SMG positive for HOXA13 is of great importance. We managed to identify one section containing SMG in another block of GEJs in our pathology archive, as confirmed by expert pathologist. The SMG area was indeed positive for *HOXA13* (please see Figure 1 for the reviewer below) which is in line with the single cell RNA sequencing data. The image is now included in the revised version of the manuscript (Figure 4c).

Figure 1 for the reviewer (Figure 4c in revised manuscript). H&E (on the left) and HOXA13 RNA scope (on the right) of GEJ. B, E – squamous epithelium negative for HOXA13, C, D – SMG area positive for HOXA13. Scale bar: upper panel 250 μm , lower panel A 50 μm , B 25 μm .

2. Additional staining of the GEJ region should be performed for $KRT7+/KRT5+/TP63+$ transitional epithelium. Previous report (DOI: 10.1038/nature24269) suggests that these cells form a small group of cells directly at the GEJ regions. In the current study HOXA13 staining seems to extend beyond the GEJ region into the gastric cardia (although this conclusion is difficult to make due to illegible size bars, see minor comments), suggesting that, in line with single cell RNA-seq data, HOXA13 cells are not $KRT7+/KRT5+/TP63+$

Reply: Studying the co-expression of $KRT7+/KRT5+/TP63+$ with HOXA13 is technically challenging due to the absence of selective antibodies for HOXA13. However, in an attempt to address the reviewer's question, we performed triple immunofluorescence staining for $KRT7+/KRT5+/TP63+$ of the 3 GEJs analyzed for HOXA13 expression by RNA scope, which would allow an assessment as to whether or not these triple positive cells are present in the same regions where we identified HOXA13-positive cells. As expected, KRT5 was expressed in the squamous epithelium of the esophagus, while p63 was mainly present in the area where basal esophageal cells reside. Excitingly, we did identify triple positive cells in the GEJ area and SMG, however, the stomach was negative in this respect. While we therefore cannot exclude that these triple positive cells may overlap with HOXA13 in SMG and GEJ, the HOXA13+ cells found in the stomach do not express these markers. It is of interest to note that while the presence of $KRT7+/KRT5+/TP63+$ in the GEJ has been speculated before based on mouse studies, only 4 cells were found by single cell sequencing, and to the best of our knowledge we are the first to actually

show the presence of these triple positive cells in the human SMG by immunofluorescence staining. Thus, we thank the reviewer for their excellent suggestion, which further adds to the discussion on the cell of origin of BE. We have added images of SMG positive for KRT7+/KRT5+/TP63+ cells to the manuscript.

A) Sample 1. Overview. Merged channels. Triple staining for KRT5 (red), KRT7 (green), p63 (white) and nuclei (dapi, blue)

B) Sample 1. Areas of interests

C) Sample 1. Split channels

D) Area A - GEJ

E) GEJ magnified

F) Area B – SMG

G) SMG magnified

H) SMG magnified

I) Area C. Squamous esophagus

J) Area D. Stomach

K) Sample 2. GEJ

L) Sample 2. GEJ magnified

Sample 3. GEJ

Figure 2 for the reviewer. Triple staining for KRT5 (red), KRT7 (green), p63 (white) and nuclei (DAPI, blue) of GEJ.

Minor comments:

1. *Figures are not ordered by the order followed in the main text. For example lines 45-107, Figure 1A is followed by 2A after which authors return to figure 1B and 1C, followed by 2B and then return to figure 1D and E. I would suggest to either merge figures 1 and 2 or to reorder the panels to fit text*

Reply: Thank you for the comment. Indeed the number and nature of figures made it difficult to follow the order the text, in particular with the supplemental data, but we have now tried to make this more consistent by combining Figure 1 and 2.

2. *The size of size bars in all images should be either provided in figure legend or on the images.*

Reply: we apologize for the omission. The size of size bars is now indicated in the figures above the bars.

Reviewer #3 (Remarks to the Author):

The authors constructively dealt with my questions and remarks concerning the previous version of this manuscript. I appreciate their effort in providing clarifications and new data. Given the broad ranging sampling and analyses presented, points of divergent interpretation may not be fully resolved in a peer review process. I admit that the revised manuscript gives important perspectives on the issues concerning Barret's oesophagus. There seems one perplexing anomaly though, the case of Meckel's diverticulum (Figure 1C, ileum samples). In this case a robust fold change of Hoxa13 gain of expression was found, the largest of all that was measured in this set of analyses. The direction of epithelial metaplasia however was of an anterior rather than posterior character. This seems difficult to fit into the explicative context invoking the overall role of HOXA13 in promoting posterior identity. Jozsef Zakany

Reply: Thank you for the positive comments and noticing this important exception. Indeed, in case of Meckel's diverticulum, the direction of the changes was opposite to the all others tissues. We acknowledge this in the text now (page 4 and line 88).

Reviewer #4 (Remarks to the Author):

I have read both the revised manuscript and the comments of the authors. I think they have addressed everything satisfactory. I recommend publication of this work.

Reply: We thank the reviewer for their time and effort assessing our manuscript.

Reviewers' Comments:

Reviewer #2:

Remarks to the Author:

The manuscript is improved and the outstanding comments have been largely addressed. I have two further comments:

i) HOXA13 staining in figure 4c is rather faint (especially when compared with 4b) but it is there (when compared to properly negative samples). It should also be born in mind that this has been shown for a single sample only.

ii) With regards to the triple K7/K5/p63 cells in the SMG, I am not convinced that the correct structure is shown. In the rebuttal on image B (page 3) there is a clear SMG (stained in green for K7) that does not have K5/p63 staining. A magnified view of region B is shown and there appears to be a duct in this image (page 6) that is surrounded by something else (that does not look like SMG in my opinion). It might be ductal squamous hyperplasia or oncocytic glands. I think it would be good to show the H&E on page 6 or provide another line of evidence.

REVIEWERS' COMMENTS

Reviewer #2 (Remarks to the Author):

The manuscript is improved and the outstanding comments have been largely addressed. I have two further comments:

i) HOXA13 staining in figure 4c is rather faint (especially when compared with 4b) but it is there (when compared to properly negative samples). It should also be born in mind that this has been shown for a single sample only.

Reply: Indeed, in the figure 4c (and 4e) the intensity of the signal is lower comparatively to 4b but dots are clearly distinct from the background. One dot represents a single RNA target regardless of intensity or size (<https://acdbio.com/science/how-it-works>), although some larger dots can be explained by clusters resulting from overlapping signals from multiple mRNA molecules. Thus, we believe this signal is sufficient to demonstrate the presence on *HOXA13* mRNA in this slide. We agree with the reviewer that this was shown for the single sample in which SMG were present as supportive information to single cell RNA seq data and we clearly stated this in the text “Esophageal submucosal glands were present in one sample and were HOXA13 positive” (page 5).

ii) With regards to the triple K7/K5/p63 cells in the SMG, I am not convinced that the correct structure is shown. In the rebuttal on image B (page 3) there is a clear SMG (stained in green for K7) that does not have K5/p63 staining. A magnified view of region B is shown and there appears to be a duct in this image (page 6) that is surrounded by something else (that does not look like SMG in my opinion). It might be ductal squamous hyperplasia or oncocytic glands. I think it would be good to show the H&E on page 6 or provide another line of evidence.

Reply: We thank the reviewer for this comment. We chose area B for magnification because this analysis was performed on the same block a few sections below the submucosal glands demonstrated to be positive for *HOXA13* in figure 4c positive for *HOXA13*. We did this, as the aim of the experiment was to investigate whether *HOXA13*-positive cells are co-localized in the same region as K7/K5/p63 cells. While the image in 4c clearly represents SMG as confirmed by expert pathologist, we indeed neglected to have this verified for the section where we performed triple positive staining. Upon reanalysis of this image our pathologist agrees with the reviewer that ductal structures may be present, but he also sees a part which he identifies as SMG. At the very least, this analysis indicates that the triple-positive cells and *HOXA13* reside within the same submucosal glandular structure. However, we have now also magnified another part of image B which the reviewer agrees has clear SMG (and which was not yet present on the section on the figure 4c). We can see that although the strong green signal (KRT7) masked K5/p63 staining in the overview image, those cells are clearly triple-positive under closer examination (signal intensity adjusted). We also provide H&E staining of the area. Please see images below. In addition, we also found structures in other sections that our pathologist indicates are consistent with seromucinous glands in the submucosa and that are triple positive for these markers (not included in the current manuscript). Thus, we feel confident to conclude that SMG contain K7/K5/p63 triple positive cells.

A.

Figure 1 for the editor and reviewer. **A)** overview of stained tissue Red: p63, Green: KRT7 Blue: DAPI White: KRT5. Red square shows the area chosen for the magnification. **B)** H&E confirming the presence of SMG. **C)** SMG positive for p63/KRT5/KRT7. **D)** magnification of image shown in C. **E).** magnified H&E image shown in B.